# Epigenetic divergence during early stages of speciation in an African crater lake cichlid fish

Grégoire Vernaz [1,2,3] ✉, Alan G. Hudson [4,8], M. Emília Santos[5], Bettina Fischer[2], Madeleine Carruthers [4], Asilatu H. Shechonge[6], Nestory P. Gabagambi[6], Alexandra M. Tyers[7,9], Benjamin P. Ngatunga[6], Milan Malinsky [3,10], Richard Durbin [2,3], George F. Turner [7], Martin J. Genner [4,11] ✉ and Eric A. Miska [1,2,3,11] ✉

Epigenetic variation can alter transcription and promote phenotypic divergence between populations facing different environmental challenges. Here, we assess the epigenetic basis of diversification during the early stages of speciation. Specifically, we focus on the extent and functional relevance of DNA methylome divergence in the very young radiation of *Astatotilapia calliptera* in crater Lake Masoko, southern Tanzania. Our study focuses on two lake ecomorphs that diverged approximately 1,000 years ago and a population in the nearby river from which they separated approximately 10,000 years ago. The two lake ecomorphs show no fixed genetic differentiation, yet are characterized by different morphologies, depth preferences and diets. We report extensive genome-wide methylome divergence between the two lake ecomorphs, and between the lake and river populations, linked to key biological processes and associated with altered transcriptional activity of ecologically relevant genes. Such genes differing between lake ecomorphs include those involved in steroid metabolism, hemoglobin composition and erythropoiesis, consistent with their divergent habitat occupancy. Using a common-garden experiment, we found that global methylation profiles are often rapidly remodeled across generations but ecomorph-specific differences can be inherited. Collectively, our study suggests an epigenetic contribution to the early stages of vertebrate speciation.

The genomic basis of adaptive phenotypic diversification and speciation has been extensively studied but many questions remain[1–3]. Recent studies in plants and animals provided initial evidence for a contribution of heritable epigenetic divergence to functional phenotypic traits[1,4–14], including DNA methylation, non-coding RNAs and histone posttranscriptional modifications. However, whether epigenetic

[1]Wellcome/Cancer Research UK Gurdon Institute, University of Cambridge, Cambridge, UK. [2]Department of Genetics, University of Cambridge, Cambridge, UK. [3]Wellcome Sanger Institute, Hinxton, UK. [4]School of Biological Sciences, University of Bristol, Bristol, UK. [5]Department of Zoology, University of Cambridge, Cambridge, UK. [6]Tanzania Fisheries Research Institute, Dar es Salaam, Tanzania. [7]School of Natural Sciences, Bangor University, Bangor, UK. [8]Present address: School of Life Sciences, University of Hawai'i at Mānoa, Honolulu, HI, USA. [9]Present address: Max Planck Institute for Biology of Ageing, Cologne, Germany. [10]Present address: Institute of Ecology and Evolution, University of Bern, Bern, Switzerland. [11]These authors contributed equally: Martin J. Genner, Eric A. Miska. ✉e-mail: gv268@cam.ac.uk; m.genner@bristol.ac.uk; eam29@cam.ac.uk

**Fig. 1 | Whole-genome DNA methylation landscape of the *Astatotilapia* cichlid radiation in Lake Masoko. a**, Map of Lake Masoko/Kisiba, Tanzania (modified from www.d-maps.com). **b,c**, Dissolved oxygen ($O_2$) concentration (%) and water temperature (°C) by depth (metres, m) in Lake Masoko. The oxy- (**b**) and thermo-clines (**c**) separate the habitats of the two Lake Masoko *A. calliptera* ecomorphs: the littoral (yellow) population thrives in shallow (≤5 m), well-oxygenated waters, while the benthic (blue) population is found in deeper, colder and less oxygenated habitats of the lake. Data from Delalande[17] (locally estimated scatterplot smoothed curves). **d**, Violin plots of stable isotope ratios ($\delta^{13}C_{V-PDB}$, ‰) by population indicate a significantly more offshore zooplankton-based diet for the benthic fish. Two-sided adjusted *P* values for Games–Howell multiple comparison tests using the Tukey's method are shown together with mean differences and 95% CI mean differences (5,000 bootstrap resamples; Supplementary Notes). *n*, number of fish per population. **e,f**, PCA (PC1 and PC2) of liver methylome (mCG) variation using both the RRBS (**e**) and WGBS (**f**) datasets. The principal component scores in (**e**) significantly segregate the three populations apart (two-sided *P* value for MANOVA tests; Extended Data

Fig. 2c). The percentage of total variance is given in parentheses. The asterisks in (**e**) indicate samples used for WGBS. *n*, number of biological replicates per population. **g**, Unbiased hierarchical clustering and heatmap of the average DNA methylation levels at all significant DMRs found between the three pairwise comparisons (numbered 1–3) reveal population-specific methylome patterns. mCG/CG levels (%) averaged by population over each DMR (Methods). The total number of DMRs for each comparison is shown on the left-hand side of each heatmap. **h**, Histogram of the closest distances in bp (log scale; median, dotted line) between DMRs and HDRs, when on the same chromosome. **i**, Enrichment plots (observed/expected ratio; expected values calculated from 500 random resampling data) for methylome variation (DMRs) in different genomic features for each pairwise comparison (the categories 'promoter', 'gene body' and 'intergenic' are mutually exclusive). Chi-squared tests and one-sided *P* values are shown above the graphs. Repeat, transposon-only repeats. **j**, GO enrichment analysis for the genes associated with methylome divergence among populations (either in promoter, gene body or intergenic regions).

processes, and in particular DNA methylation, facilitate adaptive diversification, especially during the early stages of speciation, is unknown.

To investigate the potential role of epigenetic processes during vertebrate speciation, we focused on the incipient *Astatotilapia* radiation in crater Lake Masoko, southern Tanzania, which is within the Lake Malawi catchment (Fig. 1a and Extended Data Fig. 1a–c)[15]. The two *Astatotilapia* ecomorphs present are characterized by different depth preferences, male breeding colors, morphology and

diets[15,16]. Specifically, the littoral ecomorph (yellow males) occupies the well-oxygenated shallow waters (≤5 m) and has a diet of littoral macroinvertebrates, while the benthic ecomorph (blue males) thrives in deeper (20–30 m), less oxygenated, dimly lit habitats of the lake and has a zooplankton-rich diet (Fig. 1b–d and Extended Data Fig. 1c)[17]. Previous genome-level sequencing revealed overall very low sequence divergence (fixation index, $F_{ST} = 0.038$) between the ecomorph pair[16] and suggests that the colonization and adaptation to

the benthic habitat from littoral fish took place approximately 1,000 years ago (200–350 generations)[16]. Additionally, elevated genetic differentiation ($F_{ST} > 0.3$) has been found in 98 genomic regions (highly diverged regions [HDRs]), which were enriched for functional targets of divergent selection related to vision, morphogenesis and hormone signalling[2,16]. More recent work revealed the presence of 'hybrid' individuals of varying levels of admixture[18], demonstrating incomplete reproductive isolation, and consistent with these ecomorphs being in the early stages of divergence.

Collectively, the two ecomorphs are monophyletic with regard to *Astatotilapia* in the neighbouring Mbaka River system (Extended Data Fig. 1a), from which they separated approximately 10,000 years ago. Moreover, there are significant differences in δ[13]C stable isotope ratios of both ecomorphs compared to the riverine fish (Welch's analysis of variance [ANOVA], $F_{2,14.83} = 111.56$, $P = 1.2 × 10^{-9}$ with all post-hoc tests $P < 0.008$; Fig. 1d and Supplementary Table 4), which is consistent with both the benthic and littoral lake habitats being fundamentally different from riverine habitats in ecological characteristics. Against this background, the incipient *Astatotilapia* radiation of Lake Masoko and nearby riverine population, from which they likely originate, provides a valuable system to investigate the role of epigenetic processes during the early stages of ecological speciation. In this study, we combined reduced-representation bisulphite sequencing (RRBS), whole-genome bisulphite sequencing (WGBS) and whole-transcriptome sequencing (RNA sequencing, RNA-seq) to assess functional methylome divergence in the two *Astatotilapia* ecomorphs of Lake Masoko and their neighbouring riverine population.

## Methylome divergence during the early stages of speciation

To compare genome-wide population-level methylome divergence between the two incipient populations of Lake Masoko and in the neighbouring riverine population, we combined two bisulphite sequencing approaches that enabled us to consider variation both between populations and at whole-genome scale resolution. First, we generated RRBS data from liver—a highly homogenous tissue involved in dietary metabolism, hormone production and hematopoiesis, therefore highly relevant for ecological diversification. In total, 12 wild-caught adult males for each of the littoral and benthic *Astatotilapia* ecomorphs of Lake Masoko and 11 wild-caught adult males from the neighbouring Mbaka River were used to generate the RRBS dataset. On average, 11.1 ± 3.4 (mean ± s.d.) million single-end 75 base pair (bp)-long reads were generated across the RRBS samples. Next, we produced WGBS data from liver tissue of at least two individuals from each of the three populations. On average, 341 ± 84.3 million paired-end 150 bp-long reads were generated per individual for this WGBS dataset (Supplementary Tables 1 and 2 and Methods). The different populations showed similar read mapping rates to the reference genome assembly (*Maylandia zebra* GCF_000238955.4; Methods), consistent with low interpopulation sequence divergence (Extended Data Fig. 2a,b and Methods)[16]. Principal component analysis (PCA) of both RRBS and WGBS datasets revealed strong methylome segregation among the three populations on the first two axes of variation (PC1 and PC2; Fig. 1e,f). At the population level (RRBS data), PC1 and PC2 significantly segregated the 3 populations (multivariate analysis of variance [MANOVA], $F_{4,62} = 47.97$, $P < 2.2 × 10^{-16}$; all post-hoc Games–Howell multiple tests with Tukey's adjusted $P$ values <0.001; Extended Data Fig. 2c), which is also broadly reflected at the whole-genome scale using the WGBS samples.

Since functional methylation variation tends to occur over neighbouring CG dinucleotide sites, we next identified significantly differentially methylated regions (DMRs; Methods) between each pair of populations for the WGBS datasets. In total for the WGBS data, we found 5,244 DMRs between the riverine and littoral fish, 4,594 between riverine and benthic fish, and 341 between littoral and benthic fish (Fig. 1g and Extended Data Fig. 3a). Methylome divergence inferred

at the genome-wide level using the WGBS dataset was highly correlated with divergence for the same genomic regions using the population-scale RRBS dataset (Extended Data Fig. 4a,b), highlighting that the WGBS dataset closely recapitulates methylome variation at the population level while providing genome-wide resolution (Supplementary Notes). Given the substantially higher coverage of the genome provided by the WGBS data relative to the RRBS data, we focused further analyses on the DMRs identified using the WGBS data.

Most of these WGBS DMRs (79% and 63.5% total DMRs, respectively) showed a substantial gain in methylation (gain DMRs; ≥44% methylation increase, median values) in the littoral or benthic fish compared to neighbouring riverine fish, respectively, with median methylation levels at gain DMRs of ≥72% mCG/CG (Fig.1g and Extended Data Fig. 3b,c). Between littoral and benthic fish, two-thirds showed an increase in methylation in the benthic fish population. In all pairwise comparisons, DMRs varied in length from 50 bp to 3 kilobase pairs (kbp) (median, 250 bp), in CG site counts from 4 to 232 (median, 15 CG sites) and they were distributed across all chromosomes (Extended Data Fig. 3d,e). Next, to investigate the relationship between the underlying genetic polymorphism and methylome divergence, regions of elevated genetic differentiation (HDR, $F_{ST} ≥ 0.3$)[16] between the littoral and benthic populations were examined. These regions showed high methylome conservation and were not colocalized with DMRs. Notably, the distances between HDRs and the nearest DMRs ranged from 15.9 kb to 31.1 megabases (Mb) (median, 5.3 Mb), suggesting that large regions of high genetic differentiation were in general not associated in *cis* with epigenetic divergence between the two incipient Masoko ecomorphs (Fig. 1h and Extended Data Fig. 3f), although any *trans*-acting effect of genetic variation on epigenetic variation cannot be ruled out.

We next examined the genomic localization of DMRs among populations and found promoter regions to be highly enriched in DMRs in all comparisons (greater than twofold enrichment), consistent with their *cis*-regulatory functions[19] (Fig. 1i). In the littoral-benthic comparison, the differences in methylomes were particularly overrepresented in CpG-dense regions (CpG islands; ≥4.5-fold enrichment) located both within (promoter CpG islands [CGIs]) and outside promoters (orphan CGIs). Orphan CGIs may represent distant *cis*-regulatory regions, such as ectopic promoters and enhancers[20], both of which are known targets of methyl-sensitive DNA-binding proteins[19,21]. In contrast, methylome variation within intergenic regions and transposable elements only showed slight enrichment in the comparison of the Masoko ecomorphs relative to comparisons involving the riverine population. Additionally, the methylome divergence in gene bodies—the function of which remains unclear in vertebrates but could be involved in alternative splicing[19]—was generally very low and even highly underrepresented (2.5-fold depletion) in the comparison of Masoko ecomorphs. This suggests high methylome conservation between gene bodies of the most closely related populations.

We then identified biological processes associated with methylome divergence by performing gene ontology (GO) enrichment analysis (Fig. 1j and Methods). We first noted that genes involved in transcription regulation were highly enriched in DMRs and then identified three further sets of biological processes for the genes enriched in methylome divergence: (1) immune function and hematopoiesis; (2) embryogenesis and development; and (3) metabolism (Fig.1j and Extended Data Fig. 5a). Notably, methylome divergence in developmental genes was shown previously to account for close to half of all species-specific epigenetic differences among three species part of the Lake Malawi cichlid radiation[9]. Additionally, regions showing benthic-specific methylome patterns and located in promoters, gene body and intergenic regions were significantly enriched for specific transcription factor binding motifs (Extended Data Fig. 5b; $P$ values derived from HOMER for motif enrichment based on cumulative hypergeometric distributions; see Methods) with functions associated with erythropoiesis, immune functions, development and liver metabolism, consistent with the

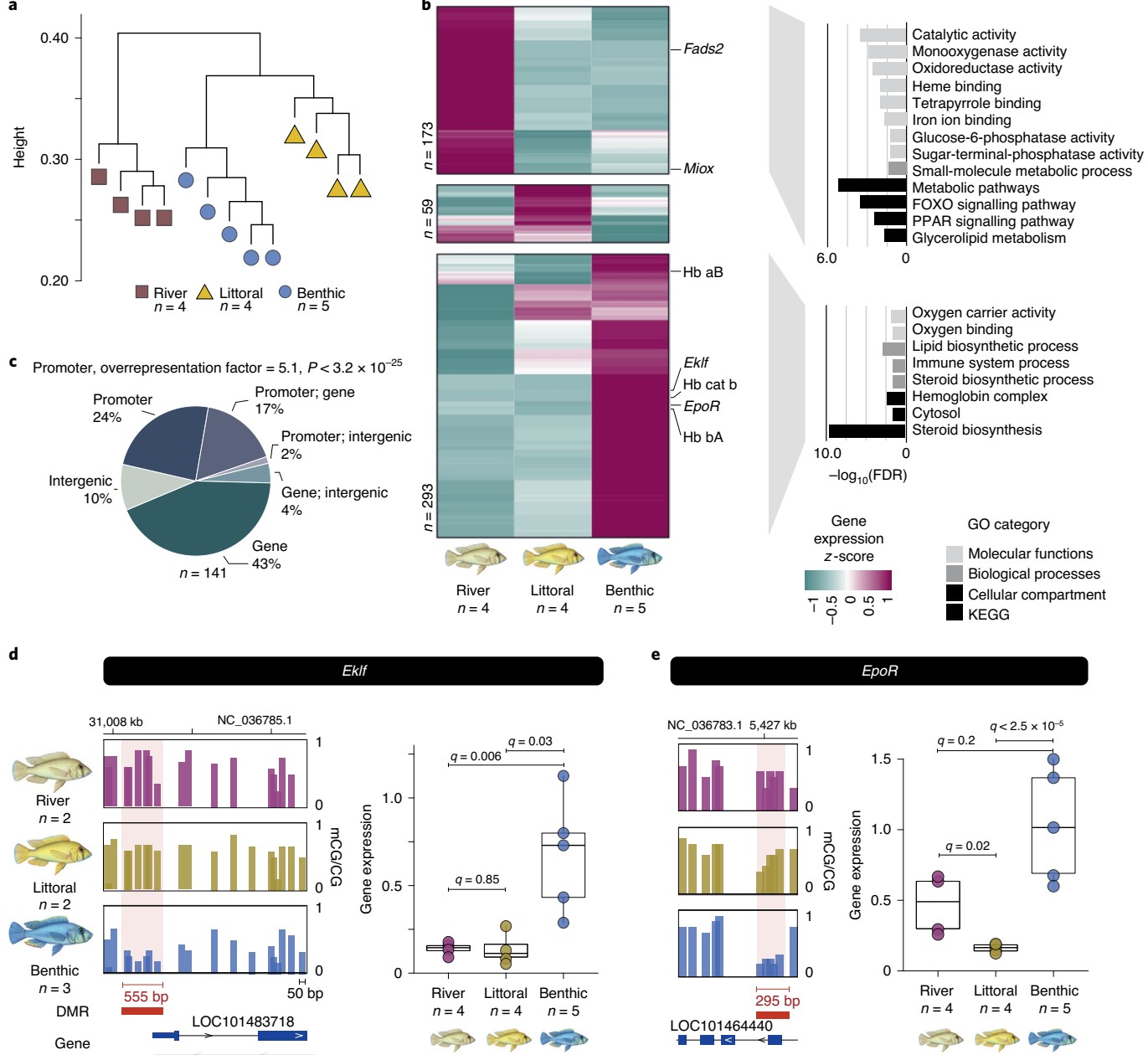

**Fig. 2 | Methylome variation is associated with altered transcriptional activity of genes related to hematopoiesis, erythropoiesis and fatty acid metabolism. a**, Unbiased hierarchical dendrogram based on whole-transcriptome variation (Euclidean distances), highlighting population-specific transcription patterns. **b**, Unbiased hierarchical clustering and heatmap of transcriptional activity (z-score, row-scaled) for all significant DEGs among the three wild populations, showing three different clusters of transcriptional activity (Wald test FDR-adjusted two-sided $P$ value using Benjamini–Hochberg <0.05, fold change ≥2 and high gene expression [top 90th percentile] in any one sample). Examples of DEGs are shown on the right-hand side of the graph. Right: GO enrichment for the DEGs from each of the three transcriptional activity clusters (only FDR < 0.01 is shown). KEGG, Kyoto Encyclopedia of Genes and Genomes. **c**, Pie chart representing the genomic localization of DMRs associated with DEGs. Significant overlap between DMRs at promoter and

transcriptional changes (overrepresentation factor = 5.1; exact hypergeometric test, two-sided $P < 3.2 \times 10^{-25}$). DEGs can be associated with multiple DMRs in different locations (promoter, intergenic and genic DMRs). **d,e**, The promoters of the *Eklf* (**d**) and *EpoR* (**e**) genes, both involved in erythropoiesis and red blood cells differentiation, show hypomethylation levels in the livers of benthic fish compared to the littoral populations. The genome browser view of the methylome profiles for each ecomorph is shown. Each bar represents the average mCG/CG levels in 50 bp-long non-overlapping windows for each ecomorph population. DMRs are highlighted in red and their lengths are indicated in red. Right-hand side of (**d**) and (**e**): box plots of gene expression in liver of benthic, littoral and river fish for *Eklf* (**d**) and *EpoR* (**e**) are shown ($q$ values: Wald test FDR-adjusted two-sided $P$ values using Benjamini–Hochberg <0.05). All box plots indicate the median (middle line), 25th and 75th percentiles (box) and 5th and 95th percentiles (whiskers), as well as outliers (single points).

biological pathways associated with genes enriched for DMRs (Fig.1j). Transcription factors included the hematopoiesis-related stem cell leukemia (SCL)/T-cell acute lymphocytic leukemia 1 (TAL1; hypergeometric test, $P = 1 \times 10^{-145}$), forkhead hepatocyte nuclear factor 3-alpha

(FOXA1; hypergeometric test, $P = 1 \times 10^{-9}$), the embryogenesis-related mothers against decapentaplegic homolog 2 (SMAD2; hypergeometric test, $P = 1 \times 10^{-125}$) and the metabolic/circadian clock transcriptional activator aryl hydrocarbon receptor nuclear translocator-like protein

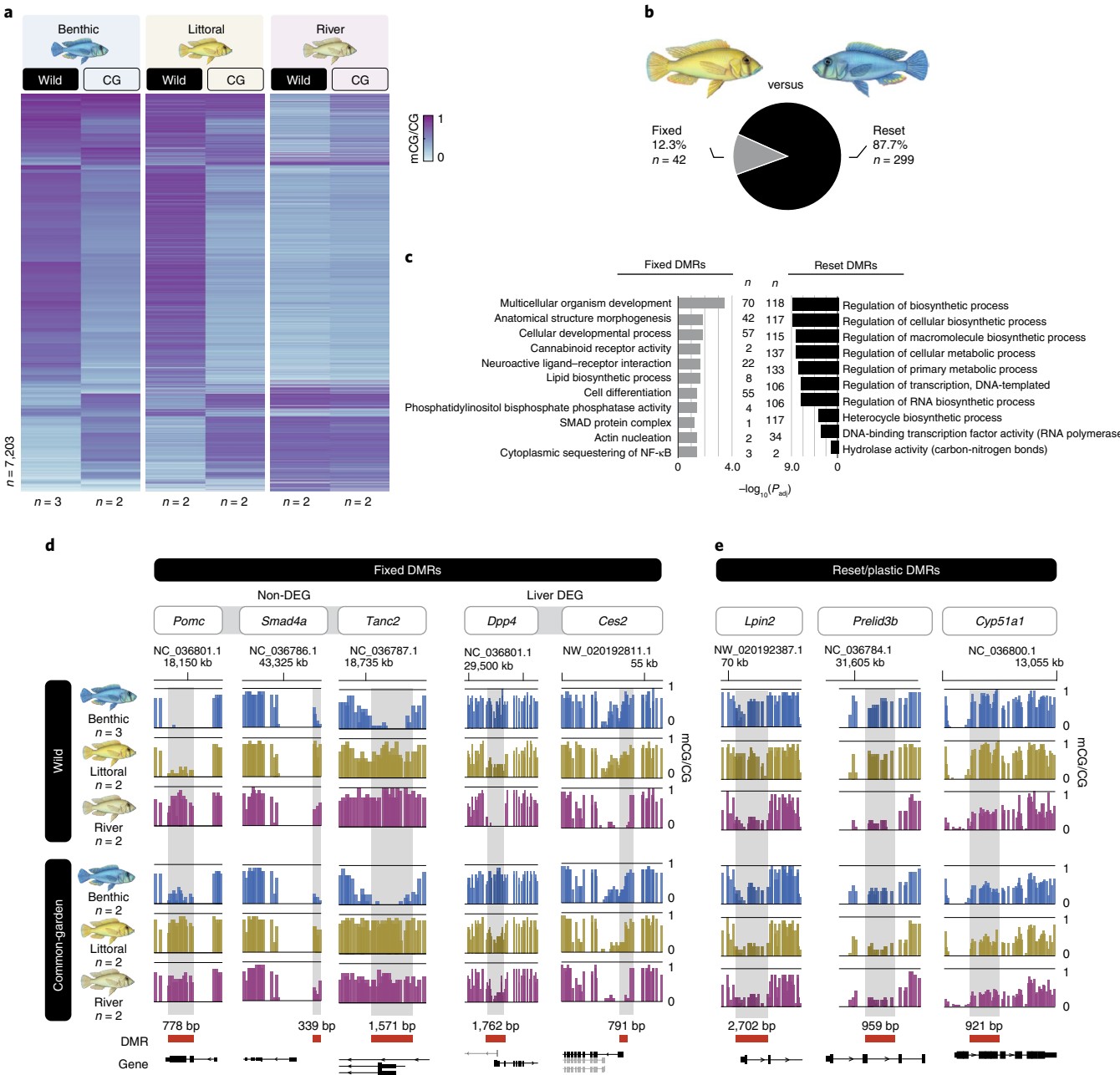

**Fig. 3 | Common-garden experiment results in both global resetting of methylome profiles in wild benthic and littoral fish to resemble riverine methylome profiles and inheritance of fixed methylome differences. a**, Heatmap of the average DNA methylation levels (mCG/CG, %) at all DMRs found in wild populations in Fig. 1g for all wild and common-garden fish. Methylome profiles revealed global epigenetic resetting in wild benthic and littoral fish to resemble neighbouring river fish methylome profiles, which were mostly unaffected by environmental perturbation. **b**, The proportion of reset (on common-garden experiment and within one generation) and population-fixed DMRs between littoral and benthic fish. See Extended Data Fig. 9a for the other pairwise comparisons. **c**, GO analysis showing significant enrichment for fixed

and reset DMRs in genes involved primarily in developmental and metabolic processes, respectively. **d,e**, Examples of DMRs fixed between populations (**d**) in wild and common-garden fish, with some fixed DMRs also associated with altered transcriptional activity in the liver (liver DEG; using Wald test FDR-adjusted two-sided *P* value using Benjamini–Hochberg <0.05; see Extended Data Fig. 9f–h for *P* values associated with each DEG) and of DMRs reset on the common-garden experiment, all associated with population-specific transcriptional differences (**e**). Each bar represents the average mCG/CG levels in 50 bp-long non-overlapping windows for each fish population (*n* ≥ 2 biological replicates). DMRs are highlighted in red and the length (bp) of each DMR is indicated in black.

1 (BMAL1; hypergeometric test, $P = 1 \times 10^{-8}$), consistent with altered transcription factor activity. This suggests *cis*-regulatory functions for such population-specific DMRs in development, hematopoiesis and metabolism, possibly correlated with acclimation to the benthic habitat. It is well established that differential methylation in promoter

regions might impact the activity of methyl-sensitive transcription factors, therefore resulting in an altered transcriptional landscape[21]. Differential expression of transcription factor genes and genetic variation in transcription factor genes or transcription factor binding sites can in turn also affect promoter methylation[19,22]. Although many

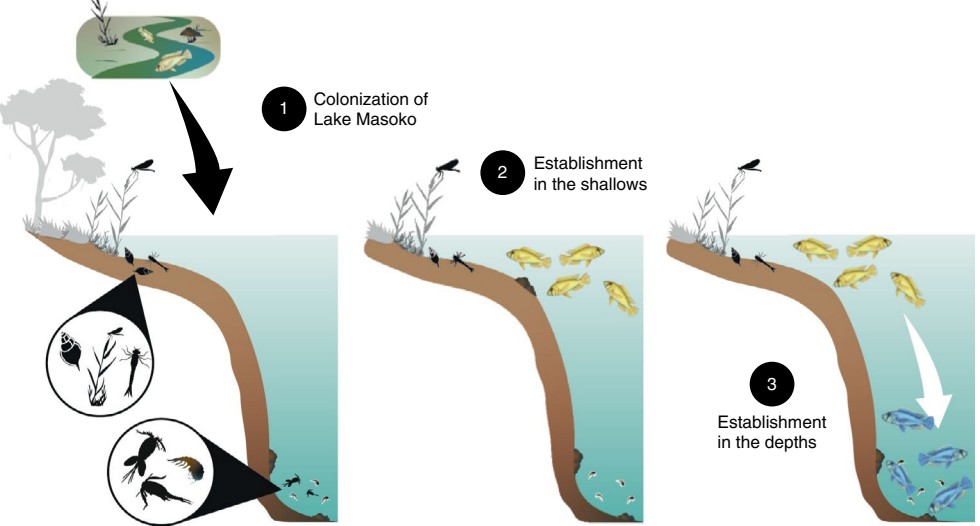

**Fig. 4 | Hypothesized three stages of epigenetically associated *Astatotilapia* ecomorph speciation in Lake Masoko.** (1) Colonization of the shallow habitats of Lake Masoko by the generalist riverine *Astatotilapia* population approximately 10,000 years ago (Malinsky et al.[16]). (2) Occupancy of shallow, reedy and highly oxygenated habitats by fish with a high level of depth philopatry. Phenotypic plasticity, partially linked to global methylome changes, enables utilization of littoral macroinvertebrate prey. (3) Colonization of deep, zooplankton-rich and lowly oxygenated habitats by the shallow population approximately 1,000 years ago (Malinsky et al.[16]). Extreme methylome changes in the benthic population associated with diet (for example, fatty acid metabolism) and environment (for example, hemoglobin composition) are shown. Epialleles are reciprocally fixed in the two populations, plausibly leading migrants, and those of intermediate epi-genotypes, to suffer a fitness disadvantage. Eventually, selection leads to differential fixation of genomic variation.

transcription factor genes are differentially expressed between the three populations and are associated with methylome differences, suggesting an altered transcription factor activity landscape arising from population-specific methylome divergence, further work is needed to decipher the underlying mechanisms.

## Methylome divergence is associated with transcriptional changes

Transcriptional variation can underlie ecological diversification[23,24]; however, the role of epigenetics in facilitating rapid transcriptional divergence in the context of the early stages of speciation is currently unknown. We investigated the link between population-specific methylome divergence and transcriptional activity by generating total RNA-seq data from the liver tissues of 4–5 individuals for each population (mean ± s.d., 32.9 ± 3.8 million paired-end 100/150 bp-long reads per fish; Supplementary Table 3 and Supplementary Notes). As with whole-methylome variation, we observed population-specific transcription patterns across all samples (Fig. 2a and Extended Data Fig. 6a). Moreover, methylation levels at promoter regions were significantly negatively correlated with transcriptional activity overall (Spearman's rank correlation rho = −0.33, $P < 2.2 \times 10^{-16}$; Extended Data Fig. 6b), confirming the association between DNA methylation and gene expression repression in vertebrates[9,25]. We then performed differential gene expression analysis (false discovery rate [FDR] adjusted $P$ values using Benjamini–Hochberg <1%, fold change ≥1.5 and high gene expression in ≥1 population [top 90th percentile]; Methods) and found a total of 525 significantly differentially expressed genes (DEGs) between the three populations, including 119 genes that were differentially expressed between the Masoko ecomorphs (Fig. 2b). Close to 33% of all DEGs showed reduced transcriptional activity in Masoko ecomorphs relative to the riverine population and these genes were significantly enriched for functions related to energy balance/homeostasis and steroid metabolism, including the peroxisome proliferator-activated receptor (PPAR) and forkhead box O (FOXO) signalling pathways, consistent with metabolic adaptation to different diets (Fig. 2b). Conversely, almost all the remaining DEGs (56%) showed high transcriptional activity almost exclusively in benthic fish

and were primarily enriched for functions associated with hemoglobin complex/oxygen-binding activities and iron homeostasis as well as with fatty acid metabolism, in line with the occupation of a hypoxic environment and possibly related to their zooplankton-rich diet (Fig. 1a,d). Critically, significant changes in transcriptional activity were strongly associated with methylome divergence at promoter regions (43% of all DEGs, overrepresentation factor = 5.1, exact hypergeometric probability: $P < 3.2 \times 10^{-25}$; Fig. 2c).

Focusing on the functional categories identified above, we examined in detail several examples of transcriptional diversification associated with divergence in methylome landscapes, in particular for genes showing a negative correlation between methylation levels at promoters and gene expression activity (Extended Data Fig. 6b). First, loss of methylation in two genes associated with active hematopoiesis and erythrocyte differentiation, *Ekfl* and *EpoR*, was associated with significant gain of transcriptional activity in benthic fish compared to the littoral population (fold change ≥ 3.8, FDRs < 0.03; Fig. 2d,e). *Eklf* plays an essential role in erythropoiesis, heme synthesis and in the modulation of globin gene expression[26]. Its transcriptional activity has been previously linked to methylation changes in humans[27]. Moreover, three hemoglobin subunit (Hb) genes, adjacent to each other within the MN (major) globin cluster on chromosome LG4 (NC_036783.1) were significantly upregulated (≥4.1-fold) in benthic fish specifically (FDR ≤ 0.006; Extended Data Fig. 7a). This includes the cathodic Hb beta, which is known to have higher oxygen-binding affinity in anoxic environments and is expressed exclusively in benthic fish[28]. Notably, a 2 kbp-long DMR adjacent to this globin gene cluster showed benthic-specific hypermethylation compared to the intermediate and unmethylated levels seen in littoral and river fish, respectively (Extended Data Fig. 7b). This suggests that it bears *cis*-regulatory functions, possibly similar to the vertebrate locus control regions, known to be bound by many essential erythroid transcription factors, such as EKLF, SCL/TAL1 and GATA1 (refs. [29,30]), whose methylation state has been linked to globin gene expression regulation in mammals[31]. Although methylation at promoters is generally associated with weaker transcriptional activity in vertebrates (refs. [9,25] and Extended Data Fig. 6b), certain classes of methyl-sensitive transcription factors require methylated binding

sequences to activate transcription[21]. Further work is therefore required to demonstrate the affinity of such transcription factors for methylated binding sequences at the putative cichlid locus control regions and their association with increased Hb transcription, which is currently unknown in cichlids. Finally, the hematopoietic transcription factor gene *Scl*/*Tal1*, a major actor in red blood cell differentiation[29,32], whose sequence recognition binding motif is highly enriched in DMR sequences (Extended Data Fig. 5b), is also expressed in benthic fish only (FDR < 0.013; Extended Data Fig. 7a), suggesting altered transcription factor activity associated with methylome differences in benthic fish. Collectively, these results suggest significant divergence in epigenetic and transcriptional landscapes affecting erythropoiesis and hemoglobin composition in benthic fish, which may facilitate occupation of anoxic conditions of the benthic habitat.

As noted above, the Lake Masoko ecomorph populations are mainly characterized by an overall reduced transcriptional activity in many genes related to steroid metabolic pathways and energy homeostasis compared to riverine fish (Fig. 2b). Many of these genes show significant gain in methylation compared to the riverine population, in line with a general repressive role for DNA methylation in transcription regulation (Extended Data Fig. 6b). The examples of genes we examined include *Fads2*, part of the PPAR signalling pathway with functions in fatty acid metabolism and dietary metabolic adaptations[33], and the catabolic enzyme inositol oxygenase (*Miox*). Both genes show hypermethylation associated with weaker transcriptional activity in Masoko ecomorphs compared to neighbouring riverine fish (Extended Data Fig. 7c,d). The interplay between metabolism and epigenetic variation has been well established[7,33,34] and suggests that epigenetic divergence may have facilitated differences in dietary resource use patterns during colonization of Lake Masoko habitats.

## Plasticity and inheritance of methylome divergence

We examined the plasticity and inheritance of population-specific methylome divergence found between wild populations using a common-garden experiment, whereby wild-caught *A. calliptera* specimens from Lake Masoko (both littoral and benthic populations) and from the neighbouring Mbaka River system, were bred and first-generation fish were reared under the same controlled laboratory conditions (Methods). Liver methylomes (WGBS) were then generated for two common-garden individuals of each population (Extended Data Fig. 8a,b). Under a common rearing environment and within 1 generation, most DMRs (88.8%) found between any wild populations of Lake Masoko were reset (that is, no longer significant DMRs between the respective common-garden groups) to resemble mostly unchanged methylome levels of river fish (reset DMRs). The remaining 11.2% of DMRs were retained/fixed between populations (fixed DMRs, that is, statistically significant DMRs showing the same directionality in methylation differences between wild and common-garden fish for each respective comparison; Fig. 3a,b and Extended Data Figs. 8c and 9a), consistent with a potential transgenerational retention of population-specific methylome patterns. Moreover, reset DMRs were on average almost twice as long as fixed DMRs (median: 416 and 227 bp, respectively, with a mean difference of 242 bp [95% confidence interval, 208–281]; Extended Data Fig. 9b), suggesting potential functional differences associated with them. We then performed GO enrichment analysis and found that reset DMRs were significantly enriched in the promoter of genes with functions related to liver metabolism, transcription and DNA-binding activity (Fig.3b). In total, 85.7% of all DEGs found among the livers of wild populations were associated with reset methylome patterns on environmental perturbation (Extended Data Fig. 9c). Such genes include, for example, the ones coding for *Lpin2* involved in fatty acid metabolism, *Preli3b* implicated in phospholipid transport, and *Cyp51a1* with functions in sterol biosynthesis (Fig. 3e), suggesting a close link between environmental conditions

and methylome divergence associated with altered metabolic processes. Furthermore, the erythropoietic genes *Eklf* and *EpoR*, which have both benthic-specific methylome and transcriptome patterns in wild fish, all resembled river and littoral highly methylated profiles in the common-garden experiment (Extended Data Fig. 9d).

While most of population-specific methylome patterns in wild populations showed high levels of plasticity, highlighting the tight interaction between environmental conditions and epigenetic variation, methylome patterns fixed between populations showed an overall significant association with genes related to development, embryogenesis and cell differentiation, in particular associated with brain development (Fig. 3c). Therefore, some fixed DMRs found between livers might represent population-specific, tissue-independent methylome patterns, similar to what has been previously shown in Lake Malawi cichlids[9]. Indeed, tissue-independent methylome divergence could reflect distinct core developmental processes between the populations, participating in early-life phenotypic differences, although methylome analysis of other somatic or embryonic tissues would be needed to further investigate their functions. Examples of genes with functions during brain development and neuron morphogenesis and showing fixed population-specific methylome patterns include *Smad4a* (hypomethylated in littorals only) and the one encoding the scaffolding protein TANC2 (hypomethylated in benthics) (Fig. 3d). Other pathways, related to biosynthetic processes among others are also associated with fixed population-specific methylome patterns (Fig. 3c). This includes the *Pomc* gene, a precursor polypeptide produced in the brain and involved in energy homeostasis and immune functions, showing retention of benthic-specific methylome patterns (Fig. 3d). Although a large fraction of all DEGs is associated with reset DMRs, we found some association between fixed population-specific methylome patterns and transcriptional divergence in liver-specific genes (14.3% of all DEGs), with functions in metabolic pathways and immune functions. Two genes in particular, coding for the enzymes *Dpp4* and *Ces2*, part of the insulin and fatty acid metabolic pathways, respectively, showed significant transcriptional downregulation in benthic livers and were associated with fixed benthic-specific hypermethylated levels at their promoters (Fig. 3d and Extended Data Fig. 9e–g), suggesting retained epigenetic divergence associated with different diets.

These results suggest that although methylome landscapes are highly environment-specific, showing high plasticity and significant association with altered transcriptional activity of functional genes, some methylome divergence has become fixed and may be inherited in populations of Lake Masoko (fixed DMRs; Fig. 3d). While in mammals two waves in DNA methylation reprograming occur early on, in zebrafish the paternal methylome is retained on fertilization, possibly allowing for transgenerational epigenetic inheritance[35]. The underlying epigenetic reprograming mechanisms have been shown to vary across teleost fish[36] and are currently unknown in cichlids. Our study lays the groundwork to investigate further the extent of the inheritance of epigenetic patterns in East African cichlids and assess any adaptive roles associated with methylome divergence. Future work will also evaluate any genetic basis acting in *trans* that could affect methylome variation, such as genetic polymorphism in transcription factor domain sequences[22].

## Conclusion

Our results provide direct and new evidence for functional and heritable methylome divergence associated with the early stages of speciation in the very young radiation (approximately 1,000 years ago) of *Astatotilapia* ecomorphs in Lake Masoko using whole-genome methylome sequencing. We suggest that colonization of the shallow lake habitat by a generalist riverine *Astatotilapia* population (approximately 10,000 years ago) was followed by the colonization of the benthic habitat (approximately 1,000 years ago). This was enabled in part by the

establishment of both reversible and heritable methylome divergence in key functional genes (Fig. 4), including those related to hemoglobin synthesis, erythropoiesis and sterol metabolism, possibly linked to the low oxygen and zooplankton-rich environment present in the depths of the lake (Fig.1b,d). This potentially greatly increased the fitness of benthic fish in the deep environment of the lake, enabling them to outcompete any later littoral intruders that did not possess the appropriate epigenetic variation. In principle, therefore, epigenetic processes may provide the capacity for rapid occupation and competitive dominance in new ecological niches before fixation of epigenetic and genomic variation.

To our knowledge, our study demonstrates for the first time substantial methylome divergence, in part inherited, and associated with altered transcription in a very young vertebrate radiation at an unprecedented whole-genome resolution. Therefore, this work further builds on evidence of epigenetic divergence seen among populations of fish[37–39], intraspecific methylome remodeling in different rearing environments[40,41], heritability of population-specific methylome divergence[38,42,43] and an epigenetic basis for diversification of functional eco-morphological traits in vertebrates (for example, the eyes of the cavefish *Astyanax mexicanus*[4]). Additionally, other epigenetic processes might be at play in parallel to DNA methylation in facilitating phenotypic diversification in teleost fish radiation, such as micro-RNAs[1,10] and histone posttranslational modifications[13], potentially affecting gene expression. Further work is required to investigate their functions, alongside DNA methylation, during the early stages of speciation in sympatric species. A key challenge now is to determine the mechanisms and rates of fixation of heritable epigenetic variation within populations[6,7,44], including epigenetic inheritance and reprograming, the extent of the adaptive advantage associated with methylome divergence, and how this associates with the genomic fixation observed during the later stages of speciation[4–8].

## Methods

### Field sampling
Lake Masoko fish were chased into fixed gill nets and SCUBA by a team of professional divers at different target depths determined by diver depth gauge (12× male benthic, 12× male littoral). Riverine fish (11× Mbaka River and 1× Itupi river) were collected by local fishermen. On collection, all fish were euthanized using clove oil. Collection of wild fish was done in accordance with local regulations and permits in 2015, 2016, 2018 and 2019. On collection, fish were immediately photographed with color and metric scales, and tissues were dissected and stored in RNA*later* (Sigma-Aldrich); some samples were first stored in ethanol. Only male specimens (showing bright nuptial coloration) were used in this study for the practical reason of avoiding any misassignment of individuals to the wrong population (only male individuals show clear differences in phenotypes and could therefore be reliably assigned to a population). Furthermore, we assumed that any epigenetic divergence relevant to speciation should be contributing to between-population differences in traits possessed by both sexes (habitat occupancy, diet). To investigate the role of epigenetics in phenotypic diversification and adaptation to different diets, homogenized liver tissue – a largely homogenous and key organ involved in dietary metabolism, hormone production and hematopoiesis – was used for all RNA-seq and WGBS experiments.

### Common-garden experiment
Common-garden fish were bred from wild-caught fish specimens, collected and imported at the same time by a team of professional aquarium fish collectors according to approved veterinary regulations of the University of Bangor, UK. Wild-caught fish were acclimatized to laboratory tanks and reared to produce first-generation (G1) common-garden fish, which were reared under the same controlled laboratory conditions in separate tanks (light–dark cycles, diet: algae

flakes daily, 2–3 times weekly frozen diet) for approximately 6 months (post hatching). G1 adult males showing bright nuptial colors were culled at the same biological stages (6 months post hatching) using MS222 in accordance with the veterinary regulations of the University of Bangor, UK. Immediately on culling, fish were photographed and tissues collected and snap-frozen in tubes.

### Stable isotopes
To assess dietary/nutritional profiles in the three ecomorph populations, carbon ($\delta^{13}$C) and nitrogen ($\delta^{15}$N) isotope analysis of muscle samples (for the same individuals as RRBS; 12, 12 and 9 samples for benthic, littoral and riverine populations, respectively) was undertaken by elemental analyzer isotope ratio mass spectrometry by Iso-Analytical Limited. It is important to note that stable isotope analysis does not depend on the use of the same tissue as the ones used for the RRBS/WGBS samples[45]. Normality tests (Shapiro–Wilk, using the R package rstatix v.0.7.0), robust for small sample sizes, were performed to assess sample deviation from a Gaussian distribution. Levene's test for homogeneity of variance was then performed (R package carData v.3.0-5) to test for homogeneity of variance across groups. Finally, Welch's ANOVA was performed followed by Games–Howell all-pairs comparison tests with adjusted *P* value using Tukey's method (rstatix v.0.7.0). Mean differences in isotope measurements and 95% CI mean differences were calculated using Dabestr v.0.3.0 with 5,000 bootstrapped resampling.

Throughout this manuscript, all box plots are defined as follows: centre line, median; box limits, upper and lower quartiles; whiskers, 1.5× interquartile range; points, outliers.

### RNA-seq
**Next-generation sequencing library preparation.** Total RNA from liver tissues stored in RNA*later* was extracted using a phenol/chloroform approach (TRIzol reagent; Sigma-Aldrich). Of note, when tissues for bisulphite sequencing samples were not available, additional wild-caught samples were used (Supplementary Table 3). The quality and quantity of RNA extraction were assessed using TapeStation (Agilent Technologies), Qubit and NanoDrop (Thermo Fisher Scientific). Next-generation sequencing (NGS) libraries were prepared using poly(A) tail-isolated RNA fraction and sequenced on a NovaSeq system (S4; paired-end 100/150 bp; Supplementary Table 3), yielding on average 32.9 ± 3.9 Mio reads.

**Read alignment and differential gene expression analysis.** Adaptor sequence in reads, low-quality bases (Phred score < 20) and reads that were too short (<20 bp) were removed using TrimGalore v.0.6.2 (options: --paired --fastqc; https://github.com/FelixKrueger/TrimGalore). Paired-end 150 bp sequencing read samples were trimmed to 100 bp (both read pairs) to account for read length differences using TrimGalore's options: --three_prime_clip_R1 50 --three_prime_clip_R2 50. Paired-end reads were aligned to the *M. zebra* reference genome (GCF_000238955.4_M_zebra_UMD2a_genomic.fa) using kallisto[46] v.0.46.0 (options: --bias -b 100), resulting in high mapping rates (83.9 ± 1.6%, mean ± s.d.). Using transcription levels at all annotated genes for all RNA-seq samples, unbiased hierarchical clustering was done using the R script pheatmap v.1.0.12 (Euclidean distances and complete-linkage clustering using Spearman's correlation matrix). Differential gene expression analysis was then carried out with sleuth v.0.30.0 (ref. [47]) using Wald's test with FDR-adjusted two-sided *P* value (Benjamini–Hochberg method). Only genes with a *q* value < 0.05, log$_2$ fold change ≥ 1.5 between any pairwise population comparison and showing high expression levels in ≥1 biological samples (maximal gene expression ≥10 transcripts per million (TPM) in any one sample, which represents the 91st percentile of gene expression in the benthic liver samples) were analysed further. Heatmaps of scaled gene expression values (*z*-score) for all DEGs were generated using pheatmap v.1.0.12 (Euclidean distances and complete-linkage clustering). Data for gene

expression values (TPM) across different *A. calliptera* tissues were used from Vernaz et al.[9].

### High-molecular-weight genomic DNA extraction

High-molecular-weight genomic DNA from liver tissues stored in RNA*later* was isolated using the QIAamp DNA Mini Kit (catalogue no. 51304; QIAGEN). The quality and quantity of the extracted DNA samples were assessed using TapeStation, Qubit and NanoDrop.

### WGBS

**NGS library preparation (wild and common-garden samples).** Unmethylated lambda phage genome (0.5% w/w) was first spiked in every sample (catalogue no. D1521; Promega Corporation). DNA samples were then fragmented to approximately 400 bp in length by sonication (E220 Focused-ultrasonicator, Covaris). The length and quality of DNA fragments were assessed using TapeStation. NGS libraries were prepared using approximately 400 ng sonicated DNA fraction using NEBNext Ultra II DNA Library Kit for Illumina (catalogue no. E7645; New England Biolabs) and methylated adaptors (catalogue no. E7535; New England Biolabs) according to the manufacturer's instructions. DNA libraries were then treated with sodium bisulphite (catalogue no. MOD50; Sigma-Aldrich Imprint) according to the manufacturer's instructions. Bisulphite-treated DNA libraries were then amplified by PCR (14 cycles) and sequenced as paired-end 150 bp reads on Illumina HiSeq 4000 and NovaSeq systems (the latter for 1 littoral and 1 benthic wild fish) to generate 322.02 ± 58.94 million paired-end reads per sample (mean ± s.d.).

**WGBS read mapping.** Adaptor sequence in reads, low-quality bases (Phred score ≤20) and reads that were too short (<20 bp) were removed using TrimGalore (options: --paired --fastqc). Sequencing reads (FASTQ) for the same sample generated on multiple lanes were merged. Paired-end reads were first mapped against the lambda genome (GenBank accession no. J02459) to assess bisulphite conversion (98.4 ± 1.0%, mean ± s.d. spike-in conversion rate for all wild and common-garden samples) and then to single-nucleotide polymorphism (SNP)-corrected version of the *M. zebra* reference genome (GCF_000238955.4_M_zebra_UMD2a_genomic.fa) to account for *A. calliptera*-specific genotype/SNP (following the same protocol as developed in Vernaz et al.[9]) using Bismark v.0.20.0 (options: -N 0 -p 4 -X 500)[48]. Mapping rates were similar across samples, yielding 54.7.1 ± 4.3% best unique read mapping (mean ± s.d., n = 13; Extended Data Fig. 2a), similar to the mapping rates observed in other WGBS studies[49]. Clonal paired-end reads (that is, PCR duplicates) were removed using Bismark's deduplicate_bismark function (options: -p --bam). Methylation scores (read count supporting mC/total read count) at each CpG site genome-wide were extracted using Bismark's bismark_methylation_extractor function (options: -p --multicore 6 --no_overlap --comprehensive --merge_non_CpG --bedGraph).

**WGBS DMR prediction.** DMR prediction was performed using DSS[50] v.2.34.0 (smoothing=TRUE). First, Wald tests were performed on methylation difference at all CG sites between any two populations. Predicted DMRs consisted of CG sites showing significant methylation differences (Wald test, two-sided $P < 0.05$). Then, to identify putatively biologically relevant DMRs (that is, *cis*-regulatory elements of a typical size, possibly bound by DNA-binding proteins), the following stringent cut-off parameters were chosen based on previous methylome studies[20,51,52]: only DMRs that showed substantial methylation differences (≥25% average methylation difference at any one DMR), covering ≥4 CG sites and with a minimal length of ≥50 bp were analysed further. Overall, predicted DMRs showed on average approximately 45% methylation differences (Extended Data Fig. 3b,c), ranged in length from 50 to 3,000 bp (median length, 250 bp; Extended Data Fig. 3d) and covered 4–232 CG sites (median, 15 CG sites; Extended Data Fig. 3d).

**WGBS methylome analysis.** For subsequent analyses, only CpG sites with ≥4 ≤ 100 unique (non-clonal) paired-end read coverage were used (genome-wide CG site coverage across all samples: 9.14 ± 1.4, mean ± s.e.m.; Extended Data Fig. 2b). Methylation scores at single CpGs were calculated using Bismark output files as follows: number of methylated reads/total number of reads. Methylation levels in non-overlapping 50 bp-long genomic windows for each biological sample or each population (averaged mCG/CG levels) were generated with BEDTools v.2.27.1 (ref. [53]) and visualized as BIGWIG files (bedGraphToBigWig v.4; https://genome.ucsc.edu/) in IGV genome browser v.2.9.2 (Broad Institute). PCA (centred and scaled) was carried out using R v.3.6.3 (prcomp) using all CG sites. Unbiased hierarchical clustering (complete-linkage clustering method) was carried out using R based on Euclidian distances (dist) of pairwise Spearman's correlation scores (cor). Heatmaps were created using pheatmap (complete-linkage clustering method using Euclidean distances). Circos plots were generated on R using circlize v.0.4.12 to visualize DMR genomic distribution across LG chromosomes only (NC chromosomes). Transcription factor binding motif enrichment analysis within DMR sequences was performed on DMRs both located outside gene bodies (excluding the first 1 kbp downstream transcription start site [TSS]) and in promoters/intergenic regions using HOMER v.4.9.1-6 ('findMotifs.pl' to identify enriched motifs; scrambleFasta.pl on DMR FASTA sequence to generate background sequences [approximately 50,000 scrambled sequences]).

**Fixed/reset DMRs.** DMRs predicted between any pairwise comparisons of wild populations (from Fig. 1g) were considered fixed when statistically found between the respective common-garden populations, consistently across all samples and with the same methylation direction (Wald test two-sided $P < 0.05$), or reset if no longer significant ($P ≥ 0.05$) or if showing change in methylation direction. The within-population methylome variation arising from the common-garden experiment itself was excluded from this analysis. Wild DMRs found among wild riverine, littoral and benthic fish were merged when found in >1 pairwise comparison using the BEDTools mergeBed function (v.2.27.1). Methylation levels at all wild DMRs were then plotted using pheatmap (Fig. 3a).

### RRBS

**NGS library preparation and analysis.** High-molecular-weight gDNA from liver tissue from 12 adult male fish per ecomorph (36 in total) was isolated using a modified version of the Wizard Genomic DNA Purification Kit (Promega Corporation). The quality and quantity of extracted DNA samples were assessed using Qubit and NanoDrop. Approximately 100 ng of liver high-molecular-weight gDNA were used to make RRBS libraries according to the manufacturer's instructions (Premium RRBS kit, catalogue no. C02030032; Diagenode). Each of the three RRBS sequencing libraries multiplexed 12 different samples, with ecomorph representation randomized among libraries. The quality and quantity of all libraries were assessed using TapeStation, Qubit and NanoDrop. RRBS libraries were sequenced on an Illumina NextSeq 500 system (to generate single-end 75 bp-long reads).

Due to poor read quality and low read counts, assessed using FastQC, one riverine ecomorph sample was excluded from further analysis. Analysis of spike-in controls gave a mean CpG bisulphite conversion efficiency across samples of 98.6%. Adaptor sequence in reads, low-quality end bases (Phred score ≤20), reads that were too short (<20 bp) and the first 5 bp (5′-end to avoid sequencing bias) were removed using TrimGalore (options: --rrbs --fastqc --clip_R1 5). In total, after quality trimming, there were 11.1 ± 3.4 Mio reads per RRBS sample (mean ± s.d.). Reads were then aligned to the same *M. zebra* reference genome (GCF_000238955.4_M_zebra_UMD2a_genomic.fa; see above) using Bismark (options: -N 1; ref. [48]). Mapping rates were 83.8 ± 0.8%, 81.6 ± 1.7% and 82.9 ± 1.6%, for benthic, river and littoral populations, respectively, similar to what has been reported in other RRBS studies[54]. Differences in mapping rates between the WGBS and

RRBS datasets stem from technical and sequencing differences, such as sequencing read length, single-/paired-end reads and genome coverage[48]. Methylation scores (read count supporting mC/total read count) at each CpG site genome-wide were extracted using Bismark's bismark_methylation_extractor (options: -s --multicore 4 --comprehensive --merge_non_CpG --bedGraph). PCA of methylation levels at CpG sites found across all samples (common CG sites, $n = 151,900$) was carried out using R 'prcomp' (centered and scaled). MANOVA followed by post-hoc Games–Howell multiple comparison tests using Tukey's correction were used to assess PC1 and PC2 score differences between populations using the R packages stats v.3.6.3 and rstatix.

**RRBS DMR prediction.** RRBS DMRs were identified using the same method as for WGBS DMRs (see above).

### Genomic annotation

**DMR localization.** Since no functional annotation exists for Lake Malawi cichlid genomes, promoter regions were defined in silico as regions ±1 kbp around the TSS. Gene bodies comprise exon and intron minus the first 1 kbp downstream of the TSS to avoid any overlap with promoter regions. Intergenic regions were defined as regions outside promoters and gene bodies for DMR localization. Only transposon repeats (TE) were analysed (excluding simple repeats, low complexity repeats, ribosomal RNA repeats and satellite repeats) and were annotated using RepeatMasker v.4.0.9 according to Vernaz et al.[9]. The annotation of CGIs was defined as in Vernaz et al.[9]. Overlaps between DMR coordinates and each respective genomic annotation were counted using the BEDTools intersectBed function (v.2.27.1).

### Enrichment for genomic features

Enrichment for methylome divergence (DMR) in different genomic features was performed by dividing the observed number of DMRs overlapping each genomic feature by the expected values (observed/expected ratio). The expected values were obtained by randomly shuffling the DMR coordinates genome-wide (1,000 iterations) for each genomic feature (BEDTools shuffleBed). One-sample $t$-tests were performed to test whether expected values were significantly different from the observed values. Chi-squared tests ($R$) were then performed for all observed/expected distributions among the three DMR comparison groups for each genomic feature.

### Assignment of DMRs to genes and GO

DMRs were assigned to genes when located in gene promoters (that is, TSS ± 1 kbp [promoter DMRs]; BEDTools intersect -f 0.5, ≥50% DMR sequence overlap required), in their gene bodies (excluding the first 1 kbp downstream TSS [gene DMRs]; BEDTools intersect -f 0.5, ≥50% DMR sequence overlap). When located outside promoter and gene bodies, intergenic DMRs were assigned to the closest gene if located 1–5 kbp away from it (closestBed; v.2.27.1). DMRs were associated with DEGs following the same method. An exact hypergeometric test (and representation factor) for the overlap between promoter DMRs and DEGs was performed. GO enrichment analysis using the genes associated with each DMR category was then performed using g:Profiler (https://biit.cs.ut.ee/gprofiler/gost; version March 2021; ref. [55]). Only annotated genes for *M. zebra* were used with a statistical cut-off of FDR < 0.05.

**Colocalization with HDRs.** The coordinates of HDRs from Malinsky et al.[16] were translated to the UMD2a *M. zebra* reference genome (GCF_000238955.4_M_zebra_UMD2a_genomic.fa) using the UCSC liftOver tool (namely, axtChain and liftOver; kent source v.418), based on a whole-genome alignment between the original by Brawand et al.[1] (https://www.ncbi.nlm.nih.gov/assembly/GCF_000238955.1) and the UMD2a *M. zebra* genome assemblies. The pairwise whole-genome alignment was generated using lastz v.1.02 (ref. [56]) with the following

parameters: "B = 2 C = 0 E = 150 H = 0 K = 4,500 L = 3,000 M = 254 O = 600 Q = human_chimp.v2.q T = 2 Y = 15,000". This was followed by using the USCS genome utilities (https://genome.ucsc.edu/util.html) axtChain tool with -minScore = 5,000. Additional tools with default parameters were then used after the UCSC whole-genome alignment paradigm (http://genomewiki.ucsc.edu/index.php/Whole_genome_alignment_howto) to obtain a contiguous pairwise alignment and the 'chain' file input for liftOver. All 98 HDRs mapped to the new assembly, although some HDRs were split into more than 1 region in the UMD2a assembly, resulting in 141 regions. The distances between DMRs between littoral and benthic populations and the closest HDR were inferred using the BEDTools closestBed function v.2.27.1 (ref. [53]).

### RRBS-WGBS cross-validation

To cross-compare the RRBS and WGBS datasets and validate the use of the whole-genome unbiased methylome sequencing technique, methylation variation at the WGBS DMRs using the RRBS methylome data ($n = 413$ DMRs in total) was analysed. In detail, methylome levels for all RRBS samples were averaged over all DMRs predicted using the WGBS samples (BEDTools intersect). Unbiased hierarchical clustering and the heatmap of the Spearman's correlation matrix using RRBS methylome variation at the WGBS DMRs were produced using pheatmap (Euclidean distances and complete-linkage clustering). The same clustering and heatmap approaches were used to plot methylation levels (averaged mCG/CG per population for both WGBS and RRBS samples) of RRBS samples at WGBS DMRs.

### Correlation of DNA methylation and transcription activity

To assess the overall correlation between DNA methylation and transcriptional activity, all annotated genes were split into 5 categories based on their gene expression levels: from genes not expressed (TPM < 5; 'OFF', $n = 24,598$) to expressed genes (4 'ON' categories; TPM ≥ 5), from lowest to highest gene expression activity ($n = 1,269$–$1,270$ genes for the ON categories) using the tidyverse v.1.3.1 function cut_number. For each gene expression category, the average methylome profile (average mCG/CG from 2 kbp upstream of the TSS to 2 kbp downstream of the transcription end site including the entire gene bodies were plotted using deepTools v.3.2.1. Spearman correlation tests were performed between transcriptional activity and methylation levels at gene bodies and promoters using cor.test ('stats' R package v.4.2.0). Benthic individuals were used for this analysis and are representative of the other populations ($n = 3$ biological replicates for liver WGBS [average mCG/CG levels] and $n = 5$ biological replicates for liver RNA-seq).

### Reporting summary

Further information on research design is available in the Nature Research Reporting Summary linked to this article.

## Data availability

The WGBS, RRBS and RNA-seq raw data have been deposited in the Gene Expression Omnibus under accession no. GSE174120.

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

## Acknowledgements

We thank the staff of the Tanzania Fisheries Research Institute for their assistance and support. The Tanzania Commission for Science and Technology provided research permits and the Tanzania Ministry of Livestock and Fisheries provided export permits. We thank M. Ndawala (TAFIRI) and G. Kazumbe for their help with sample collection, and J. Johnson for her fish paintings. We thank N. B. Ramakrishna for critical comments on the manuscript, the staff at the Gurdon Institute and the sequencing facilities at the Cancer Research UK (CRUK) Cambridge Institute, Wellcome Gurdon and Wellcome Sanger Institutes, and Genomics facility at the University of Bristol for their expertise and support. This work was supported by the following grants to E.A.M.: Wellcome Trust Senior Investigator award (nos. 104640/Z/14/Z and 219475/Z/19/Z) and CRUK award (no. C13474/A27826); to R.D.: Wellcome award (no. WT207492); to G.F.T.: Leverhulme Trust Award no. RPG-2014-214; to M.J.G., G.F.T. and B.P.N.: the Leverhulme Trust – Royal Society Africa Awards (nos. AA100023 and AA130107); to M.J.G. and G.F.T.: Natural Environment Research Council (NERC) award (no. NE/S001794/1); to M.J.G.: Leverhulme Trust award (no. RF-2014-686); to A.G.H.: Marie Skłodowska-Curie Individual Fellowship (no. GA-659791). G.V. acknowledges Wolfson College, University of Cambridge, and the Genetics Society, London, for financial support. M.E.S. is supported by an NERC Independent Research Fellowship no. NE/R01504X/1. The authors also acknowledge core funding to the Gurdon Institute from Wellcome (nos. 092096/Z/10/Z, 203144/Z/16/Z) and CRUK (no. C6946/A24843). For open access, the authors have applied a CC BY public copyright licence to any author-accepted manuscript version arising from this submission.

## Author contributions

G.V., A.G.H., E.S., B.P.N., N.P.G., M.C., A.H.S., M.E.S. and A.M.T. collected the samples. G.F.T. carried out the common-garden experiment. G.V. and B.F. extracted the RNA. A.G.H. and G.V. prepared the RRBS libraries and carried out the analysis. G.V. prepared the WGBS libraries. M.M. produced the HDR annotation. G.V. carried out the WGBS, RNA-seq and RRBS analyses. M.J.G., E.A.M., R.D., G.V. and M.E.S. supervised the study. M.J.G., G.F.T., E.A.M. and G.V. conceptualized the study. G.V., M.J.G. and E.A.M. wrote the manuscript with contribution from all the authors.

## Competing interests

The authors declare no competing interests.

## Additional information

**Extended data** is available for this paper at https://doi.org/10.1038/s41559-022-01894-w.

**Correspondence and requests for materials** should be addressed to Grégoire Vernaz, Martin J. Genner or Eric A. Miska.

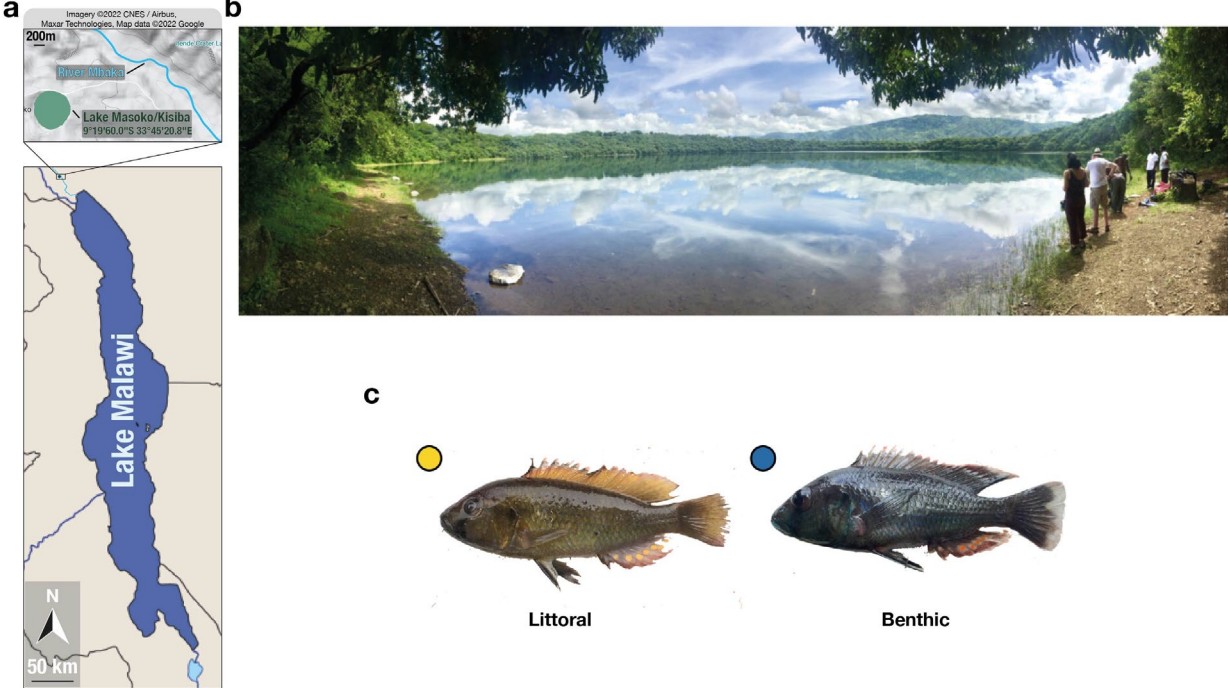

**Extended Data Fig. 1 | Map of Lake Masoko, Tanzania, and photographs of _Astatotilapia_ cichlid ecomorphs of Lake Masoko. a.** Lower panel: Lake Masoko/ Kisiba is part of the Lake Malawi catchment geographical area (location of Lake Masoko is indicated by the rectangle). Upper panel: Geographical map inset (rectangle) of Lake Masoko area. Credits: Imagery ©2022 CNES / Airbus, Maxar Technologies, Map data ©2022 Google (upper panel) and d-maps.com (lower panel). **b.** Panoramic photograph of Lake Masoko (11 April 2018). **c.** Photographs of male _A. calliptera_ specimens of Lake Masoko in breeding colors: littoral ecomorph (left), benthic ecomorph (right). Photo credits: GV.

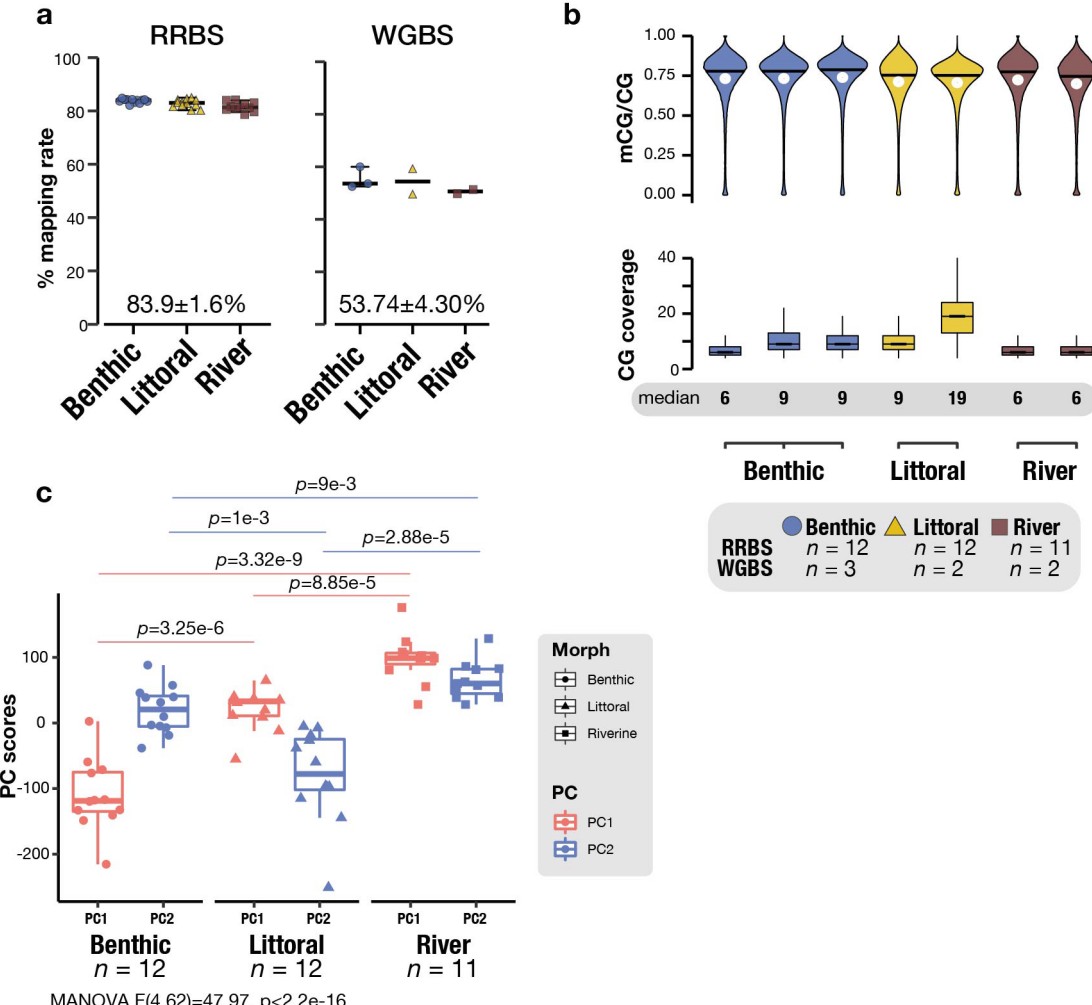

**Extended Data Fig. 2 | Mapping of sequencing reads and genome-wide methylome levels. a.** Mapping rates (percentage total reads) for RRBS and WGBS sequencing reads aligned to the SNP-corrected *Maylandia zebra* reference assembly (GCF_000238955.4 M_zebra_UMD2a). Black midlines and whiskers represent median values with 95% CI of mapping rates for each population. Overall mapping rates are shown at the bottom of each graph (mean ± sd). **b.** Upper panel: Genome-wide average liver methylation levels for each sample of each population (B, benthic, L, littoral and R, river; w, wild-caught). Average mCG/CG levels in non-overlapping 1kbp-long windows. Median and mean values are indicated with black midlines and white dots, respectively. Lower panel:

CG coverage (count of unique mapped reads) of all WGBS samples. **c.** Boxplots showing PC scores for PC1 and PC2 associated with the PC analysis (Fig.1a) of RRBS-related genome-wide CG methylation variation among all populations. Statistical report for MANOVA test is shown at the bottom of the graph and post-hoc Games-Howell multiple comparison two-sided *P*-values adjusted with the Tukey's method are shown for each pairwise comparison above each boxplot. $n \geq 2$ and $n \geq 11$ biological replicates for WGBS and RRBS datasets, respectively, for all graphs. All box plots indicate median (middle line), 25th, 75th percentile (box), and 5th and 95th percentile (whiskers) as well as outliers (single points).

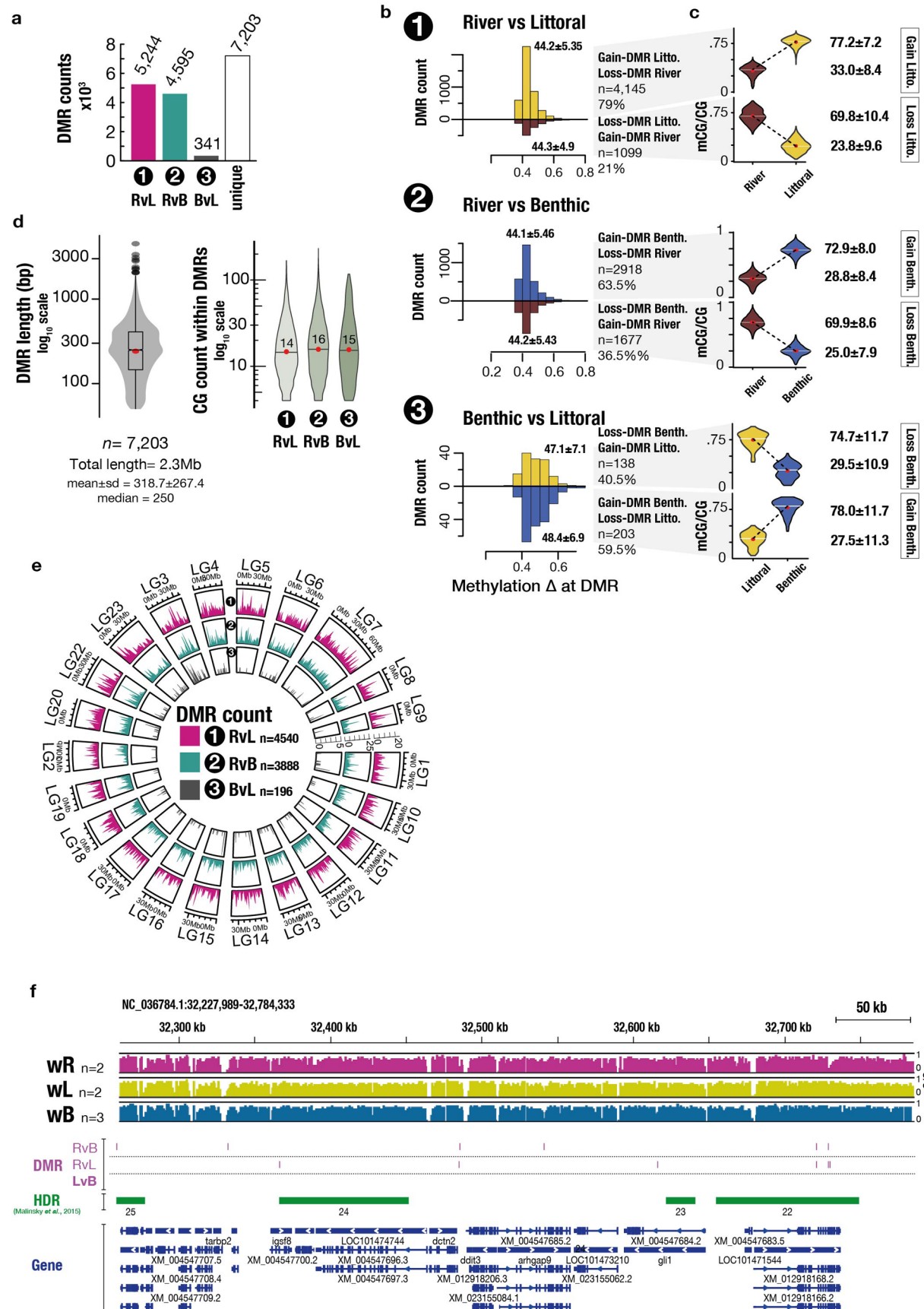

**Extended Data Fig. 3 | See next page for caption.**

**Extended Data Fig. 3 | Properties of the WGBS-DMRs found between wild populations of Lake Masoko and riverine *A. calliptera* ecomorph populations. a**. Total count of differentially methylated regions (DMR) found between each pairwise comparison using the WGBS dataset (B, benthic, L, littoral and R, river; v., versus; see Fig. 1g) and total number of unique DMRs (found in ≥1 pairwise comparison; in grey). **b**. Histograms of methylation difference (mCG/CG Δ) at DMRs found for each pairwise comparison. For each pairwise comparison, DMRs are split between Gain-/Loss-DMRs for gain/loss of methylation. Gain-DMRs in benthic, littoral or river fish, respectively, are indicated with histograms of different colors (blue, yellow, or red, respectively). Average DMR methylation difference (mean ± sd) is shown above/below each graph. **c**. Violin plots showing average DMR mCG/CG levels for each DMR group found in (**c**). Values on the left of each graph represent mean ± sd for mCG/CG levels. **d**. Left: Violin and box plots of the length (bp, log₁₀ y-axis) of all unique DMRs found in ≥1 comparison.

Right: violin plots showing the number of CG sites within all precited DMR sequences found between each pairwise comparison (log₁₀ y-axis). Red dots and black horizontal bars represent mean and median values, respectively. Box plots indicate median (middle line), 25th, 75th percentile (box), and 5th and 95th percentile (whiskers) as well as outliers (single points). **e**. Circos plot showing DMR density found between each pairwise comparison (from Fig. 1g) across all chromosomes (data shown only for linkage groups LG [putative chromosomes] 1-23 in GCF_000238955.4 M_zebra_UMD2a). **f**. Example of methylome landscape (mean mCG/CG over 50 bp-long windows) at four highly diverged regions (HDRs between littoral and benthic populations, in green; from Malinsky et al., 2015) for wild-caught (w) fish from river (R), littoral (L) and benthic populations (B). *n*, number of biological replicates. DMR, differentially methylated regions in the three pairwise comparisons (v, versus; in purple). HDR annotations refer to Ref.[16].

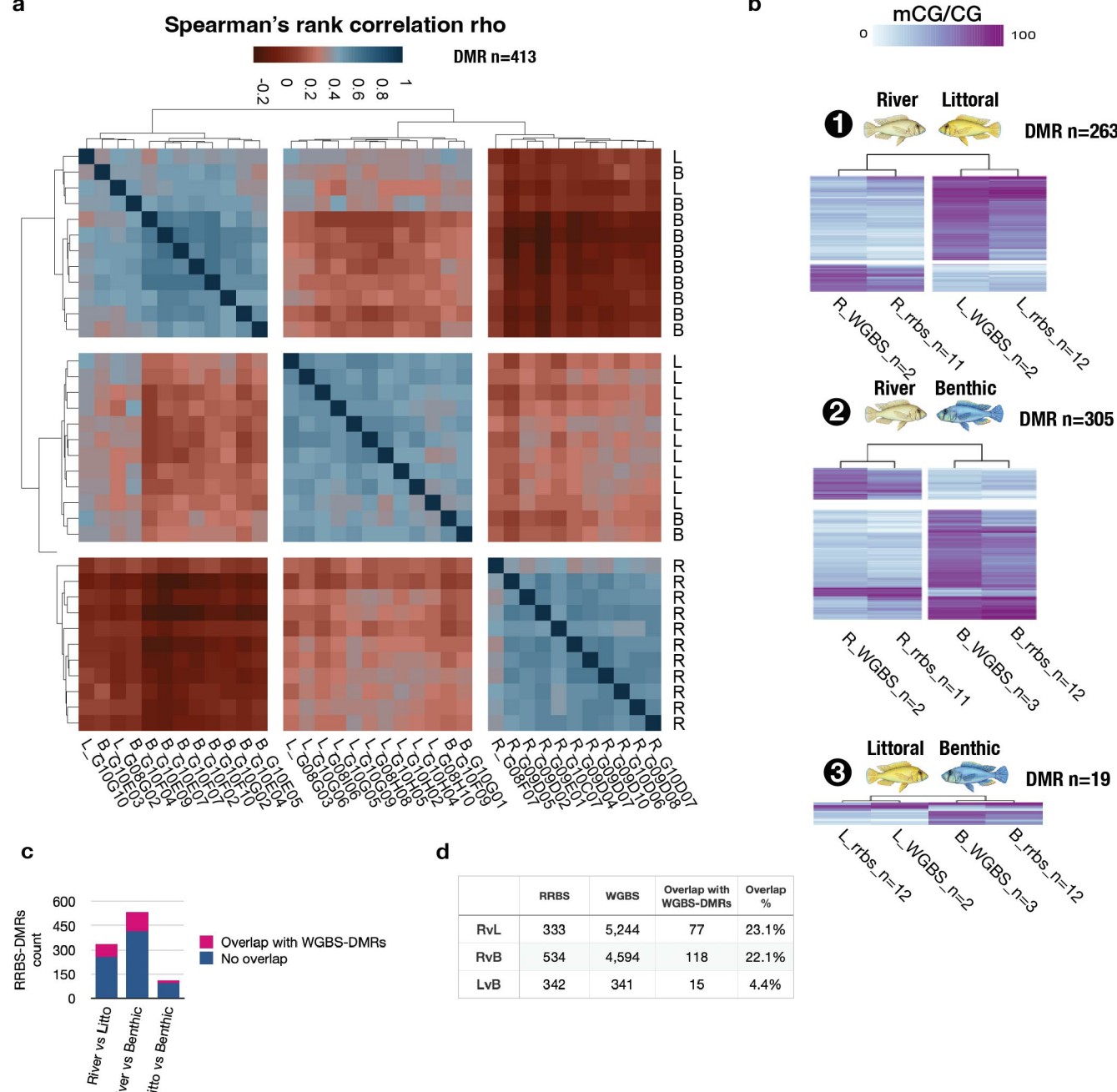

**Extended Data Fig. 4 | WGBS and RRBS datasets cross-validation. a**. Heatmap showing Spearman's correlation scores (Euclidean distances and complete-linkage clustering) of methylome variation at WGBS-DMRs covered by all RRBS samples (DMR count, *n* = 413) and showing strong population-specific methylome patterns (see Methods and Supplementary Notes). **b**. Heatmap showing average methylation levels averaged across all RRBS and WGBS samples (average mCG/CG per group) at WGBS-DMRs for each of the three WGBS-DMRs pairwise comparison (Fig.1g). Number of DMRs shown next to each heatmap. **c**. Total number of DMRs identified using the RRBS samples. The proportion of RRBS-DMRs overlapping with WGBS-DMRs is shown in pink. **d**. Table summarising the total number of DMRs found using both RRBS and WGBS datasets. Biological replicates: WGBS: Benthic (B), *n* = 3; Littoral (L) and Riverine (R), *n* = 2; RRBS, Benthic and Littoral, *n* = 12; River, *n* = 11.

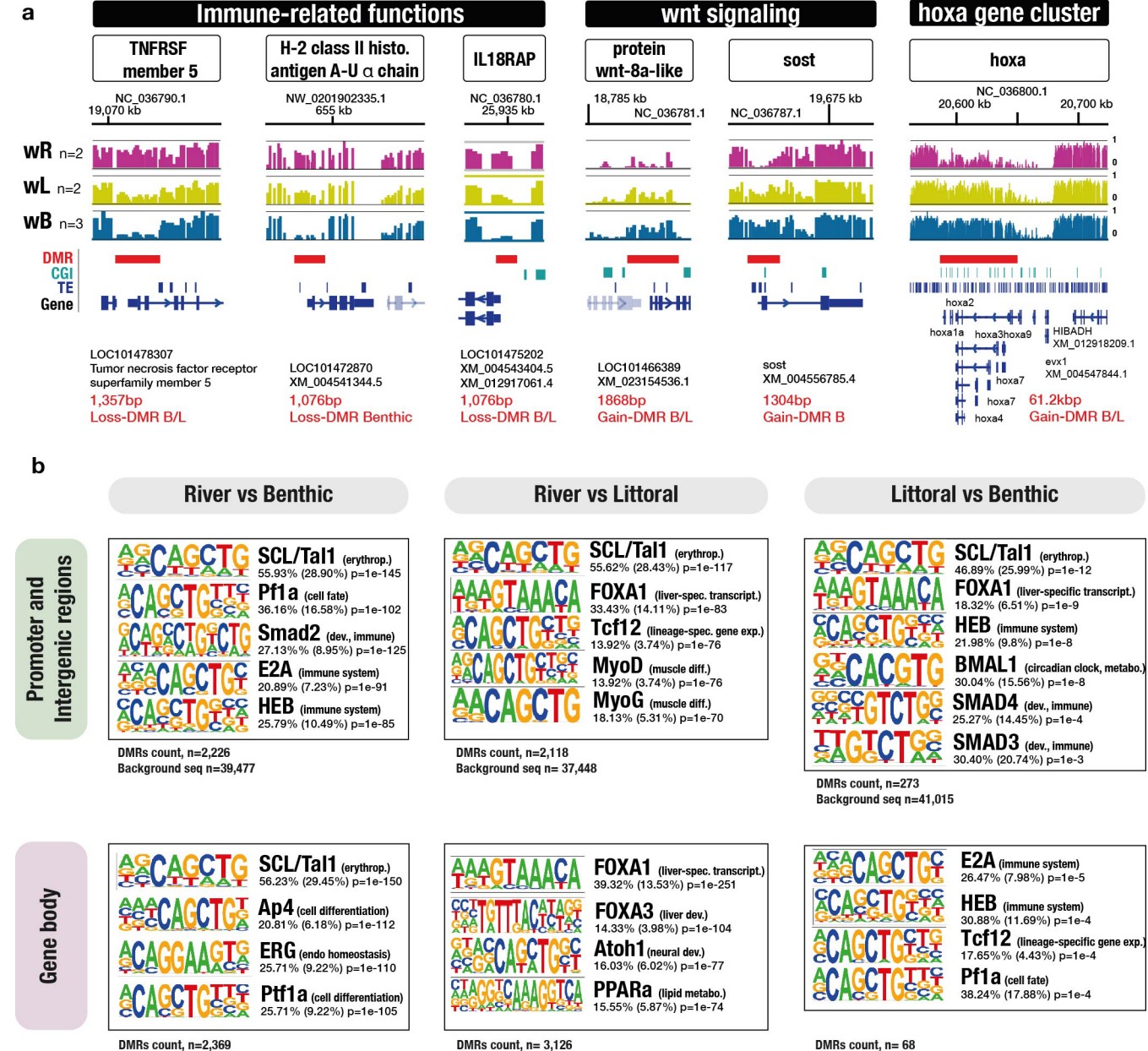

**Extended Data Fig. 5 | Examples of WGBS DMRs and enrichment analysis for TF sequence binding motifs within DMRs. a**. Methylome landscape (liver) for the three wild-caught (w) populations at six DMRs. Average mCG/CG [0-1] in 50 bp long windows (*n* indicates the number of biological replicates). The length (bp) of each DMR is indicated in red below each example. Loss-/Gain-DMRs indicate DMRs showing significant decrease/increase in mCG/CG levels (at least 25% mCG/CG difference, *P* < 0.05). **TNFRSF** member 5, tumor necrosis factor receptor superfamily member 5 (immune response, apoptosis); **H-2 class II histocompatibility antigen**, A-U alpha chain (adaptive immune response); **IL18RAP**, Interleukin-18 receptor accessory protein (immune response); **protein wnt-8a** and **sost**, sclerostin, both part of the wnt signalling pathway (early embryo embryogenesis). The homeobox **hoxa** genes cluster. **b**. Motif enrichment analysis for transcription factor (TF) binding sites in the sequence of DMRs located in promoter/intergenic regions (upper graphs) and within gene bodies (lower graphs) for the three pairwise comparisons. Enrichment over background using HOMER (scrambled DMR sequences, 50,000 iterations; see Methods). Two-sided *P*-values for motif enrichment based on cumulative hypergeometric distributions (tests) are shown.

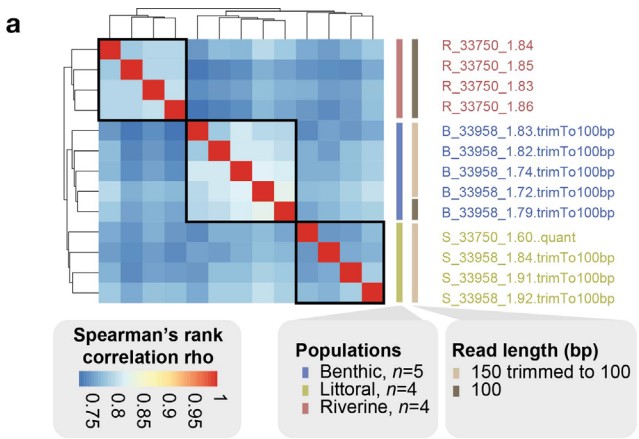

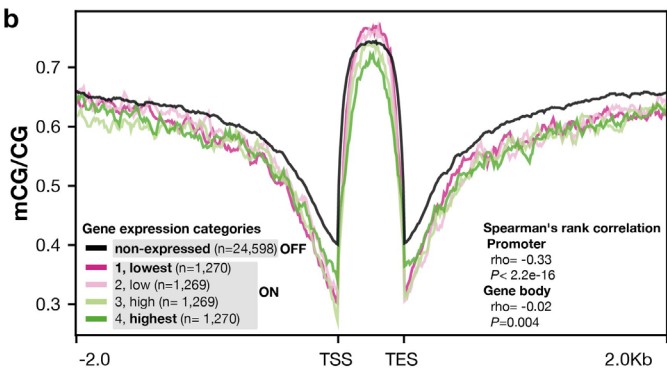

**Extended Data Fig. 6 | Transcriptional divergence among populations and negative correlation between promoter methylation and transcriptional activity. a**. Unbiased hierarchical clustering and heatmap of transcriptional variation among all RNAseq samples (based on Spearman's correlation scores). Gene expression patterns segregate the populations apart, independently of sequencing read lengths. All annotated genes were used (tpm). Of note, paired-end 150bp-long reads were trimmed *in silico* down to paired-end 100 bp to account for different read lengths (see Methods). **b**. Methylation profiles (averaged mCG/CG levels for *n* = 3 WGBS benthic samples) along gene bodies and at promoters according to gene expression levels. Genes were split into five categories according to their gene expression activity (averaged tpm per gene for *n* = 5 benthic fish RNAseq samples). Spearman's correlation tests are shown in plot and show significant negative correlation between methylation at promoters and transcriptional activity (rho = -0.33, two-sided *P* < 2.2E-16).

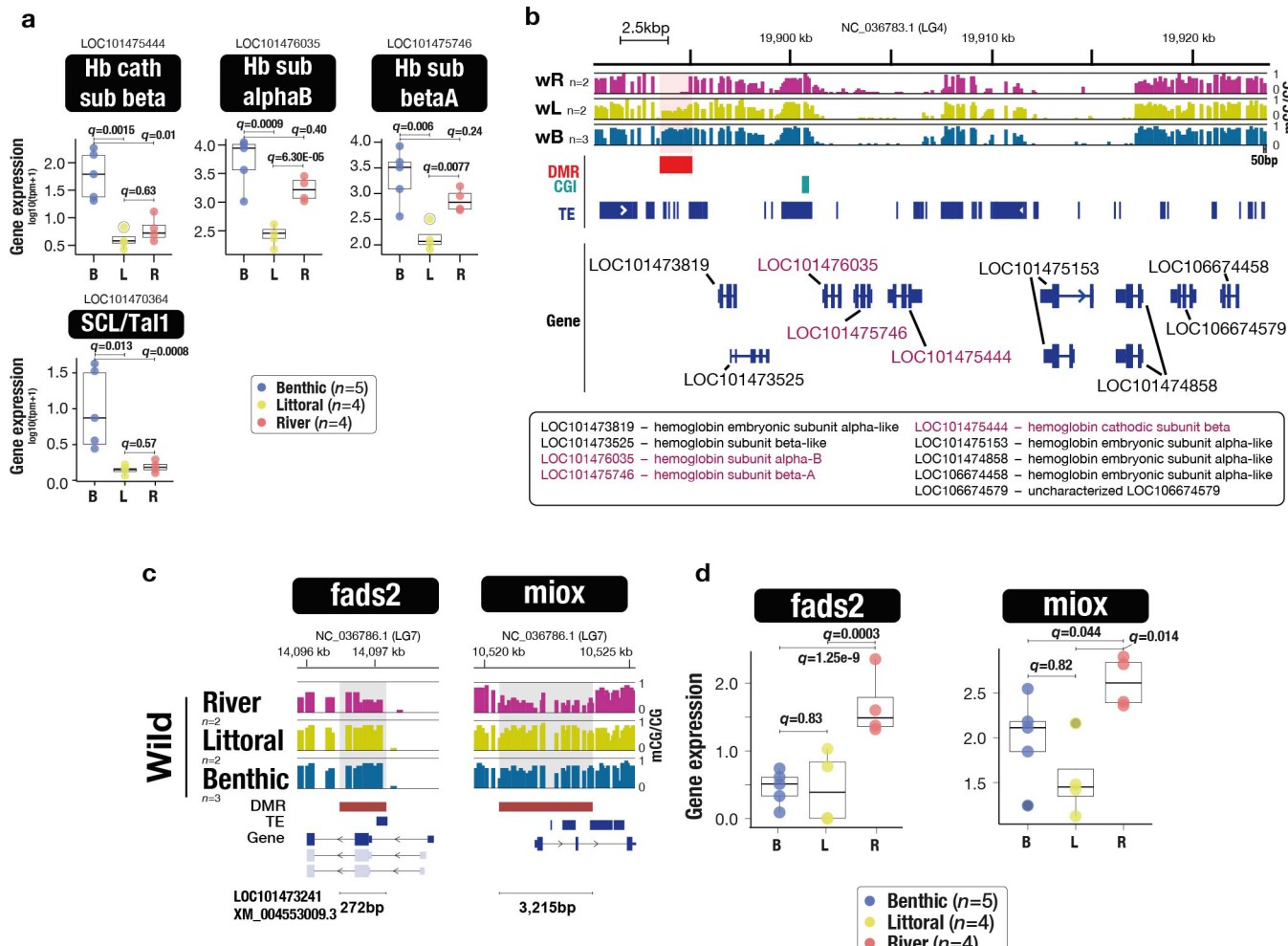

**Extended Data Fig. 7 | Genes with functions related to haematopoiesis, hemoglobin and lipid metabolism show higher transcriptional activity in Benthic fish specifically. a.** Boxplot of liver gene expression ($\log_{10}(\text{tpm} + 1)$) for the three hemoglobin (Hb) subunit genes and for SCL/Tal1 gene, all significantly upregulated in benthic fish. False discovery rate adjusted *P*-values (*q*) using the Benjamini-Hochberg method (sleuth; Wald test; see Methods) shown for Benthic-Littoral comparison only. **b.** The 2kbp-long DMR is located 7kbp away from the 'MN' globin cluster containing the three differentially expressed hemoglobin genes (see **a**.). Gain-DMR in the wild Benthic fish. Littoral and river fish show intermediate and low methylation levels respectively. w, wild-caught; R, L, B,

river, littoral and benthic fish, respectively. Each bar plot represents average mCG/CG levels for each population in 50bp-long windows. **c.** Methylation landscape over the genes *fads2* (gene body DMR) and *miox* (promoter DMR). Each bar represents the average liver mCG/CG levels in 50bp-long windows for each population. **d.** Boxplot of gene expression values ($\log_{10}(\text{tpm} + 1)$ for *fads2* and *miox* in Benthic (*n* = 5), Littoral (*n* = 4) and River (*n* = 4) liver tissues. *q*, false discovery rate adjusted two-sided *P*-value, using the Benjamini-Hochberg methods (sleuth; Wald test; see Methods) are shown for all comparisons in graphs. All box plots indicate median (middle line), 25th, 75th percentile (box), and 5th and 95th percentile (whiskers) as well as outliers (single points).

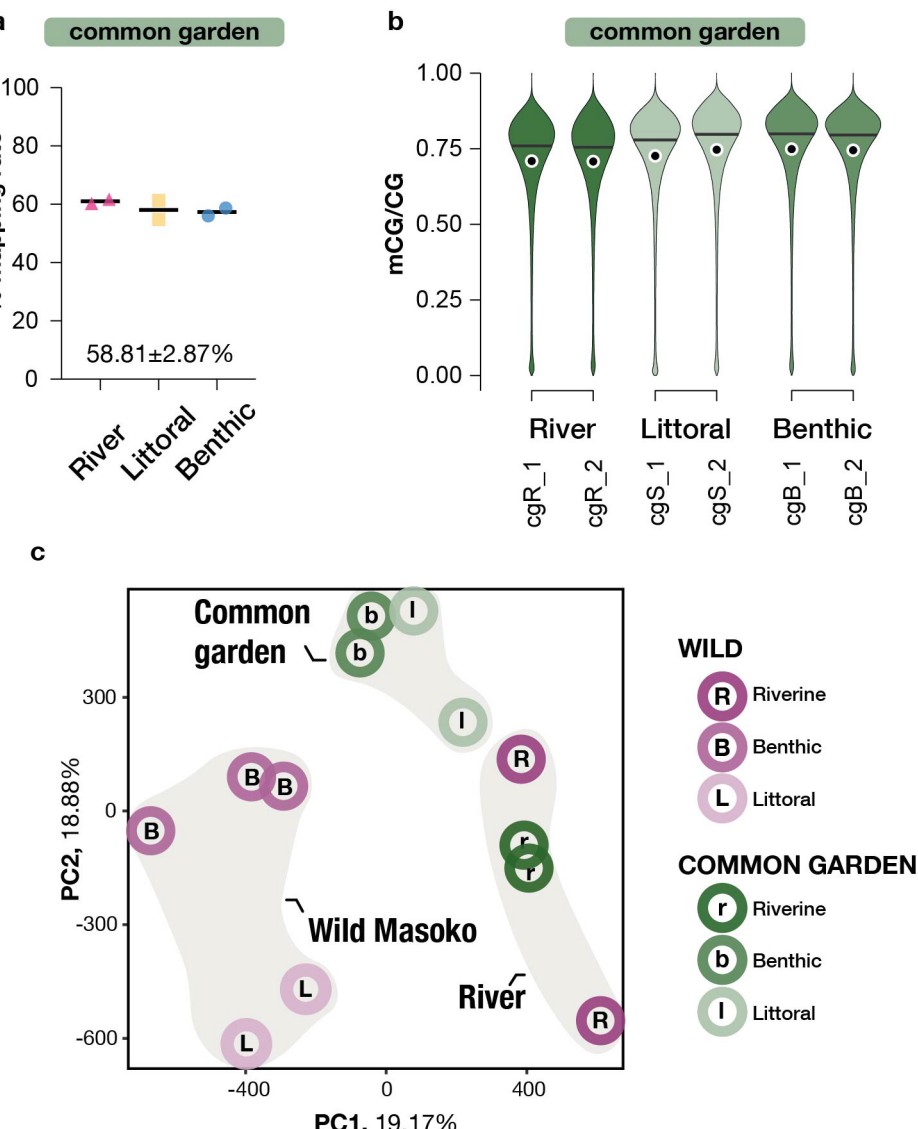

**Extended Data Fig. 8 | Mapping results, methylome variation and PC analysis for Common-Garden/tank-reared WGBS fish samples. a.** Mapping rates for Common Garden fish WGBS reads aligned to *M. zebra*_UMD2a reference genome (see Methods). Black midlines and whiskers represent mean ± sd of mapping rates within each population. Average mapping rates shown at the bottom of each graph (mean ± sd). **b.** Violin plots of average mCG/CG levels in 1kbp-long non-overlapping windows genome-wide for common garden (cg) fish of each population. Black circles and lines indicate mean and median values, respectively. **c.** PCA of liver methylome variation in wild (shades of purple) and common garden (shades of green) *A. calliptera* populations reveals global methylation remodeling in wild benthic and littoral fish (Wild Masoko) to resemble river fish methylome profiles. Upper- and lower-case letters distinguish wild from common garden fish for each population (river R/r, benthic B/b, littoral L/l). Biological replicates: wild fish: Benthic (B), $n = 3$; Littoral (L) and Riverine (R), $n = 2$; common garden fish: benthic (b), littoral (l) and riverine (r), $n = 2$.

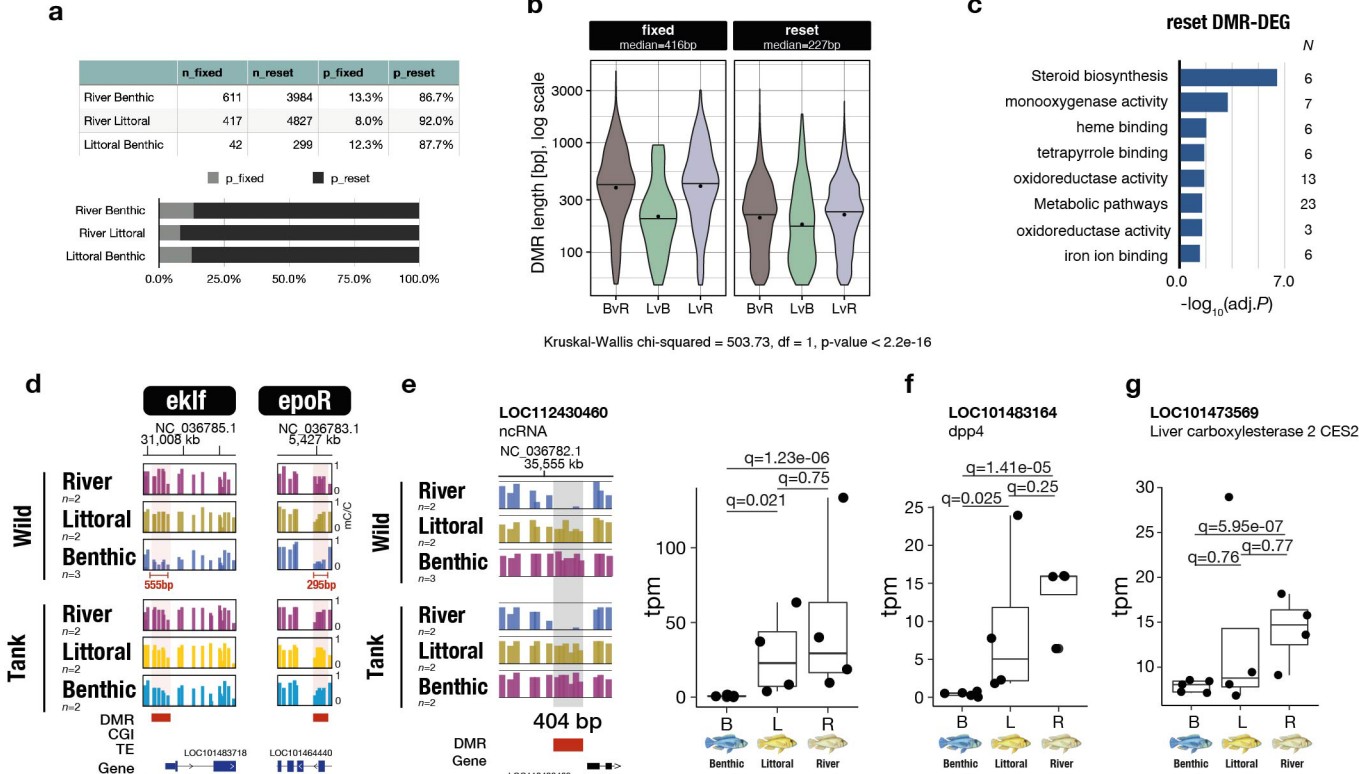

Kruskal-Wallis chi-squared = 503.73, df = 1, p-value < 2.2e-16

**Extended Data Fig. 9 | Fixed and reset methylome variation among wild populations of Lake Masoko _A. calliptera_ cichlids. a**. Plot showing the proportion of fixed versus reset DMRs between each pairwise comparison upon common garden experiment. **b**. Violin plots showing length in base pair (bp) of fixed and reset DMRs found for each pairwise comparison. y-axis, logarithmic scale. Median lengths for all fixed and reset DMRs are indicated above the graphs. Two-way ANOVA, _P_ < 4.6e-05. **c**. GO analysis for reset DMRs associated with differentially expressed genes (DEG). **d**. Methylome profiles of the genes _eklf_ and

_epoR_ for wild and common garden ('Tank') populations showing reset methylome patterns in benthic fish upon environmental perturbation. **e**. Example of fixed benthic-specific hypomethylation in an uncharacterised non-coding RNA gene associated with its downregulation in benthic fish only. **f-g**. Box plot showing gene expression profiles for the genes _dpp4_ and _ces2_, associated with fixed benthic-specific methylome patterns. B, benthic; L, littoral; R, river populations, respectively. Box plots indicate median (middle line), 25th, 75th percentile (box), and 5th and 95th percentile (whiskers) as well as outliers (single points).

| | |
|---|---|

# Reporting Summary

## Statistics

For all statistical analyses, confirm that the following items are present in the figure legend, table legend, main text, or Methods section.

| n/a | Confirmed | |
|---|---|---|
| ☐ | ☒ | The exact sample size (*n*) for each experimental group/condition, given as a discrete number and unit of measurement |
| ☐ | ☒ | A statement on whether measurements were taken from distinct samples or whether the same sample was measured repeatedly |
| ☐ | ☒ | The statistical test(s) used AND whether they are one- or two-sided<br>*Only common tests should be described solely by name; describe more complex techniques in the Methods section.* |
| ☐ | ☒ | A description of all covariates tested |
| ☐ | ☒ | A description of any assumptions or corrections, such as tests of normality and adjustment for multiple comparisons |
| ☐ | ☒ | A full description of the statistical parameters including central tendency (e.g. means) or other basic estimates (e.g. regression coefficient) AND variation (e.g. standard deviation) or associated estimates of uncertainty (e.g. confidence intervals) |
| ☐ | ☒ | For null hypothesis testing, the test statistic (e.g. *F*, *t*, *r*) with confidence intervals, effect sizes, degrees of freedom and *P* value noted<br>*Give P values as exact values whenever suitable.* |
| ☒ | ☐ | For Bayesian analysis, information on the choice of priors and Markov chain Monte Carlo settings |
| ☒ | ☐ | For hierarchical and complex designs, identification of the appropriate level for tests and full reporting of outcomes |
| ☐ | ☒ | Estimates of effect sizes (e.g. Cohen's *d*, Pearson's *r*), indicating how they were calculated |

*Our web collection on statistics for biologists contains articles on many of the points above.*

## Software and code

Policy information about availability of computer code

| | |
|---|---|
| Data collection | No software was used to collect data. |
| Data analysis | TrimGalore v.0.6.2, bismark v.0.20.0, kallisto v0.46.0, bedtools v2.27.1, R v.3.6.3 (R packages: DSS v.2.32.0, pheatmap v1.0.12, ggplot2 v.3.3.0, tidyverse v1.3.1, FSA v0.8.25, sleuth v0.30.0,deepTools v3.2.1, phangorn v2.5.5, circlize v0.4.12, rstatix v0.7.0), IGV v2.9.2, RepeatMasker v4.0.9.p2, makeCGI, g:Profiler (e100_eg47_p14_7733820), homer-4.9.1-6, bedGraphToBigWig (v4), lastz v1.02, UCSC liftOver tool (which included scripts: axtChain and liftOver; kent source version 418) |

For manuscripts utilizing custom algorithms or software that are central to the research but not yet described in published literature, software must be made available to editors and reviewers. We strongly encourage code deposition in a community repository (e.g. GitHub). See the Nature Portfolio guidelines for submitting code & software for further information.

## Data

Policy information about availability of data

All manuscripts must include a data availability statement. This statement should provide the following information, where applicable:

- Accession codes, unique identifiers, or web links for publicly available datasets
- A description of any restrictions on data availability
- For clinical datasets or third party data, please ensure that the statement adheres to our policy

WGBS, RRBS and RNAseq raw sequencing data have been deposited in the Gene Expression Omnibus (GEO) database under the accession number GSE174120 . Refer to Supplementary Tables 1-3 for sample IDs.

# Field-specific reporting

Please select the one below that is the best fit for your research. If you are not sure, read the appropriate sections before making your selection.

☐ Life sciences ☐ Behavioural & social sciences ☒ Ecological, evolutionary & environmental sciences

For a reference copy of the document with all sections, see nature.com/documents/nr-reporting-summary-flat.pdf

# Ecological, evolutionary & environmental sciences study design

All studies must disclose on these points even when the disclosure is negative.

| Study description | This study aimed at generating high-coverage whole-genome bisulphite sequencing and total RNA sequencing of liver tissues from wild-caught and tank-reared Lake Masoko and River Mbaka A.calliptera cichlid specimens to investigate epigenetic divergence and inheritance during early stages of speciation. |
|---|---|
| Research sample | In total, wild-caught A. calliptera specimens from Lake Masoko (both littoral/yellow and benthic/blue ecomorph populations) and from the neighboring river Mbaka (related to the ancestral population of Lake Masoko ecomorphs) were used in this study. In addition, tank-reared specimens from the three A.calliptera populations were used as well (first generation, bred from wild caught parental lines). Only adult males in full nuptial coloration were used in this study. DNA and RNA samples were extracted from liver tissues.<br><br>For RRBS experiments, 11-12 wild caught specimens for each of the three A. calliptera populations were used. For WGBS, 2-3 wild/tank-reared specimens were used for each of the three populations (wild WGBS samples were the same as for RRBS dataset). For RNAseq, 4-5 wild specimens were used for each population. Only males specimens were used. |
| Sampling strategy | The main strategy was to select 2-3/11-12/4-5 (WGBS/RRBS/RNAseq) adult, males (independent biological replicates) for each of the three A.calliptera populations (littoral and benthic populations from Lake Masoko and the riverine population of River Mbaka) in order to assess methylome and transcriptome divergence during early stages of speciation in this Crater Lake cichlid system. No statistical method was used to define sample size - sample sizes were based on literature. Within-species variation for all analyses and statistical procedures was taken into account. All fish were size-matched wild caught male specimens displaying full nuptial colorations (when males) and collected by collaborators. |
| Data collection | Fish were identified (photographs), registered in an excel sheet and dissected by AMT, GV, AGH, MC and MES. DNA extractions were performed by GV, AGH and MC. RNA extractions were performed by GV and BF. Illumina HiSeq sequencing was performed by the sequencing facility at CRUK/CI, Cambridge UK (WGBS), by the sequencing facility of the Wellcome Sanger Institute (RNAseq) and the sequencing facility of the University of Bristol (RRBS). All sequencing data was analyzed by GV. |
| Timing and spatial scale | Wild specimens were caught by professional divers in 2015, 2016, 2018 and 2019 in Tanzania in collaboration and compliance with the Tanzania Fisheries Research Institute (various collaborative projects). Upon collection, tissues were immediately dissected and placed in RNAlater (Sigma), and were then stored at -80°C upon return. Tank-reared specimens were culled at the same time according to the ethical and veterinary regulations in place at the University of Bangor, using MS222 method in 2017-2018. |
| Data exclusions | No data were excluded fro the analysis. |
| Reproducibility | All attempts to reproduce the experiments were successful. |
| Randomization | Species were selected based on their respective ecological niches and morphologies (using published literature). Randomization was not appropriate. |
| Blinding | All analyses related to genome-wide methylome and transcriptome differences (including hierarchical clustering, principal component analysis) were performed in an unbiased, blind approach (populations not known). For other analysis (DMR, DEG, isotope), investigators were aware of the population identity. |

Did the study involve field work? ☐ Yes ☒ No

# Reporting for specific materials, systems and methods

We require information from authors about some types of materials, experimental systems and methods used in many studies. Here, indicate whether each material, system or method listed is relevant to your study. If you are not sure if a list item applies to your research, read the appropriate section before selecting a response.

## Materials & experimental systems

| n/a | Involved in the study |
|-----|----------------------|
| ⊠ ☐ | Antibodies |
| ⊠ ☐ | Eukaryotic cell lines |
| ⊠ ☐ | Palaeontology and archaeology |
| ☐ ⊠ | Animals and other organisms |
| ⊠ ☐ | Human research participants |
| ⊠ ☐ | Clinical data |
| ⊠ ☐ | Dual use research of concern |

## Methods

| n/a | Involved in the study |
|-----|----------------------|
| ⊠ ☐ | ChIP-seq |
| ⊠ ☐ | Flow cytometry |
| ⊠ ☐ | MRI-based neuroimaging |

# Animals and other organisms

Policy information about studies involving animals; ARRIVE guidelines recommended for reporting animal research

| | |
|---|---|
| Laboratory animals | Tank-reared A. calliptera (river, benthic and littoral morphs), first generation (G1) were bred from wild specimens. Only adult males displaying full nuptial coloration were used for all experiments (two males per population). |
| Wild animals | All wild specimens of the three A.calliptera populations (benthic, littoral, river) were caught by professional SCUBA divers using fixed gill nets (at specific depths) in compliance permits issued to GF Turner, MJ Genner R Durbin, EA Miska by the Tanzania Fisheries Research Institute. |
| Field-collected samples | Wild specimens of A.calliptera (benthic, littoral and river) were imported to the UK by aquarium-trade import specialists. Specimens were cared for, reared and bred in the fish facilities of the University of Bangor in compliance with the ethical and veterinary regulations in place. Fish were reared under the same controlled laboratory conditions in separate tanks (light/dark 12/12 cycles, diet: algae flakes daily, 2-3times weekly frozen diet). Only adult males (2 per population) displaying full nuptial colouration were culled (MS222 anesthesia) for this study. |
| Ethics oversight | Sampling collection and shipping were approved by permits issued to GF Turner, MJ Genner R Durbin, EA Miska by the Tanzania Fisheries Research Institute. Tank-reared fish populations were maintained in compliance with the ethical and veterinary regulations in place at the University of Bangor. |

Note that full information on the approval of the study protocol must also be provided in the manuscript.

