## [Peer Review File · Nature Ecology & Evolution]

Peer Review Information

Journal: Nature Ecology & Evolution

Manuscript Title: Epigenetic Divergence during Early Stages of Speciation in an African Crater Lake Cichlid Fish

Corresponding author name(s): Grégoire Vernaz, Martin J. Genner, Eric A. Miska

Editorial Notes:

Reviewer Comments & Decisions:

Decision Letter, initial version:

21st September 2021

Dear Professor Miska,

Your Article entitled "Epigenetic Divergence during Early Stages of Speciation in an African Crater Lake Cichlid Fish" has now been seen by four reviewers, whose comments are attached. In the light of their advice, we have decided that we cannot offer to publish your manuscript in Nature Ecology & Evolution.

From the reports, you will see that while they find your work of potential interest, the reviewers raise concerns about the strength of the novel conclusions that can be drawn at this stage. We feel that these criticisms are sufficiently important as to preclude publication of your work in Nature Ecology & Evolution.

I am sorry that we cannot be more positive on this occasion, but hope that you find the reviewers' comments helpful when preparing your paper for resubmission elsewhere.

[REDACTED]

Reviewer expertise:

Reviewer #1: evolutionary ecology, epigenomics

Reviewer #2: evolutionary epigenomics

Reviewer #3: cichlids, genomics

Reviewer #4: speciation, genomics

Reviewers Comments:

Reviewer #1 (Remarks to the Author):

In this study, Vernaz et al. focused on the early stage of cichlid fish species radiation by analyzing DNA methylation and gene expression divergence between the ancestral riverine population and two derived lake populations. While the cichlid system is a well-studied system, results from this study add to the small body of studies focusing on the relationship between epigenetics and speciation, and provide some new insights into how an environmentally sensitive and rapid changing molecular mechanism contributes to the macroevolutionary event. That said, although the topic covered in this study is of immediate interest to the general community of ecology and evolution, I have both general comments on experimental design and specific comments on methods that prevent this manuscript to be published in its current shape. I would suggest the authors to thoroughly revise their MS before resubmission or submitting to a different journal.

General comments:

1. The inclusion of riverine population is a little bit weird. It makes reader confusing on the focus of this study: are you focusing on the speciation between riverine and lake populations, or are you focusing on the divergence between the benthic and littoral ecomorphs? I think the authors may need to make the focus of this paper more clear.
2. The combination of WGBS and RRBS is very confusing (also see my specific comments below). It seems that the data from RRBS were only used in one PCA, with its result quite different from the PCA from WGBS data.

3. Following my last comment, the methylation-gene expression correlation analysis also can be improved. The samples used in WGBS/RRBS were not the same samples or only a subset of samples used in RNA-seq. So, the discrepancy of samples used in methylation analysis vs. gene expression analysis can potentially bias the results.

4. The choice of male fish for this study needs more explanations. I can understand that methylation and gene expression are gender-specific, but to totally exclude females from analysis needs good justification. Is this because males show more phenotypic/physiological divergence between ectomorphs? Also, because it has been suggested that epigenetic patterns at early developmental stages can often reprogram to reflect the paternal state, identifying transgenerationally stable CpGs in male fish seems not unexpected but makes me wondering the magnitude of methylation stability in females.

5. A lot of test statistics are missing (e.g., lines 139-142), which needs to be improved.

Specific comments:

Lines 37-38: Do you refer to methylation or expression variation between genes involved in key biological processes? This sentence sounds like you are talking about two things, with transcriptional changes in 'ecologically-relevant genes' as the result of 'methylome divergence between populations'? If so, I would suggest adding a logical link between the methylome variation part and the transcriptional change part in this sentence.

Also, it seems some key information are missing in the Summary section, which are (1) a brief description of how the habitat differs between ecomorphs. It makes readers hard to understand how the genes and their related functions, e.g., steroid metabolism, erythropoiesis, are ecologically relevant to speciation without a context of environmental difference between habitats. (2) Definition of early stage speciation. Why are you interested in early stage but not other stages, and how do you know the ectomorphs were at the early stage of speciation?

Line 48: I agree that the role of epigenetic processes in speciation is less understood, but there are some recent studies focusing on epigenetic difference between ecotypes/populations in a wide range of taxa, for example, Darwin finches (Skinner et al. 2014. *Genome Biol. Evol.*), medaka (Ichiwaka et al. 2017. *Nat. Comm.*), cichlid fish (Kratochwil & Meyer. 2015. *Mol. Ecol.*; Franchini et al. 2021. *Mol. Biol. Evol.*), general teleost fish (Desvignes et al. 2021. *Mol. Biol. Evol.*), and suggesting DNA methylation, histone modifications, and microRNAs may contribute to speciation. In addition, other studies have demonstrated the gene expression mechanisms underlying cichlid fish divergence/speciation (e.g., Rajkov et al. 2020. *Mol. Ecol.*; El Taher et al. 2021. *Nat. Ecol. Evol.*). So, I would suggest the authors to cite at least the above papers to acknowledge the recent advances, and emphasize more on the knowledge gap of the role of epigenetic processes in early stage of speciation.

Lines 55-58: While *Fst* is one of the key indicators of divergence or speciation stage, more results such as gene flow may better support the statement of 'early stages of divergence'. In addition, the relatively low overall genetic divergence can be caused by relatively high gene flow between ectomorphs, so it may be more appropriate to report gene flow levels rather than *Fst*.

Lines 76-77: It is evident that the mapping rate is different between RRBS and WGBS. It would be good to have some explanations for this difference, and test for if such difference have an impact on downstream analysis. In addition, the choice of liver tissue needs to be justified.

fig. 1d: Maybe I am missing something, but I do not find any text related to fig. 1d. This panel suggests that there is a clear difference between food choice between littoral or river population vs. benthic population, but no difference between littoral and river populations. However, this panel is somewhat contradicted to the PCA results in fig. 1e-f, where the three populations are separated. In addition, data used for fig. 1d were generated from muscle tissue, which is different from the tissue used for RRBS and WGBS, so it is hard to cross-validate or combine results between different panel.

For fig. 1e-f: It would be good to provide explanations for why PC1 and PC2 explain different variation in the same tissue between RRBS and WGBS.

Lines 103-105: It is quite confusing why WGBS data were chosen for DMR analysis, considering only 2-3

samples were sequenced for each population. There is clearly a trade-off between the number of CpGs analyzed and the statistical power to detect true DMRs, and choosing WGBS based on very low sample size over RRBS based on good sample size needs to be justified.

Lines 112-118: Using genetic divergence result from a previous study is OK, but it would be better to directly extract SNPs and estimate genetic divergence from bisulfite sequencing data. Also, I do not agree that a distant distance between HDR and DMR suggests no association. It could be due to trans-acting genetic variants controlling methylation variation, so the statement between these lines are not well supported, especially considering the association between SNPs and DMRs were not specifically tested in this study.

Lines 119-130: There is a flaw in the genomic feature analysis. The authors did not give a precedence when features were overlapped, so fig. 1i could be significantly biased due to repeatedly counting DMRs overlapping with multiple features.

Lines 143-144: Implying adaptation directly from the genomic locations of DMRs is not suitable, especially without measuring fitness. I would suggest the authors to soften language here.

Fig. 2: In fig. 2a, One of the littoral fish is clustered with riverine fish, but there is no explanation for this. I would exclude this fish in downstream analysis. In addition, there are two different read lengths (100 vs. 150 bp PE) used in RNA-seq. Is there any batch effect due to read length? What is the correlation between samples from the same ecomorph but were sequenced in different read length? Finally, the color spectrum in fig. 2b is qualitative and not very informative, it would be good to show absolute or scaled FC instead of 'lowest'/highest'.

Lines 183-203: Do the genes, *epoR*, *chbB*, *Sci/Tal1*, etc., have hypomethylated promoters? Because the authors emphasized the overrepresentation of DMRs in DEGs, I guess the examples in this paragraph are genes with decreased methylation in promoters but increased transcription activities, but I do not see any clear statement about such information. Also, it would be good to report or depict the correlation between methylation levels and transcription activities, either using genome-wide data or DMRs-DEG pairs.

Line 235: I do not understand the process to call 'reset DMRs' and 'fixed DMRs'. I believe some DMRs identified between riverine and benthic/littoral populations will have similar methylation level difference in common garden environment, which can be called as 'fixed', but how about methylated regions with decreased methylation difference between, say, benthic and riverine populations? Do you also consider them as 'fixed DMRs'? Finally, it seems that methylation levels also changed between wild and common garden environments within the same population, so even for some DMRs you called 'fixed' between populations, they can be 'reset' or 'plastic' within populations.

Extended Fig. 8: following my last comment, in wild samples, the variation among riverine samples is higher than littoral or benthic samples, but lower in common garden environments than littoral or benthic samples, suggesting a large number methylated CpGs were plastic or responsive to environmental transition, especially in riverine populations, which adds more complexity in defining reset and fixed DMRs. One possible way to identify transgenerationally stable CpGs is to compare methylation levels of F1 and F2 samples under common garden environment, similar to Heckwolf et al. 2020. *Sci. Adv.* and Hu et al. 2021. *Genetics*. And to identify reset/plastic and fixed DMRs, it should be better to reciprocally transplant wild samples to opposite environments, e.g., transplant riverine samples to benthic or littoral environment, and identify CpG sites with altered methylation levels that are explained by ecomorph (fixed) or environments (reset/plastic).

Line 238: What is the biological meaning or explanations for reset DMRs having longer length than fixed DMRs? Are you suggesting longer length correlates with higher epimutation rate?

Line 253: While I agree that, to some extent, the fixed DMRs represent population-specific methylation patterns, because only liver tissue was used in methylation analysis, it is hard to say such patterns are also tissue-independent.

Line 355: Why sequencing WGBS samples in two different platforms? How did you incorporate and correct data from these two platforms?

Line 369: CpGs with only at least 4 reads to be included in downstream analysis seem less favorable due to low coverage. Can you provide justifications for such low depth cut-off? Also, the coverage for genome needs to be provided for WGBS.

Reviewer #2 (Remarks to the Author):

In the current study Vernaz et al present compelling evidence that DNA methylation might act as a major driver of vertebrate speciation. The study is a timely follow up of authors' recent work, which put forward the exciting hypothesis that DNA methylation participated in the phenotypic diversification of lake Malawi cichlid fish. The study is well-written, the analyses are properly executed and overall, my comments for improvements are minor.

- 1) It is not clear from the current data how DMRs were associated to genes for the purpose of GO analyses. Did the authors employ the "nearest gene" approach or was more elaborate methodology used for this (such as GREAT for example)?
- 2) How many DNMTs / TETs were identified in cichlids? Given the enrichment of DMRs in CpG-rich regions, it would be interesting to explore if these differences could be caused by expression differences of DNA methylation machinery components?
- 3) Also, TET proteins which actively remove DNA methylation are oxygenases and thus require oxygen for their catalytic activity. Could the difference in oxygen concentration therefore act as a driver for such epigenetic diversification? i.e less, oxygen  less TET activity  less capacity for 5mC removal  more hypermethylated regions?
- 4) Instead of speculating about the possibility of fixation of epigenetic differences, could the authors perform RRBS / WGBS on sperm (or eggs, but more likely sperm given the current literature) on fish from their "garden experiment"? Given the strong enrichment for CpG-rich regions even RRBS might do. Such an experiment would demonstrate the potential for inter- or trans-generational inheritance of acquired methylation differences. Even if only a small subset of DMRs is picked up in sperm, this would be an extremely valuable piece of data.
- 5) Reference 24 is wrong – the authors probably refer to Skvortsova et al, 2019 Nature Comms, rather than Skvortsova et al 2018 (Nat Rev Mol Cell Biol), where the inheritance of paternal methylome patterns in the zebrafish germline is not discussed.

Reviewer #3 (Remarks to the Author):

NatEcolEvo-210814388-T

This work by Vernaz and colleagues assess variation in DNA methylation in benthic versus littoral populations of *A. calliptera* cichlids from river and Lake populations. The role of such epigenetic variation in an ongoing sympatric speciation event like this is largely unexplored. This work assesses overall patterns of methylation, potential relationship to gene expression, and plastic potential in a common garden experiment. While a strong work, there are a number of instances of missing logic of analyses or weak explanations that would strengthen conclusions. These are needed to bring this work from statistical connections in large-scale genomic analyses to the biological relevance and importance for evolutionary processes as pitched.

Major comments:

A. Biological relevance of methylation differences

A1. The suggestion in the paper is that differences in metabolism due to methylation are driving benthic versus littoral adaptation. However, samples were taken from these 2 environments that are very different in terms of oxygen, etc, which will cause metabolism differences. Is it that methylation differences drive this habitat divergence or that methylation differences in the liver become distinct once animals diverge in habitat usage?

A2. L63 notes that system is addressing sympatric speciation. However, the populations used do not live in sympatry, but are already distinct in their habitat usage.

A3. The connection between differential methylation patterns and transcription factor binding was not clearly defined. Is the suggestion that differential methylation is happening in a way that only impacts certain TFs? How would this work biologically? Or is it that those genes active in the liver are enriched for those TF binding sites in their promoters? See also point C5.

A4. I love the common garden experiment as it begins to address considerations such as that in point A1. However, details of this seem unclear. Were the conditions in this experiment similar to any of the natural environments? For example, were the tanks similar in oxygen and temperature of the littoral, but not benthic fish? How would this bias analysis or alter conclusions? How long was the experiment conducted, what is the logic of that choice, what is the developmental age of the fish at start and end of the experiment, and how does that affect the data obtained? Again, is it that you are catching all the regions of the genome that are responsive to environmental changes or those that are driving habitat changes and speciation? Some of the analysis in this section also highlights limitations of drawing conclusions from large-scale enrichment analyses. For example, GO analysis shows enrichment for fixed DMRs in developmental genes, but genes associated with neuron projection, neural tube development, and mastoid bone, none of which make sense given the tissue analyzed was liver. While interesting, the section about effect of fixed DMGs and brain development in L250-261 thus reads as examples picked from a list rather than making a functional connection between liver activity and brain development.

B. Choice of samples

B1. There was no explanation of why liver tissue was used for this analysis. What biases and limitations of conclusions does this introduce into the analysis?

B2. Also not discussed is why males only were used. What is the logic of choosing males with bright nuptial coloring, and could this add any bias to your analysis or our understanding of evolutionary processes?

C. Analyses

C1. PCA of data sets is stated as having segregation in L78-79, but this is not supported with statistical analysis such as an ANOVA. Does hierarchical clustering of all regions, not just differentially with your specific cutoffs, also show segregation of populations?

C2. What is the logic of cutoffs used for analysis? For example, differentially methylated regions were called based on >25% mCG difference and >4 CpG sites. What difference would using 5 sites make in the conclusions? Why 4 sites? This is stated as "significantly" different in L93 and a p-value is given, but there is not information on how this pvalue is calculated in methods L368-370.

C3. Assigning non-coding regions to gene function is a critical assumption that can bias downstream assessment of biological processes. What is the quality of the annotation of genes in the genome used for this (a critical factor for these analyses)? Why did you chose 1kb for a promoter versus 5kb? How were genes assigned to a specific gene for GO analysis?

C4. Analysis for transcription factor binding motifs appears to only be conducted on DMRs in one particular region (outside the gene body). How does this compare to the groupings presented in Figure 1I? How do conclusions change when different genomic features are used? Is there a reason only 2 of 3 combinations are shown (river vs littoral is omitted in extended data Figure 3F)?

C5. What is the directionality of relationships between DMRs and DEGs? Given that DNA methylation has a predictable effect on gene expression, are those regions with high methylation associated with low gene expression? This is explored for *eklf* and *epoR* in L183-203. However, the *chbB* locus is exclusively expressed in benthic fish that hypermethylation, and not expressed in fish populations with low to no methylation. Thus, further explanation is needed to clarify clear how the biological function of methylation relates to gene expression. Without it, the text in L166-213 reads more like anything differentially expressed in the methylation data and transcriptional data were simply compared for overlap, regardless if the biology lines up.

Minor comments:

D. Abbreviations. I encourage authors to critically evaluate if all abbreviations are necessary. Three specific examples, though I encourage a careful read for this throughout: (1) HDR and DMR get confusing for readers and are not always clearly defined in figure legends. (2) oCGI, pCGI, and LCR abbreviations are only used once, so there does not seem to be a utility to adding more abbreviations here. (3) GO categories MF, BP, CC, and KEGG in Figure 3 are never defined in the legend or the text.

Reviewer #4 (Remarks to the Author):

In this paper, the authors have investigated divergence in DNA methylation pattern of the liver between littoral and benthic ecomorphs of an African crater lake cichlid fish. They found that these two ecomorphs diverge in DNA methylation patterns in several genes related to ecological adaptation and also report that some of the variations may be associated with gene expression variation. They also claim that the majority of the variations are reset during a common garden experiment, but some epigenetic variations are genetically determined.

I agree that the topics of this study, namely the role of DNA methylation in incipient speciation, is one of the important topics in evolutionary biology. However, I found several weaknesses in the paper.

First, they used only 2-3 wild-caught fish for the the whole genome bisulfite sequencing (WGBS). Although they used more individuals for reduced representation bisulfite sequencing (RRBS), the majority of the analysis are based on WGBS. I wonder whether they can validate this small sample size by comparing the results between WGBS and RRBS. Some of the RRBS sites should be included in WGBS data. I suggest that they should test how consistent the results of WGBS with N=2-3 and RRBS with N=11-12 are in such overlapping sites.

Second, I found no evidence for supporting that the DNA methylation differences they found are adaptive. I agree that DNA methylation differences in physiologically important genes are consistent with their claim, but they are not evidence for the adaptive roles of DNA methylation. Do the differentially methylated sites show any signatures of divergent selection?

Third, they found that some of the DNA methylation differences detected in the wild-caught fish were reset in the lab-raised fish. However, I am afraid that this may simply result from the chance effect due to the small sample size of the wild fish (i.e., morph-specific DNA methylation in the wild fish may be an artifact).

Overall, I think that this paper tries to address an important topic, but the conclusions on the adaptive significance and the inheritance of DNA methylation are not very solid.

Additionally, given the fact that there are already several papers reporting DNA methylation differences between young species or ecomorphs in darter fishes (eg. Smith et al. 2016 Mol Ecol) and stickleback fishes (Hu et al 2021 Genetics), I have difficulty in finding novelty of the work.

Minor comments

How did you treat the sex? Is there any difference in sex difference in DNA methylation and gene expression? Sometimes, the sex explains the largest amount of gene expression variation.

L181-182: Is there any correlation between the direction of DNA methylation and gene expression (i.e., high methylation is associated with low expression etc)?

L408: Is it common to exclude the 1 kb downstream of TSS for defining the gene body?

**Although we cannot publish your paper, it may be appropriate for another journal in the Nature Portfolio. If you wish to explore the journals and transfer your manuscript please use our <https://mts-natecolevol.nature.com/cgi-bin/main.plex?el=A4Cn3FeF3A1EmB4X7A9ftdjXfwlz35q4KkId3135sAZ>>manuscript transfer portal. If you transfer to Nature journals or the Communications journals, you will not have to re-supply manuscript metadata and files. This link can only be used once and remains active until used.

All Nature Portfolio journals are editorially independent, and the decision on your manuscript will be taken by their editors. For more information, please see our http://www.nature.com/authors/author_resources/transfer_manuscripts.html?WT.mc_id=EMI_NPG_1511_AUTHORTRANSF&WT.ec_id=AUTHOR>manuscript transfer FAQ page.

Note that any decision to opt in to In Review at the original journal is not sent to the receiving journal on transfer. You can opt in to *[In Review](https://www.nature.com/nature-research/for-authors/in-review)* at receiving journals that support this service by choosing to modify your manuscript on transfer. In Review is available for primary research manuscript types only.

** For Nature Research general information and news for authors, see <http://npg.nature.com/authors>.

Author Rebuttal to Initial comments

Appeal: Response to Reviewers Comments

Reviewer #1 (Remarks to the Author):	Our response
In this study, Vernaz et al. focused on the early stage of cichlid fish species radiation by analyzing DNA methylation and gene expression divergence between the ancestral riverine population and two derived lake populations. While the cichlid system is a well-studied system, results from this study add to the small body of studies focusing on the relationship between epigenetics and speciation, and provide some new insights into how an environmentally sensitive and rapid changing molecular mechanism contributes to the macroevolutionary event. That said, although the topic covered in this study is of immediate interest to the general community of ecology and evolution, I have both general comments on experimental design and specific comments on methods that prevent this manuscript to be published in its current shape. I would suggest the authors to thoroughly revise their MS before resubmission or submitting to a different journal.	We thank the reviewer for highlighting the timely importance and value of our work to ecology and evolution research community.
1. The inclusion of riverine population is a little bit weird. It makes reader confusing on the focus of this study: are you focusing on the speciation between riverine and lake populations, or are you focusing on the divergence	Two major evolutionary steps characterise the Lake Masoko cichlid radiation: first, the colonisation of the lake by an ancestral riverine population, forming the littoral population. Second, the colonisation of the

between the benthic and littoral ecomorphs? I think the authors may need to make the focus of this paper more clear.	benthic habitat by the shallow population, forming the benthic population. It is therefore highly relevant and important to include the ancestral riverine population (phenotype and genotype) to fully study this instance of speciation. We will revise the text to clarify the importance of studying the ancestral riverine population.
2. The combination of WGBS and RRBS is very confusing (also see my specific comments below). It seems that the data from RRBS were only used in one PCA, with its result quite different from the PCA from WGBS data.	RRBS enriches for certain CG-rich loci (limited overall genome coverage) but enables affordable population-scale methylome analysis of larger sample size. WGBS enables us to characterise and validate the methylome landscape genome-wide. In this case, both RRBS and WGBS datasets strongly indicate population-specific methylome divergence, reaching the same conclusion and adding support to our core hypothesis of functional epigenetic diversity during the speciation. We will revise the text to clarify the rationale using these two complementary approaches. We will also undertake further cross-validation analyses of the RRBS and WGBS datasets (using DMRs) and quantify biologically relevant overlap.
3. Following my last comment, the methylation-gene expression correlation analysis also can be improved. The samples used in WGBS/RRBS were not the same samples or only a subset of samples used in RNA-seq. So, the discrepancy of samples used in methylation analysis vs. gene expression analysis can potentially bias the results.	Like any biological sampling of individuals in a population, we assume that each sample analysed here is representative of a broader population. Our analytical approach does not depend on the same samples being used. We will clarify this in the text.
4. The choice of male fish for this study needs more explanations. I can understand that methylation and gene expression are gender-specific, but to totally exclude females from analysis needs good justification. Is this because males show more phenotypic/physiological divergence between ecomorphs? Also, because it has been suggested that epigenetic patterns at early developmental stages can often reprogram to reflect the paternal state, identifying transgenerationally stable CpGs in male fish seems not unexpected but makes me wondering the magnitude of methylation stability in females.	There are three reasons we focus on males:  1) Practically, we are more confident in our assignment to male individuals to ecomorphs. 2) In our sampling, to test our hypotheses, we aimed to maximise between-population variability, while minimising within-population variability, within the constraints of resources available. 3) Epigenetic inheritance / reprogramming varies substantially among species (Potok et al. 2013; Skvortsova et al. 2019; Wang and Bhandari 2019), yet the nature of this in cichlid species was hitherto unknown. We surmised that if epigenetic inheritance is relevant to speciation, then it must be contributing to between-morph differences in traits possessed by both sexes (habitat occupancy, diet). Since we see differences in males, then the same may well apply to females, and this is a clear hypothesis that can be tackled with future work. We will clarify the rationale for our choice of samples to analyses in the revised text and highlight the limitations.

5. A lot of test statistics are missing (e.g., lines 139-142), which needs to be improved.	Some full test statistic reports are missing, and we will ensure complete reporting in the revised manuscript.
Specific comments: Lines 37-38: Do you refer to methylation or expression variation between genes involved in key biological processes? This sentence sounds like you are talking about two things, with transcriptional changes in 'ecologically-relevant genes' as the result of 'methylome divergence between populations'? If so, I would suggest adding a logical link between the methylome variation part and the transcriptional change part in this sentence.	This sentence will be reformulated accordingly to better clarify the link between methylation and transcription.
Also, it seems some key information are missing in the Summary section, which are (1) a brief description of how the habitat differs between ecomorphs. It makes readers hard to understand how the genes and their related functions, e.g., steroid metabolism, erythropoiesis, are ecologically relevant to speciation without a context of environmental difference between habitats. (2) Definition of early stage speciation. Why are you interested in early stage but not other stages, and how do you know the ecomorphs were at the early stage of speciation?	We will amend the abstract and introduction to more comprehensively describe key (published) background information related to ecomorph habitats and the evidence that the ecomorphs are at the early stages of speciation. We are interested in these early stages as this is the most challenging phase to study. It is easy to identify long-diverged species pairs, but identifying ongoing divergence where we know the precise ecological and historical context of the system is extremely rare. We have "caught" these populations undergoing speciation, and that is what excites us (and many other researchers who cite our work).
Line 48: I agree that the role of epigenetic processes in speciation is less understood, but there are some recent studies focusing on epigenetic difference between ecotypes/populations in a wide range of taxa, for example, Darwin finches (Skinner et al. 2014. Genome Biol. Evol.), medaka (Ichiwaka et al. 2017. Nat. Comm.), cichlid fish (Kratochwil & Meyer. 2015. Mol. Ecol.; Franchini et al. 2021. Mol. Biol. Evol.), general teleost fish (Desvignes et al. 2021. Mol. Biol. Evol.), and suggesting DNA methylation, histone modifications, and microRNAs may contribute to speciation. In addition, other studies have demonstrated the gene expression mechanisms underlying cichlid fish divergence/speciation (e.g., Rajkov et al. 2020. Mol. Ecol.; El Taher et al. 2021. Nat. Ecol. Evol.). So, I would suggest the authors to cite at least the above papers to acknowledge the recent advances, and emphasize more on the knowledge gap of the role of epigenetic processes in early stage of speciation.	We will cite those references.
Lines 55-58: While F_{ST} is one of the key indicators of divergence or speciation stage, more results such as gene flow may better support the statement of 'early stages of divergence'. In addition, the relatively low overall genetic divergence can be caused by relatively	Our evidence of the nature and timescale of divergence does not solely rely on F_{ST}. Key evidence is demographic modelling by Malinsky et al. (2015), which shows ecomorph divergence 500-1000 years ago (200-350 generations). Moreover, more recent work has

high gene flow between ectomorphs, so it may be more appropriate to report gene flow levels rather than Fst.	revealed the presence of 'hybrid' individuals of varying levels of admixture, demonstrating incomplete reproductive isolation (Munby et al. 2021, preprint). We will describe this evidence consistent with the early stage of speciation in a revision.
Lines 76-77: It is evident that the mapping rate is different between RRBS and WGBS. It would be good to have some explanations for this difference, and test for if such difference have an impact on downstream analysis.	Differences in mapping rates between RRBS vs WGBS datasets are expected and derive from the distinct techniques used: the length of sequencing reads (single end 75bp vs paired end 150 bp reads) and CG-rich DNA fragment bias of RRBS (Krueger and Andrews 2011). The mapping efficiencies in our analysis (~80%/~62% for RRBS/WGBS respectively) are comparable to other published methylome analyses. Such differences have limited or null impact on downstream analyses, as we have stringent quality and coverage filters in place to avoid any technical biases. We will provide the technical explanations for the differences in mapping rates in a supplementary text in the revision, citing relevant studies.
In addition, the choice of liver tissue needs to be justified.	Liver is a key organ involved in dietary metabolism and hormone production, and is involved in haematopoiesis. We know that the ecomorphs differ in their diet (Fig.1d and (Malinsky et al. 2015) and live in habitats substantially differing in oxygen levels (Fig.1b). Therefore, liver is a highly relevant organ to investigate the role of epigenetic in phenotypic diversification. Moreover, the liver is a largely homogenous organ, composed at >70% of hepatocytes, which limits confounding variation associated with heterogenous cell populations when performing methylation studies. In the revision we will explain the rationale behind the use of liver tissue.
fig. 1d: Maybe I am missing something, but I do not find any text related to fig. 1d. This panel suggests that there is a clear difference between food choice between littoral or river population vs. benthic population, but no difference between littoral and river populations. However, this panel is somewhat contradicted to the PCA results in fig. 1e-f, where the three populations are separated. In addition, data used for fig. 1d were generated from muscle tissue, which is different from the tissue used for RRBS and WGBS, so it is hard to cross-validate or combine results between different panel.	Figure 1D shows differences in stable isotope ratios, that provide evidence for different food source/diets. We will describe the differences more clearly in the manuscript. Muscle tissue is commonly used as a time-integrated indicator of the diet of the fish as whole, and it is simply used here to illustrate the contrasting diets of the three focal populations to provide the reader with ecological context. Our analyses do not depend on the usage of the same tissue as in the RRBS/WGBS samples. We will clarify this point in the revision.
For fig. 1e-f: It would be good to provide explanations for why PC1 and PC2 explain different variation in the same tissue between RRBS and WGBS.	The two PCA plots (Fig 1e and 1f) reveal remarkably similar methylome patterns of variation, given that we are dealing with two very different datasets. Roughly 24 million CpG sites are analysed in the case of WGBS, which is reduced to 3-4 million in the case of RRBS. In both datasets, the major source of variation is among

	different populations [PC1]. As suggested already above, in the revision we will explain the main technical differences between RRBS/WGBS, including references.
Lines 103-105: It is quite confusing why WGBS data were chosen for DMR analysis, considering only 2-3 samples were sequenced for each population. There is clearly a trade-off between the number of CpGs analyzed and the statistical power to detect true DMRs, and choosing WGBS based on very low sample size over RRBS based on good sample size needs to be justified.	In the revised manuscript we will clearly mention the rationale behind the use of only the WGBS dataset for DMR prediction. Both RRBS and WGBS datasets show the same pattern of methylation variation segregating primarily among populations. Therefore, we chose to focus on WGBS as it enables DMR prediction genome-wide, without the length or DNA fragment biases that are associated with RRBS. Although DMRs predicted using the RRBS dataset will be based on a bigger sample size, only ~10% of the genome will be analysed compared to WGBS, which we believe would be a substantive limitation for our analysis. Nevertheless, to address this point and clearly highlight the high level of similarity between the two datasets, we will expand on the DMR analysis, and test if the same DMRs are found using WGBS and RRBS. Based on results of previous analyses, we expect to be able to show DMRs found using RRBS are as present in the WGBS dataset, justifying our choice of using WGBS-derived DMRs.
Lines 112-118: Using genetic divergence result from a previous study is OK, but it would be better to directly extract SNPs and estimate genetic divergence from bisulfite sequencing data. Also, I do not agree that a distant distance between HDR and DMR suggests no association. It could be due to trans-acting genetic variants controlling methylation variation, so the statement between these lines are not well supported, especially considering the association between SNPs and DMRs were not specifically tested in this study.	In the revision we will explain that our analysis cannot exclude trans-acting effects of SNPs on methylation variation. Nevertheless, our analysis suggests that, locally, DMRs are not a key force driving HDRs - which we believe is an important result.
Lines 119-130: There is a flaw in the genomic feature analysis. The authors did not give a precedence when features were overlapped, so fig. 1i could be significantly biased due to repeatedly counting DMRs overlapping with multiple features.	Genes were counted only once even when associated with multiple DMRs (per genomic features). We will clarify this point in the method section. This is not a flaw, but key detail regarding our methodology was missing.
Lines 143-144: Implying adaptation directly from the genomic locations of DMRs is not suitable, especially without measuring fitness. I would suggest the authors to soften language here.	We agree with the reviewer and will rephrase the sentence, removing the term adaptation.
Fig. 2: In fig. 2a, One of the littoral fish is clustered with riverine fish, but there is no explanation for this. I would exclude this fish in downstream analysis. In addition, there are two different read lengths (100 vs. 150 bp PE) used in RNA-seq. Is there any batch effect due to read	We will include a PCA plot to better visualise overall RNAseq variation in two dimensions (rather than a single dimension of hierarchical clustering). We have confirmed an absence of confounding variation due to sequencing read length and batch effects. We will

length? What is the correlation between samples from the same ecomorph but were sequenced in different read length? Finally, the color spectrum in fig. 2b is qualitative and not very informative, it would be good to show absolute or scaled FC instead of 'lowest'/'highest'.	include such graphs and discuss the associated results in supporting text. The colour gradient in Fig 2c was incorrectly labelled - it represents Z-score associated with gene expression (i.e., scaled gene expression for each gene). This will be corrected.
Lines 183-203: Do the genes, epoR, cHbB, ScI/Tal1, etc., have hypomethylated promoters? Because the authors emphasized the overrepresentation of DMRs in DEGs, I guess the examples in this paragraph are genes with decreased methylation in promoters but increased transcription activities, but I do not see any clear statement about such information. Also, it would be good to report or depict the correlation between methylation levels and transcription activities, either using genome-wide data or DMRs-DEG pairs.	We will include a plot showing the correlation between methylation and transcription, and clarify in the text the interpretation of this result.
Line 235: I do not understand the process to call 'reset DMRs' and 'fixed DMRs'. I believe some DMRs identified between riverine and benthic/littoral populations will have similar methylation level difference in common garden environment, which can be called as 'fixed', but how about methylated regions with decreased methylation difference between, say, benthic and riverine populations? Do you also consider them as 'fixed DMRs'? Finally, it seems that methylation levels also changed between wild and common garden environments within the same population, so even for some DMRs you called 'fixed' between populations, they can be 'reset' or 'plastic' within populations.	We will clarify the text explaining differences between fixed and reset methylome patterns. We will more clearly emphasise how methylation is typically plastic (reprogrammed) within populations, as we see in the transition from wild and common garden conditions in both Lake Masoko populations.
Extended Fig. 8: following my last comment, in wild samples, the variation among riverine samples is higher than littoral or benthic samples, but lower in common garden environments than littoral or benthic samples, suggesting a large number methylated CpGs were plastic or responsive to environmental transition, especially in riverine populations, which adds more complexity in defining reset and fixed DMRs. One possible way to identify transgenerationally stable CpGs is to compare methylation levels of F1 and F2 samples under common garden environment, similar to Heckwolf et al. 2020. Sci. Adv. and Hu et al. 2021. Genetics. And to identify reset/plastic and fixed DMRs, it should be better to reciprocally transplant wild samples to opposite environments, e.g., transplant riverine samples to benthic or littoral environment, and identify CpG sites with altered methylation levels that are explained by ecomorph (fixed) or environments (reset/plastic).	We found that from our common garden experiment that within one generation methylome variation in Lake Masoko ecomorphs is drastically reprogrammed compared to river methylome patterns. Therefore, it is the methylome of river fish that appears most robust to environmental perturbation (see PCA Ext Dat Fig 8c and heatmap Fig 3a), We will rewrite the text reporting this result to better highlight how the methylome of Lake Masoko ecomorphs is largely reprogrammed to resemble the ancestral-related river methylome patterns. We agree with the reviewer that our work lays groundwork for further multigenerational experimentation. The reciprocal transplant experiment proposed would be technically and ethically challenging. Although this further work would be highly informative, it is beyond the scope of this current paper.

Line 238: What is the biological meaning or explanations for reset DMRs having longer length than fixed DMRs? Are you suggesting longer length correlates with higher epimutation rate?	That the two DMR classes are characterised by a least one distinct property (length) implies there may be biological/functional differences associated with them. Further work will be required to characterise any differences. We do not make any suggestion of association between DMR class and epimutation rates.
Line 253: While I agree that, to some extent, the fixed DMRs represent population-specific methylation patterns, because only liver tissue was used in methylation analysis, it is hard to say such patterns are also tissue-independent.	We agree with the reviewer and will rewrite the sentence.
Line 355: Why sequencing WGBS samples in two different platforms? How did you incorporate and correct data from these two platforms?	Samples were sequenced on different platforms due to logistic reasons (discontinuation of the first machine used), however the technology remains the same (Illumina HiSeq4000 and NovaSeq). Only one wild shallow and one wild benthic were sequenced with NovaSeq (on the same flow cell to minimise technical variation). We could not identify any evident bias in methylome variation associated with the two Illumina machines used, using PCA or hierarchical clustering analyses (see Fig.1f and Ext Data Fig 8c). We will specifically discuss the difference in platform used in supplementary information, and clearly label samples sequenced on NovaSeq in supplementary figures.
Line 369: CpGs with only at least 4 reads to be included in downstream analysis seem less favorable due to low coverage. Can you provide justifications for such low depth cut-off? Also, the coverage for genome needs to be provided for WGBS.	We used 4 as the minimum number of sequence read (coverage) required at any single CpG site - which is a widely used threshold for most methylome analyses. However, the median coverage at any single CpG was ~8-14x for all WGBS samples, and it is around 40x for RRBS samples. These represent excellent coverages, for WGBS in particular, that are rarely seen in methylome studies. In the revision we will include a graph showing CpG coverage for all WGBS and RRBS samples. We will cite other methylome studies for comparisons.

Reviewer #2 (Remarks to the Author):	Our response
In the current study Vernaz et al present compelling evidence that DNA methylation might act as a major driver of vertebrate speciation. The study is a timely follow up of authors' recent work, which put forward the exciting hypothesis that DNA methylation participated in the phenotypic diversification of lake Malawi cichlid fish. The study is well-written, the analyses are properly executed and overall, my comments for improvements are minor.	We thank the reviewer for their positive overall comments and for highlighting the quality and timely importance of our work.
1) It is not clear from the current data how DMRs were associated to genes for the purpose of GO analyses. Did the authors employ the "nearest gene" approach or was more elaborate methodology used for this (such as GREAT for example)?	The method section will be modified to include this detail. In short, DMRs were associated with genes when located in their promoter regions (TSS \pm 1kbp), gene body or intergenic regions (1-5kbp away from closest genes).
2) How many DNMTs / TETs were identified in cichlids? Given the enrichment of DMRs in CpG-rich regions, it would be interesting to explore if these differences could be caused by expression differences of DNA methylation machinery components?	We believe it would be difficult to link any new cichlid-specific copies of DNMT/TETs to the specific DNA methylation patterns we observe. Previous work in mammals have highlighted that duplication of certain DNMT3 enzyme (for example DNMT3C) could be associated with targeted methylation of certain classes of transposable elements (TE), consisting of an arm race between host DNMTs and TE sequence evolution (see Barau et al. 2016). However, the methylome divergence we observed between Lake Masoko cichlids is not exclusively localised in one genomic feature (such as cichlid-specific TEs for example), therefore we do not believe this can result from novel DNMT/TET enzymes. From our previous work (Vernaz et al. 2021), Lake Malawi cichlids are believed to have one copy of each of the three Tet enzymes, one copy of DNMT1 and multiple copies of DNMT3s, just like in zebrafish (Campos, Valente, and Fernandes 2012; Seritrakul and Gross 2014; Shimoda et al. 2005; Smith, Collins, and McGowan 2011) - however we have not characterised or identified new DNMTs/TETs in Lake Masoko cichlids. This could be the subject of further work, but would require extensive RNAseq, proteomics analyses in different tissues, together with full genomic scan, in addition to lab experiments to elucidate the sequence-specificity of such proteins, which we believe is beyond the scope and the main hypothesis of this manuscript.
3) Also, TET proteins which actively remove DNA methylation are oxygenases and thus require oxygen for their catalytic activity. Could the difference in oxygen concentration therefore act as a driver for such epigenetic diversification? i.e less, oxygen  less TET activity  less capacity for 5mC removal  more hypermethylated	This hypothesis is interesting, but our current work is not able to address this question and it is beyond the scope of this study. Nonetheless, it is important to note that we observe a widespread increase in methylation in both the shallow and benthic populations (Fig. 1G). This is despite differences in environmental oxygen levels (the benthic habitat of the lake is almost anoxic; the shallow habitat

regions?	remains well oxygenated; Fig.1D). Therefore, blood oxygen levels might be sufficient in both ecomorphs for an efficient TET enzyme activity.
4) Instead of speculating about the possibility of fixation of epigenetic differences, could the authors perform RRBS / WGBS on sperm (or eggs, but more likely sperm given the current literature) on fish from their “garden experiment”? Given the strong enrichment for CpG-rich regions even RRBS might do. Such an experiment would demonstrate the potential for inter- or trans-generational inheritance of acquired methylation differences. Even if only a small subset of DMRs is picked up in sperm, this would be an extremely valuable piece of data.	We agree with the reviewers that our study also lays the groundwork for further experiments, especially regarding the assessment of the multigenerational inheritance of methylome patterns. Our lab is currently initiating work on this aspect, but such experiments represent a significant amount (several years) of work, and are beyond the scope of this current paper.
5) Reference 24 is wrong – the authors probably refer to Skvortsova et al, 2019 Nature Comms, rather than Skvortsova et al 2018 (Nat Rev Mol Cell Biol), where the inheritance of paternal methylome patterns in the zebrafish germline is not discussed.	This will be corrected.

Reviewer #3 (Remarks to the Author):	Our response:
This work by Vernaz and colleagues assess variation in DNA methylation in benthic versus littoral populations of A. calliptera cichlids from river and Lake populations. The role of such epigenetic variation in an ongoing sympatric speciation event like this is largely unexplored. This work assesses overall patterns of methylation, potential relationship to gene expression, and plastic potential in a common garden experiment. While a strong work, there are a number of instances of missing logic of analyses or weak explanations that would strengthen conclusions. These are needed to bring this work from statistical connections in large-scale genomic analyses to the biological relevance and importance for evolutionary processes as pitched.	We are pleased the reviewer recognises the novelty of the work, and we would be pleased to clarify the sections of the paper the reviewer highlights.
Major comments: A. Biological relevance of methylation differences	We agree that we could be clearer on when methylome divergence takes place and the relevance to speciation, and we will expand and discuss this point further in the

A1. The suggestion in the paper is that differences in metabolism due to methylation are driving benthic versus littoral adaptation. However, samples were taken from these 2 environments that are very different in terms of oxygen, etc, which will cause metabolism differences. Is it that methylation differences drive this habitat divergence or that methylation differences in the liver become distinct once animals diverge in habitat usage?	revised manuscript. Our hypothesis is that: i) methylation differences occur after the fish arrive in the specific habitats (e.g., deep water). They may have low initial fitness, which does not matter initially given the abundance of food and lack of competition. After a period of acclimation due in part to DNA methylation affecting expression of key ecologically important genes, they are able to thrive. The positive traits are present in offspring, through inheritance, and through exposure to the same environment from young. ii) any new coloniser from the shallow environment will now be outcompeted in the deep water, and relative fitness will matter now due to competition. Equally, a fish of a deep-water origin will struggle in shallow water environment because they will not have been reared in such shallow environment, and/or possess the beneficial epialleles. iii) This process becomes amplified when genetic variants start to become fixed. This is important to discuss in the paper, however it does not highlight any flaws in our analysis or question our conclusions.
A2. L63 notes that system is addressing sympatric speciation. However, the populations used do not live in sympatry, but are already distinct in their habitat usage.	The exact definition of sympatric speciation is highly debated. We will replace any reference to “sympatric speciation” with “ecological speciation”.
A3. The connection between differential methylation patterns and transcription factor binding was not clearly defined. Is the suggestion that differential methylation is happening in a way that only impacts certain TFs? How would this work biologically? Or is it that those genes active in the liver are enriched for those TF binding sites in their promoters? See also point C5.	In the revised manuscript we will expand on the relationship between DNA methylation and binding affinity of transcription factor in the text, and further discuss how changes in DNA methylation can impact binding affinity of certain TFs (using examples we show in the manuscript). We will also discuss how changes in TF activity can also result in methylation changes, adding relevant references.
A4. I love the common garden experiment as it begins to address considerations such as that in point A1. However, details of this seem unclear. Were the conditions in this experiment similar to any of the natural environments? For example, were the tanks similar in oxygen and temperature of the littoral, but not benthic fish? How would this bias analysis or alter conclusions? How long was the experiment conducted, what is the logic of that choice, what is the developmental age of the fish at start and end of the experiment, and how does that affect the data obtained? Again, is it that you are catching all the regions of the genome that are responsive to environmental changes or those that are driving habitat changes and speciation? Some of the analysis in this section also highlights limitations of drawing conclusions from large-scale enrichment analyses. For example, GO analysis shows enrichment for fixed DMRs in developmental genes, but genes associated with neuron	We thank the reviewer for the positive comment. We will include all the requested experimental detail in the methods, including a rationale for the design. In brief, to minimise confounding variation, all experiments involving common garden fish were performed in the same laboratory conditions, at the same time and for the same duration. Fish were euthanised at the same biological stages (i.e., adult males). With regard to GO analysis, we have previously shown that fully differentiated organs, such as liver or muscle, can harbour epigenetic memories of early life developmental stages, carried over across cell divisions (Vernaz et al. 2021; see also Bogdanović et al. 2016). Approximately half of liver methylome divergence among Lake Malawi cichlid fishes has been associated with genes with functions related to embryonic and

projection, neural tube development, and mastoid bone, none of which make sense given the tissue analyzed was liver. While interesting, the section about effect of fixed DMGs and brain development in L250-261 thus reads as examples picked from a list rather than making a functional connection between liver activity and brain development.	developmental processes. While possibly functionally irrelevant in fully differentiated organs, such as the liver, it reflects that methylome divergence can be associated with developmental processes, possibly participating in core phenotypic diversification (Bogdanović et al. 2016; Vernaz et al. 2021). We will expand on those points in the text and include relevant references. Moreover, we will focus on the gene groups that are linked functionally to liver functions.
B. Choice of samples B1. There was no explanation of why liver tissue was used for this analysis. What biases and limitations of conclusions does this introduce into the analysis?	Reviewer 1 made the same point, that we have addressed above. In the revision we will explain the rationale behind the use of liver tissue, and highlight the limitations.
B2. Also not discussed is why males only were used. What is the logic of choosing males with bright nuptial coloring, and could this add any bias to your analysis or our understanding of evolutionary processes?	Reviewer 1 made the same point, that we have addressed above. We will clarify the rationale for our choice of samples to analyses in the revised text, and highlight the limitations.
C1. PCA of data sets is stated as having segregation in L78-79, but this is not supported with statistical analysis such as an ANOVA. Does hierarchical clustering of all regions, not just differentially with your specific cutoffs, also show segregation of populations?	We will formally test for differences between groups using ANOVA and PC1 and PC2 as response variables. Regarding the second point, it is important to note that PCAs (Fig.1e, f) were performed in an unbiased manner, using all mapped data. Specifically, Fig.1f shows the variance in methylome across all mapped CG sites genome-wide and shows clear segregation of populations.
C2. What is the logic of cutoffs used for analysis? For example, differentially methylated regions were called based on >25% mCG difference and >4 CpG sites. What difference would using 5 sites make in the conclusions? Why 4 sites? This is stated as "significantly" different in L93 and a p-value is given, but there is not information on how this pvalue is calculated in methods L368-37.	Reviewer 2 made a similar point. Those parameters were based on previous methylome studies and are recognised as being the ideal/default options to reliably identify DMRs (Krueger and Andrews 2011; Wu et al. 2015). Again, those are minimum cut-offs, but most DMRs consist of >4 CpG sites (median 15) and show larger differences (median difference of ~45% ; see Ext Data Fig 3C). We will add information on statistic tests performed to identify significant DMRs in the main text and methods - DMR prediction and associated statistics were performed following a well-established and recognised pipeline (Wu et al. 2015). We will also include graphs plotting CpG site count against read coverage for all DMRs predicted between the ecomorphs.
C3. Assigning non-coding regions to gene function is a critical assumption that can bias downstream assessment of biological processes. What is the quality of the	Annotation of the Maylandia zebra reference genome is of high quality, and we used the latest annotation file. We cautiously defined promoter regions (1kbp around TSS),

annotation of genes in the genome used for this (a critical factor for these analyses)? Why did you chose 1kb for a promoter versus 5kb? How were genes assigned to a specific gene for GO analysis?	based on published literature, as we lacked any other annotation files (such as histone marks). We appreciate the limitations of this, as promoters can sometimes expand beyond this 1kbp region. We will also run a sensitivity analysis to see how results change when increasing to 5kbp. To assign a gene to a DMR in intergenic regions, we use the closest-to-gene approach, as used in most methylome studies. We were cautious, however, only assigning genes to DMRs within 5kbp. We will include a detailed rationale in the Methods, citing other methylome work.
C4. Analysis for transcription factor binding motifs appears to only be conducted on DMRs in one particular region (outside the gene body). How does this compare to the groupings presented in Figure 1I? How do conclusions change when different genomic features are used? Is there a reason only 2 of 3 combinations are shown (river vs littoral is omitted in extended data Figure 3F)?	We will repeat this analysis for gene bodies as well, enabling a more complete comparison. The text will be reformulated accordingly.
C5. What is the directionality of relationships between DMRs and DEGs? Given that DNA methylation has a predictable effect on gene expression, are those regions with high methylation associated with low gene expression? This is explored for eklf and epoR in L183-203. However, the cHbB locus is exclusively expressed in benthic fish that hypermethylation, and not expressed in fish populations with low to no methylation. Thus, further explanation is needed to clarify clear how the biological function of methylation relates to gene expression. Without it, the text in L166-213 reads more like anything differentially expressed in the methylation data and transcriptional data were simply compared for overlap, regardless if the biology lines up.	We will discuss this link in detail and include a graph on how expression and methylation covary along the gene (see Fig.1e in Vernaz et al. 2021). In brief, we observe a negative correlation between methylation at promoters and gene expression, however this link is not always obvious/predictable. We will briefly discuss the reasons for varying responses of gene expression to methylation in the revised manuscript.
Minor comments: D. Abbreviations. I encourage authors to critically evaluate if all abbreviations are necessary. Three specific examples, though I encourage a careful read for this throughout: (1) HDR and DMR get confusing for readers and are not always clearly defined in figure legends. (2) oCGI, pCGI, and LCR abbreviations are only used once, so there does not seem to be a utility to adding more abbreviations here. (3) GO categories MF, BP, CC, and KEGG in Figure 3 are never defined in the legend or the text.	We will limit the use of abbreviations throughout the text.

Reviewer #4 (Remarks to the Author):	Our response:
In this paper, the authors have investigated divergence in DNA methylation pattern of the liver between littoral and benthic ecomorphs of an African crater lake cichlid fish. They found that these two ecomorphs diverge in DNA methylation patterns in several genes related to ecological adaptation and also report that some of the variations may be associated with gene expression variation. They also claim that the majority of the variations are reset during a common garden experiment, but some epigenetic variations are genetically determined. I agree that the topics of this study, namely the role of DNA methylation in incipient speciation, is one of the important topics in evolutionary biology. However, I found several weaknesses in the paper.	We are pleased that the reviewer recognises the importance of the work, and will be pleased to address the points made in a revision.
First, they used only 2-3 wild-caught fish for the the whole genome bisulfite sequencing (WGBS). Although they used more individuals for reduced representation bisulfite sequencing (RRBS), the majority of the analysis are based on WGBS. I wonder whether they can validate this small sample size by comparing the results between WGBS and RRBS. Some of the RRBS sites should be included in WGBS data. I suggest that they should test how consistent the results of WGBS with N=2-3 and RRBS with N=11-12 are in such overlapping sites.	Reviewer 1 made a similar point, that we respond to in full above. In brief, we will re-analyse both WGBS and RRBS to test if the same DMRs are found using WGBS and RRBS. This will validate the use of the WGBS dataset for subsequent analyses. We believe this point will strengthen the conclusions of the paper.
Second, I found no evidence for supporting that the DNA methylation differences they found are adaptive. I agree that DNA methylation differences in physiologically important genes are consistent with their claim, but they are not evidence for the adaptive roles of DNA methylation. Do the differentially methylated sites show any signatures of divergent selection?	Reviewer 1 made a similar point, and we agree. We cannot unequivocally conclude the methylome differences we observe are adaptive, without performing fitness experiments, which would be beyond the scope of this study. We will modify the text to clarify this point.
Third, they found that some of the DNA methylation differences detected in the wild-caught fish were reset in the lab-raised fish. However, I am afraid that this may simply result from the chance effect due to the small sample size of the wild fish (i.e., morph-specific DNA methylation in the wild fish may be an artifact).	With respect, we do not agree with the reviewer's comment here. The suggestion that ecomorph-specific DNA methylation differences in wild fish are an artefact is NOT supported by the RRBS dataset (Fig.1e). This is the reason why we included both the WGBS and RRBS dataset (genome-wide and population levels). We will reanalyse the RRBS and WGBS datasets to cross-validate the relevance of using wild WGBS data for the common garden experiment. Specifically, we will assess the numbers of wild RRBS-DMRs that are reprogrammed/fixed in WGBS data from common garden reared fish. We expect to see similar amount of reset/reprogrammed wild DMRs as we have seen in the WGBS DMRs.

Overall, I think that this paper tries to address an important topic, but the conclusions on the adaptive significance and the inheritance of DNA methylation are not very solid.	The adaptive significance is not tested in our study, and we do not claim that it has been tested. For that we would need to associate patterns of DNA methylation with differential fitness of individuals across different habitats. Here we show a) patterns of DNA methylation do differ between environments, b) those differences are associated with gene expression and c) there is evidence for cross-generational inheritance of epigenetic markers. Each of these patterns is known from other systems, but we are unaware of any other study that has included them collectively in a speciation context. Our evidence is collectively supportive of a role for epigenetic divergence in speciation, but we agree the direct association between epigenomic variation and fitness needs to be studied, and that will be highlighted in the revised discussion.
Additionally, given the fact that there are already several papers reporting DNA methylation differences between young species or ecomorphs in darter fishes (eg. Smith et al. 2016 Mol Ecol) and stickleback fishes (Hu et al 2021 Genetics), I have difficulty in finding novelty of the work.	There are striking differences between our work and that highlighted by the reviewer. The darter study used MSAP loci (a rather antiquated and imprecise method) and did not consider the variation in relation to gene expression or inheritance. It focussed on allopatric methylome variation in one species (which might have nothing to do with speciation) and variation among long-diverged species pairs (3-18Mya). The stickleback study focussed exclusively on RRBS data. It showed evidence of divergence of allopatric marine and freshwater ecotypes, and cross-generation inheritance. Thus, the results support those of our study. However, the stickleback study does not quantify variation at the genome scale, or in relate methylome variation to gene expression. Notably, the study suggests epigenetic differences among populations across a geographic distance of 150 kilometres, while our study – remarkably - suggests divergence across only 20 metres! We will cite and discuss this work in more detail in the revised version of the manuscript.
How did you treat the sex? Is there any difference in sex difference in DNA methylation and gene expression? Sometimes, the sex explains the largest amount of gene expression variation.	Reviewer 1 made the same point, that we address above. We will clarify the rationale for our choice of samples to analyses in the revised text, and highlight the limitations.
L181-182: Is there any correlation between the direction of DNA methylation and gene expression (i.e., high methylation is associated with low expression etc)?	Reviewer 3 made a similar point. We will include a graph showing the correlation between methylation and gene expression, and will discuss this in the text.

L408: Is it common to exclude the 1 kb downstream of TSS for defining the gene body?	Reviewer 3 commented on this, and we have responded above. We excluded 1kbp downstream of TSS to avoid overlap between TSS and gene body categories. We will provide a full rationale for our definitions of genomic annotations//regions in the revised manuscript.
---	---

References:

- Barau, Joan, Aurélie Teissandier, Natasha Zamudio, Stéphanie Roy, Valérie Nalesso, Yann Hérault, Florian Guillou, and Déborah Bourc'his. 2016. "The DNA Methyltransferase DNMT3C Protects Male Germ Cells from Transposon Activity." *Science* 354(6314):909–12.
- Bogdanović, Ozren, Arne H. Smits, Elisa de la Calle Mustienes, Juan J. Tena, Ethan Ford, Ruth Williams, Upeka Senanayake, Matthew D. Schultz, Saartje Hontelez, Ila van Kruijsbergen, Teresa Rayon, Felix Gnerlich, Thomas Carell, Gert Jan C. Veenstra, Miguel Manzanares, Tatjana Sauka-Spengler, Joseph R. Ecker, Michiel Vermeulen, José Luis Gómez-Skarmeta, and Ryan Lister. 2016. "Active DNA Demethylation at Enhancers during the Vertebrate Phylotypic Period." *Nature Genetics* 48(4):417–26.
- Campos, Catarina, Luisa M. P. Valente, and Jorge M. O. Fernandes. 2012. "Molecular Evolution of Zebrafish Dnmt3 Genes and Thermal Plasticity of Their Expression during Embryonic Development." *Gene* 500(1):93–100.
- Krueger, Felix and Simon R. Andrews. 2011. "Bismark: A Flexible Aligner and Methylation Caller for Bisulfite-Seq Applications." *Bioinformatics* 27(11):1571–72.
- Malinsky, Milan, Richard J. Challis, Alexandra M. Tyers, Stephan Schiffels, Yohey Terai, Benjamin P. Ngatunga, Eric A. Miska, Richard Durbin, Martin J. Genner, and George F. Turner. 2015. "Genomic Islands of Speciation Separate Cichlid Ecomorphs in an East African Crater Lake." *Science* 350(6267):1493–98.
- Munby, Hannah, Tyler Phillip Linderoth, Bettina Fischer, Mingliu Du, Grégoire Vernaz, Alexandra M. Tyers, Benjamin P. Ngatunga, Asilatu H. Shechonge, Hubert Denise, Shane A. McCarthy, Iliana Bista, Eric A. Miska, M. Emília Santos, Martin J. Genner, George F. Turner, and Richard Durbin. 2021. "Differential Use of Multiple Genetic Sex Determination Systems in Divergent Ecomorphs of an African Crater Lake Cichlid." *BioRxiv* (doi.org/10.1101/2021.08.05.455235):1–46.
- Potok, Magdalena E., David A. Nix, Timothy J. Parnell, and Bradley R. Cairns. 2013. "Reprogramming the Maternal Zebrafish Genome after Fertilization to Match the Paternal Methylation Pattern." *Cell* 153(4):759–72.
- Seritrakul, Pawat and Jeffrey M. Gross. 2014. "Expression of the de Novo DNA Methyltransferases (Dnmt3 - Dnmt8) during Zebrafish Lens Development." *Developmental Dynamics* 243(2):350–56.
- Shimoda, Nobuyoshi, Kimi Yamakoshi, Akimitsu Miyake, and Hiroyuki Takeda. 2005. "Identification of a Gene Required for de Novo DNA Methylation of the Zebrafish No Tail Gene." *Developmental Dynamics* 233(4):1509–16.
- Skvortsova, Ksenia, Katsiaryna Tarbashevich, Martin Stehling, Ryan Lister, Manuel Irimia, Erez Raz, and Ozren Bogdanovic. 2019. "Retention of Paternal DNA Methylome in the Developing Zebrafish Germline." *Nature Communications* 10(1):3054.
- Smith, Tamara H. L., Terry Mark Collins, and Ross A. McGowan. 2011. "Expression of the Dnmt3 Genes in Zebrafish Development: Similarity to Dnmt3a and Dnmt3b." *Development Genes and Evolution* 220(11–12):347–53.
- Vernaz, Grégoire, Milan Malinsky, Hannes Svoldal, Mingliu Du, Alexandra M. Tyers, M. Emília Santos, Richard Durbin, Martin J. Genner, George F. Turner, and Eric A. Miska. 2021. "Mapping Epigenetic Divergence in the Massive Radiation of Cichlid Fishes." *Nature Communications* 1–25.
- Wang, Xuegeng and Ramji Kumar Bhandari. 2019. "DNA Methylation Dynamics during Epigenetic Reprogramming of Medaka Embryo." *Epigenetics* 14(6):611–22.
- Wu, Hao, Tianlei Xu, Hao Feng, Li Chen, Ben Li, Bing Yao, Zhaohui Qin, Peng Jin, and Karen N. Conneely. 2015. "Detection of Differentially Methylated Regions from Whole-Genome Bisulfite Sequencing Data without Replicates." *Nucleic Acids Research* 43(21):gkv715.

Decision Letter, first revision:

2nd November 2021

Dear Professor Miska,

Thank you for your letter asking us to reconsider our decision on your Article entitled "Epigenetic Divergence during Early Stages of Speciation in an African Crater Lake Cichlid Fish". After careful consideration we have decided that we would be willing to consider a revised version of your manuscript.

Please note that while we agree with your plan for revision, we feel that we will not be able to make a final decision on whether to resume the review process until we see the revisions themselves. We understand that some of the concerns raised by the reviewers (particularly regarding heritability and adaptation) is beyond the scope of the current work and these issues can just be discussed. However, we feel that the outcome of the planned analyses, particularly regarding the several concerns raised on the RRBS and WGBS findings, is still uncertain at this stage and will be critical to the decision on whether to send the revised manuscript back to review.

Along with your revised manuscript, you should also submit a separate point-by-point response to all of the concerns raised by the reviewers, in each case describing what changes have been made to the manuscript or, alternatively, if no action has been taken, providing a compelling argument for why that is the case. If we feel that a substantial and likely successful attempt has been made to address the reviewers' comments, this response will be sent back to the reviewers - along with the revised manuscript - so that they can judge whether their concerns have been addressed satisfactorily or otherwise.

As noted above, however, we will be reluctant to trouble our reviewers again unless we believe that their comments have been resolved in full.

- ensure it complies with our format requirements for Articles as set out in our guide to authors at www.nature.com/natecolevol/authors/index.html
- state in a cover note the length of the text, methods and legends; the number of references and the number of display items.

Please ensure that all correspondence is marked with your Nature Ecology & Evolution reference number in the subject line.

Please use the following link to submit your revised manuscript:

[REDACTED]

Yours sincerely,

[REDACTED]

Author Rebuttal, first revision:

Point-by-point response to reviewers - NATECOLEVOL-210814388A-Z

Reviewer #1 (Remarks to the Author):

In this study, Vernaz et al. focused on the early stage of cichlid fish species radiation by analyzing DNA methylation and gene expression divergence between the ancestral riverine population and two derived lake populations. While the cichlid system is a well-studied system, results from this study add to the small body of studies focusing on the relationship between epigenetics and speciation, and provide some new insights into how an environmentally sensitive and rapid changing molecular mechanism contributes to the macroevolutionary event. That said, although the topic covered in this study is of immediate interest to the general community of ecology and evolution, I have both general comments on experimental design and specific comments on methods that prevent this manuscript to be published in its current shape. I would suggest the authors to thoroughly revise their MS before resubmission or submitting to a different journal.

[Authors' response]: We thank the reviewer for their time to review our work and for highlighting the timely importance and value of our work to the ecology and evolution research community.

General comments:

1. The inclusion of riverine population is a little bit weird. It makes reader confusing on the focus of this study: are you focusing on the speciation between riverine and lake populations, or are you focusing on the divergence between the benthic and littoral ecomorphs? I think the authors may need to make the focus of this paper more clear.

We agree the inclusion of the riverine population could be better explained. Two major evolutionary steps characterise the Lake Masoko cichlid radiation: first, the colonisation of the lake by an ancestral riverine population, forming the littoral population. Second, the colonisation of the benthic habitat by the shallow population, forming the benthic population. To fully study this instance of speciation, it is therefore important to include the riverine population (phenotype and genotype). We have revised the abstract and the text to clarify the importance of including and studying the riverine population.

2. The combination of WGBS and RRBS is very confusing (also see my specific comments below). It seems that the data from RRBS were only used in one PCA, with its result quite different from the PCA from WGBS data.

RRBS enriches for certain CG-rich loci (limited overall genome coverage) but enables affordable population-scale methylome analysis of larger sample size. WGBS enables us to characterise and validate the methylome landscape genome-wide and in a less biased manner (compared to RRBS). In this case, both RRBS and WGBS datasets strongly indicate population-specific methylome divergence (see Fig1e-f), reaching the same conclusion and adding support to our core hypothesis of functional epigenetic diversity during the speciation.

We have now performed a cross-validation analysis between the two bisulfite datasets (see graph below, which is now Extended Data Figure 4, and see Supplementary Notes), providing support for the use of the WGBS dataset for all subsequent analyses. In summary, methylation variation at DMRs predicted using whole-genome methylome population was highly correlated with variation in the RRBS populations. WGBS offers a genome-wide, unbiased resolution and therefore was preferred for DMR analysis and subsequent analyses. We have now explained the rationale behind the combinatory use of those techniques in the text, and have added the cross-validation analysis as Extended Data Figure 4.

Extended Data Figure 4. Cross comparison between RRBS and WGBS datasets. **a.** Unbiased hierarchical clustering and heatmap showing the Spearman's correlation scores for methylation levels in all RRBS samples at DMRs predicted using WGBS overlapped with RRBS samples ($n=413$). RRBS samples highly segregated according to the three populations (higher within-group correlations than between-group correlations) at DMRs found using WGBS. This highlights high correlation between RRBS and WGBS methylome divergence. **b.** Heatmap of methylation levels at DMRs found between WGBS samples (3 pairwise comparisons; see Fig.1d) for RRBS and WGBS samples (mean mCG/CG across samples). R, river; L, littoral/shallow; B, benthic. WGBS: benthic ($n=3$ biological replicates), littoral ($n=2$) and river ($n=2$); RRBS, benthic ($n=12$), littoral ($n=12$), river ($n=11$).

3. Following my last comment, the methylation-gene expression correlation analysis also can be improved. The samples used in WGBS/RRBS were not the same samples or only a subset of samples used in RNA-seq. So, the discrepancy of samples used in methylation analysis vs. gene expression analysis can potentially bias the results.

Like any biological sampling of individuals in a population, we assume that each sample analysed here is representative of a broader population. Our analytical approach does not depend on the same samples being used. We have clarified this point in the text. Specifically, in the Methods section we highlight that for some additional samples were used for RNAseq due to tissue availability.

In addition, we have now performed further analysis to investigate the overall correlation between methylation and gene expression activity. There was significant negative correlation between methylation at promoters regions and gene expression (Spearman $\rho=-0.33$; Extended Data Figure 6b), in line with an association between promoter methylation and repressed transcription - a highly conserved feature in vertebrates (Vemaz et al. 2021; Zemach et al. 2010), including Lake Malawi cichlids. We have now discussed this result in the main text and have added the detail of this analysis in the Methods section.

Extended Data Figure 6b. Association between DNA methylation and gene expression activity across promoters and gene bodies. Liver methylation levels (average mCG/CG) vary according to genomic features and transcriptional activity. All annotated genes were split into five categories based on their gene expression levels: from non-expressed genes (tpm<5; 'OFF', $n=24,598$), to expressed genes (four 'ON' categories; tpm ≥ 5), from lowest to highest gene expression activity ($n=1,269$ -1,270 per category). Methylation at promoters is significantly negatively correlated with gene expression (Spearman's rank correlation test, $\rho = -0.33$, $P < 2.2 \times 10^{-16}$). Benthic individuals are shown for this analysis and are representative of the other populations ($n=3$ biological replicates for liver WGBS [average mCG/CG] and $n=5$ biological replicates for RNAseq (liver)).

4. The choice of male fish for this study needs more explanations. I can understand that methylation and gene expression are gender-specific, but to totally exclude females from analysis needs good justification. Is this because males show more phenotypic/physiological divergence between ecomorphs? Also, because it has been suggested that epigenetic patterns at early developmental stages can often reprogram to reflect the paternal state, identifying transgenerationally stable CpGs in male fish seems not unexpected but makes me wondering the magnitude of methylation stability in females.

There are three reasons we focus on males:

1. Practically, we are more confident in our assignment of male individuals to ecomorphs.
2. In our sampling, to test our hypotheses, we aimed to maximise between-population variability, while minimising within-population variability, within the constraints of resources available.
3. Epigenetic inheritance/reprogramming varies substantially among species (Potok et al. 2013; Skvortsova et al. 2019; Wang and Bhandari 2019), yet the nature of this in cichlid species was hitherto unknown. We surmised that if epigenetic inheritance is relevant to speciation, then it must be contributing to between-morph differences in traits possessed by both sexes (habitat occupancy, diet). Since we see differences in males, then the same may well apply to females, and this is a clear hypothesis that can be tackled with future work.

We have clearly mentioned the rationale for our choice of samples to analyse in and Methods section of the revised manuscript and noted the limitations.

5. A lot of test statistics are missing (e.g., lines 139-142), which needs to be improved.

We thank the reviewer for pointing this out and apologise for this. We have now added all the information related to statistical tests performed in the revised version of the manuscript as well as in the Methods section. For lines 139-142 specifically, a hypergeometric test is used to evaluate TF motif enrichment.

Specific comments:

Lines 37-38: Do you refer to methylation or expression variation between genes involved in key biological processes? This sentence sounds like you are talking about two things, with transcriptional changes in 'ecologically-relevant genes' as the result of 'methylome divergence between populations'? If so, I would suggest adding a logical link between the methylome variation part and the transcriptional change part in this sentence.

We have reformulated the abstract to better clarify the association between methylation levels and transcriptional activity we observed.

Also, it seems some key information are missing in the Summary section, which are (1) a brief description of how the habitat differs between ecomorphs. It makes readers hard to understand how the genes and their related functions, e.g., steroid metabolism, erythropoiesis, are ecologically relevant to speciation without a context of environmental difference between habitats. (2) Definition of early stage speciation. Why are you interested in early stage but not other stages, and how do you know the ecomorphs were at the early stage of speciation?

*(1) We have rewritten the abstract to better describe the ecological differences between the ecomorphs.
(2) We are interested in these early stages as this is arguably the most challenging phase to study. It is easy to identify long-diverged species pairs, but identifying ongoing divergence where we know the precise ecological and historical context of the system is extremely rare. We have "caught" these populations undergoing speciation, and that is what excites us.*

Line 48: I agree that the role of epigenetic processes in speciation is less understood, but there are some recent studies focusing on epigenetic difference between ecotypes/populations in a wide range of taxa, for example, Darwin finches (Skinner et al. 2014. *Genome Biol. Evol.*), medaka (Ichiwaka et al. 2017. *Nat. Comm.*), cichlid fish (Kratochwil & Meyer. 2015. *Mol. Ecol.*; Franchini et al. 2021. *Mol. Biol. Evol.*), general teleost fish (Desvignes et al. 2021. *Mol. Biol. Evol.*), and suggesting DNA methylation, histone modifications, and microRNAs may contribute to speciation. In addition, other studies have demonstrated the gene expression mechanisms underlying cichlid fish divergence/speciation (e.g., Rajkov et al. 2020. *Mol. Ecol.*; El Taher et al. 2021. *Nat. Ecol. Evol.*). So, I would suggest the authors to cite at least the above papers to acknowledge the recent advances, and emphasize more on the knowledge gap of the role of epigenetic processes in early stage of speciation.

We thank the reviewer for this point. We have now cited all those references.

Lines 55-58: While F_{st} is one of the key indicators of divergence or speciation stage, more results such as gene flow may better support the statement of 'early stages of divergence'. In addition, the relatively low overall genetic divergence can be caused by relatively high gene flow between ecomorphs, so it may be more appropriate to report gene flow levels rather than F_{st} .

We thank the reviewer for this comment. However, our evidence of the nature and timescale of divergence does not solely rely on F_{st} . Key evidence is demographic modelling by Malinsky et al. (2015), which shows ecomorph divergence 500-1000 years ago (200-350 generations). Moreover, more recent work has revealed the presence of 'hybrid' individuals of varying levels of admixture, demonstrating incomplete reproductive isolation (Munby et al. 2021). We have now outlined this evidence consistent with the early stage of speciation in the revised version of the manuscript, and have cited the new study describing the presence of admixed individuals.

Lines 76-77: It is evident that the mapping rate is different between RRBS and WGBS. It would be good to have some explanations for this difference, and test for if such difference have an impact on downstream analysis. In addition, the choice of liver tissue needs to be justified.

Differences in mapping rates between RRBS vs WGBS datasets are expected and stem from the different techniques used: the length of sequencing reads (single end 75bp vs paired end 150 bp reads) and CG-rich DNA fragment bias of RRBS (Krueger and Andrews 2011). The mapping efficiencies in our analysis ($\sim 83.9 \pm 1.6\%$ / $\sim 53.7 \pm 4.3\%$ for RRBS/WGBS respectively) are high and comparable to other published methylome analyses, including fish studies (Anastasiadi, Esteve-Codina, and Piferrer 2018; Chen, Smith, and Chen 2016; Lee et al. 2020; Tran et al. 2014). Such differences have limited or null impact on downstream analyses (mapping rates are not significantly different between groups for each bisulfite method used), as we have stringent quality and coverage filters in place to avoid any technical biases (see Methods). We have now provided the technical explanations for the differences in mapping rates in the Methods and have cited the relevant studies.

Regarding the choice of tissues: liver is a key organ involved in dietary metabolism, hormone production, and haematopoiesis. We know that the ecomorphs differ in their diet (Fig.1d and (Malinsky et al. 2015) and live in habitats substantially differing in oxygen levels (Fig.1b). Therefore, liver is a highly relevant organ to investigate the role of epigenetic processes in phenotypic diversification. Moreover, the liver is a largely homogenous organ, composed at >70% of hepatocytes (Aloia et al. 2019), which limits confounding variation associated with heterogenous cell populations when performing methylation studies. In the revision we have now explained the rationale behind the use of liver tissue.

fig. 1d: Maybe I am missing something, but I do not find any text related to fig. 1d. This panel suggests that there is a clear difference between food choice between littoral or river population vs. benthic population, but no difference between littoral and river populations. However, this panel is somewhat contradicted to the PCA results in fig. 1e-f, where the three populations are separated. In addition, data used for fig. 1d were generated from muscle tissue, which is different from the tissue used for RRBS and WGBS, so it is hard to cross-validate or combine results between different panel.

We thank the reviewer for raising this point. We now use a more appropriate and powerful statistical method (accounting for the data distribution, sample size, patterns of variance). The differences should now be clearer (See revised main text, Fig.1d, Supplementary Notes [lines 41-56] and Methods).

Regarding the second point, muscle tissue is commonly used as a time-integrated indicator of the diet of the fish as whole, and it is simply used here to illustrate the contrasting diets of the three focal populations to provide the reader with ecological context. Our analyses do not depend on the usage of the same tissue as in the RRBS/WGBS samples, as ultimately all carbon and nitrogen in animals are derived from diet (we expect similar results with liver tissues for example) - see for example the review by O'Brien (PMID: 26048703; (O'Brien 2015)) on the use of Stable Isotope Ratio as biomarkers of diet in tissues. We have clarified this point in the Methods section.

For fig. 1e-f: It would be good to provide explanations for why PC1 and PC2 explain different variation in the same tissue between RRBS and WGBS.

We believe that the two PCA plots (Fig 1e-f) show remarkably similar methylome patterns of variation, given that we are dealing with two very different datasets derived from contrasting sequencing techniques. Roughly 24 million CpG sites are analysed in the case of WGBS, which is reduced to ~1 million in the case of RRBS. Nonetheless, the differences in methylome among populations using WGBS samples were reflected in the RRBS datasets (see your question 2, page 1 above). In both datasets, PC1 is mostly explained by river vs benthic, while PC2 segregates the lake ecomorphs (see Extended Data Figure 2c). We have now better explained the main technical differences between RRBS/WGBS (as already mentioned above) in the revised manuscript.

Lines 103-105: It is quite confusing why WGBS data were chosen for DMR analysis, considering only 2-3 samples were sequenced for each population. There is clearly a trade-off between the number of CpGs analyzed and the statistical power to detect true DMRs, and choosing WGBS based on very low sample size over RRBS based on good sample size needs to be justified.

For financial reasons, we used a combination of bisulfite techniques to assess methylome variation both at a population (RRBS) and whole-genome levels (WGBS). Those two techniques are therefore complementary, and both have limitations (sample size, reduced representation of genome). We have now cross compared the data derived from the two methods and provides evidence that methylation divergence inferred at DMRs found genome-wide (between WGBS samples) is highly correlated with divergence in the RRBS/population level (see your question 2, page 1). We chose DMRs using the WGBS dataset as it offers a more genome-wide approach, enabling direct comparison with the genome-wide gene expression data. We believe that our methodical approach represents a strength of the paper and provides great in-depth into methylome divergence among three populations, unique in the field of evolutionary epigenetics. We have better discussed our methodological approach in the text and have discuss the extensive overlapping signals of methylation from the RRBS and WGBS approaches.

Page 5 of 22

Lines 112-118: Using genetic divergence result from a previous study is OK, but it would be better to directly extract SNPs and estimate genetic divergence from bisulfite sequencing data. Also, I do not agree that a distant distance between HDR and DMR suggests no association. It could be due to trans-acting genetic variants controlling methylation variation, so the statement between these lines are not well supported, especially considering the association between SNPs and DMRs were not specifically tested in this study.

We agree that our results cannot exclude trans-acting genetic variants on methylome variation and have therefore modified this sentence accordingly. DMRs and HDRs did not co-localise, although trans-acting effect of genetic variation on methylome divergence cannot be ruled out.

Lines 119-130: There is a flaw in the genomic feature analysis. The authors did not give a precedence when features were overlapped, so fig. 1i could be significantly biased due to repeatedly counting DMRs overlapping with multiple features.

We agree there is a need for clarification regarding our methodology here, however the analysis is not flawed. The categories 'Promoter', 'Gene' and 'Intergenic' are mutually exclusive (see revised Methods; DMRs that did not overlap [$\geq 50\%$ DMR sequencing overlap] with promoter/gene body were assigned as being intergenic). However, the categories CGIs and TE should be read differently and separately to each other as they are not mutually exclusive. CGI and TEs could overlap multiple genomic regions at once (for example a TE-derived CGI-containing promoter DMR). For the sake of clarity, we have split CGIs only into CGI-promoter and CGI-outside promoter. The fact that they are not mutually exclusive does not alter or bias the conclusion of this analysis. We have modified the figure to clearly show the categories that are mutually exclusive or not (dotted lines) and have mentioned this in the figure legend. The method section has been improved accordingly.

Lines 143-144: Implying adaptation directly from the genomic locations of DMRs is not suitable, especially without measuring fitness. I would suggest the authors to soften language here.

We agree with the reviewer and have rephrased the sentence, removing the term adaptation. This is the subject of future work. We have now mentioned this point in the text.

Fig. 2: In fig. 2a, One of the littoral fish is clustered with riverine fish, but there is no explanation for this. I would exclude this fish in downstream analysis. In addition, there are two different read lengths (100 vs. 150 bp PE) used in RNA-seq. Is there any batch effect due to read length? What is the correlation between samples from the same ecomorph but were sequenced in different read length? Finally, the color spectrum in fig. 2b is qualitative and not very informative, it would be good to show absolute or scaled FC instead of 'lowest'/'highest'.

We thank the reviewer for raising this point. Indeed, the RNA sequencing of wild-caught liver tissues was part of a collaborative effort to sequence RNAseq for multiple Lake Masoko cichlid tissues (unpublished yet) and resulted in the use of two sequencing read lengths. As pointed out by the reviewer, all benthic samples as well as one littoral sample were sequenced as paired-end (PE) 150bp reads; the remaining littoral and all the riverine livers were sequenced as PE100 reads.

We have reanalysed the RNAseq datasets to account for read length differences by trimming in silico all the PE150 samples to 100bp (all samples are 100bp paired end) and remapped all samples. Overall, RNAseq variation now clearly segregates populations, even in the case of the shallow/littoral sample (G08H10_33750_1.60; see graph below which is Extended Data Figure 6a, and also see Fig2a), suggesting that the extra trimming process could reduce potential confounding variation arising from different sequencing read lengths. We have added the right-hand side panel of the graph below ('Extra trimming - Heatmap of Spearman correlation scores') as Extended Data Figure 6a to inform readers on the different sequencing approaches, and have updated Figure 2a.

RNAseq divergence among the three populations. Unbiased hierarchical clustering of the overall RNAseq variation (Spearman's rank correlation rhos) using the original/pre-trimming samples (left) and post extra trimming of 150bp paired end reads to 100bp to account for different read lengths. The panel on the right is now shown as Extended Data Figure 6a.

Using extra-trimmed samples, we then reanalysed the RNAseq datasets and performed differential expression analysis again. Overall, the same genes were found to be differentially expressed (compared to the pre-trimming DEG analysis) between River vs Littoral; -7% ($n=179$ gene count differences) between RvB; and -3% ($n=22$) between LvsB. Overall, the overlap between DEGs old vs new datasets is 96%, with a total of 525 DEGs using trimming dataset. We have reanalysed or/and updated all the figures, including Fig2 and Extended Data Figures 6, 7, 9 (gene expression boxplot graphs, pvalues etc), main text and Methods to reflect those changes. In summary, this approach reduced technical variation due to different sequencing read length in the whole dataset and the main conclusion holds. Moreover, the gene count matrix file part of our GEO project has been replaced.

Regarding the colour spectrum of Fig. 2b, it represents a Z-score based on gene expression (scaled tpm per significantly differentially expressed gene) between each group, a common scaling system to compare RNA expression levels between groups. We have clarified this in the figure legend.

Lines 183-203: Do the genes, epoR, cHbB, Scl/Tal1, etc., have hypomethylated promoters? Because the authors emphasized the overrepresentation of DMRs in DEGs, I guess the examples in this paragraph are genes with decreased methylation in promoters but increased transcription activities, but I do not see any clear statement about such information. Also, it would be good to report or depict the correlation between methylation levels and transcription activities, either using genome-wide data or DMRs-DEG pairs.

We have reformulated this sentence to clarify the issue. Additionally, we have now performed detailed analysis on the correlation between methylation and transcriptional activity (see Question 3, page 2) and have discussed it in the main text.

Line 235: I do not understand the process to call 'reset DMRs' and 'fixed DMRs'. I believe some DMRs identified between riverine and benthic/littoral populations will have similar methylation level difference in common garden environment, which can be called as 'fixed', but how about methylated regions with decreased methylation difference between, say, benthic and riverine populations? Do you also consider them as 'fixed DMRs'? Finally, it seems that methylation levels also changed between wild and common garden environments within the same population, so even for some DMRs you called 'fixed' between populations, they can be 'reset' or 'plastic' within populations.

Fixed DMRs are defined as the DMRs found between any two wild populations and also show significant methylation differences between the same pairwise group within the common garden experiment. For the

example shown in Fig. 3b, DMRs found between wild benthic and littoral populations are considered fixed if they are still present between benthic and littoral common garden fish (consistently in all samples).

By contrast, “reset” DMRs are defined as DMRs found between any two wild populations and do not show significant methylation differences between the same pairwise group within the common garden experiment

There is indeed some methylome variation between wild and common garden experiment fish. We may expect this purely due to the different rearing environments. However, such changes do not influence our conclusion on potential inheritance/fixation of wild methylome variation, and are not directly studied here. We have aimed to clarify this in the text.

Extended Fig. 8: following my last comment, in wild samples, the variation among riverine samples is higher than littoral or benthic samples, but lower in common garden environments than littoral or benthic samples, suggesting a large number methylated CpGs were plastic or responsive to environmental transition, especially in riverine populations, which adds more complexity in defining reset and fixed DMRs. One possible way to identify transgenerationally stable CpGs is to compare methylation levels of F1 and F2 samples under common garden environment, similar to Heckwolf et al. 2020. Sci. Adv. and Hu et al. 2021. Genetics. And to identify reset/plastic and fixed DMRs, it should be better to reciprocally transplant wild samples to opposite environments, e.g., transplant riverine samples to benthic or littoral environment, and identify CpG sites with altered methylation levels that are explained by ecomorph (fixed) or environments (reset/plastic).

We agree there is some within-group methylome variation associated with common garden experiment, in particular with the riverine groups, although such DMRs would not be directly related to the differences we observed in the transition wild-to-common garden experiment and seems relatively small compared to the wild-to-common garden transition. Actually, the methylome of riverine fish seems to be the most robust to environmental perturbation (PC1 of Ext. Data Figure 8c shows the more substantive transition/epigenetic reprogramming of littoral and benthic fish, relative to riverine fish).

We agree that investigating methylome variation in F2's and subsequent generations, as well as the suggested transplant experiment and the assessment of epigenetic reprogramming in cichlids would be informative, however this must be the focus of future work. We have discussed such limitations in the revised manuscript and have delineated future lines of research.

Line 238: What is the biological meaning or explanations for reset DMRs having longer length than fixed DMRs? Are you suggesting longer length correlates with higher epimutation rate?

That the two DMR classes are characterised by a least one distinct property (length) implies there may be biological/functional differences associated with them. Further work will be required to characterise any differences. We do not make any suggestion of association between DMR class and epimutation rates. We have therefore reformulated the sentence.

Line 253: While I agree that, to some extent, the fixed DMRs represent population-specific methylation patterns, because only liver tissue was used in methylation analysis, it is hard to say such patterns are also tissue-independent.

We agree this sentence is ambiguous; we have therefore reformulated this part. In brief, we refer here to a previous study that showed that tissue-unspecific DMRs in some Malawi cichlids were found to be more enriched in genes associated with developmental processes (just like fixed DMRs), possibly underlying core phenotypic differences, rather than bearing tissue-specific (e.g., liver) functions. See Vernaz et al. (2021). Obviously, further methylome work would be required to confirm such hypothesis.

Line 355: Why sequencing WGBS samples in two different platforms? How did you incorporate and correct data from these two platforms?

Samples were sequenced on different platforms due to logistic reasons (discontinuation of the first machine used), however the technology remains the same (Illumina HiSeq4000 and NovaSeq). Only one wild shallow (wB_3) and one wild benthic (wL_1) (see PCA below; samples have been outlined) were sequenced with NovaSeq (on the same flow cell to minimise technical variation). PCA or hierarchical clustering analyses did not reveal technical biases for those samples (see Fig. 1f) and the same filtering QC were applied (minimum/max read coverages) to minimise sequencing biases. We have specifically mentioned and discussed the use of different platforms in Methods.

PCA of WGBS methylome variation using two sequencing techniques. Principal component analysis of WGBS samples. The two samples sequenced on NovaSeq have been outlined and cluster with other samples sequenced on HiSeq4000.

Line 369: CpGs with only at least 4 reads to be included in downstream analysis seem less favorable due to low coverage. Can you provide justifications for such low depth cut-off? Also, the coverage for genome needs to be provided for WGBS.

We required a minimum coverage at CG sites of at least 4 non-PCR clonal paired-end reads, which allows for reliable methylome analysis - such cut-off is common in methylome studies and enables reliable methylome analysis (Berrens et al. 2017; Skvortsova et al. 2019; Vernaz et al. 2021) and see review (Ziller et al. 2014). However, the average genome coverage at any CG sites genome-wide across all samples was much higher than the cut-off value (9.14 ± 1.4 , mean \pm sem). We have added a graph showing CG read coverage genome-wide for all samples genome-wide (see revised Extended Data Fig 2b) and have discussed it in the Methods.

Reviewer #2 (Remarks to the Author):

In the current study Vernaz et al present compelling evidence that DNA methylation might act as a major driver of vertebrate speciation. The study is a timely follow up of authors' recent work, which put forward the exciting hypothesis that DNA methylation participated in the phenotypic diversification of lake Malawi cichlid fish. The study is well-written, the analyses are properly executed and overall, my comments for improvements are minor.

We thank the reviewer for their positive comments and for highlighting the quality and timely importance of our work.

1) It is not clear from the current data how DMRs were associated to genes for the purpose of GO analyses. Did the authors employ the "nearest gene" approach or was more elaborate methodology used for this (such as GREAT for example)?

*We apologise for the lack of clarity. DMRs were assigned to genes when located in their promoter regions (TSS \pm 1 kbp; **intersection** with \geq 50% DMR sequence overlap), or gene body (both introns and exons; **intersection**; \geq 50% DMR sequence overlap). When DMRs did not overlap with promoters or gene bodies, they were labelled as 'intergenic regions'. Only intergenic DMRs located 1-5 kbp away from any given gene were assigned to the closest gene (closest gene approach using bedtools) GO analysis was then performed using genes associated with each category (unique genes only), using gProfiler. We have rewritten this section in the Methods to better explain how we assigned genes to DMRs and how GO analysis was performed.*

2) How many DNMTs / TETs were identified in cichlids? Given the enrichment of DMRs in CpG-rich regions, it would be interesting to explore if these differences could be caused by expression differences of DNA methylation machinery components?

We believe it would be difficult to link any new cichlid-specific copies of DNMT/TETs to the specific DNA methylation patterns we observe. Previous work in mammals have highlighted that duplication of certain DNMT3 enzymes (for example DNMT3C in Muridae rodents) could be associated with the targeted methylation of certain classes of transposable elements (TE), shedding light on an arms race between host DNMTs and TE sequence evolution in particular (see Barau et al. 2016). However, the methylome divergence we observed between Lake Masoko cichlids is not exclusively localised in one genomic feature (such as cichlid-specific TEs for example), therefore we do not believe this can result from novel DNMT/TET enzymes. From our previous work (Vernaz et al. 2021), Lake Malawi cichlids are believed to have one copy of each of the three Tet enzymes, one copy of DNMT1 and multiple copies of DNMT3s, just like in zebrafish (Campos, Valente, and Fernandes 2012; Serittrakul and Gross 2014; Shimoda et al. 2005; Smith, Collins, and McGowan 2011) - however we have not characterised or identified new DNMTs/TETs in Lake Masoko cichlids. This could be the subject of further work, but would require extensive RNAseq, proteomics analyses in different tissues, together with full genomic scan, in addition to lab experiments to elucidate the sequence-specificity of such proteins, which we believe is beyond the scope and the main hypothesis of this manuscript.

Of note, we plot below the gene expression levels of all DNMTs and TETs enzymes in Lake Malawi cichlids: none of them pass the minimal cut-off gene expression value (tpm 5) for actively transcribed genes (they all have a mean gene expression levels of 2 tpm) and therefore we can rule out any differentially expressed DNMT/TET enzymes between groups). We have included two other genes as a comparison (the house keeping genes GAPDH and the differentially expressed Haemoglobin cathodic subunit beta). Our DEG cut-offs, to robustly identify differentially expressed genes, are any genes with a gene expression difference between groups of \geq 10TPM, \geq 2 fold-change difference and $p < 0.05$

Gene expression profiles for all DNMTs and TETs predicted in Lake Malawi genomes, including GAPDH and Hb cath subunit beta. Boxplots showing expression levels for all samples of the three groups with individual sample points plotted. $n=4-5$ biological replicates (liver). The black horizontal line represents $tpm=5$ (the minimal cutoff value for active genes).

3) Also, TET proteins which actively remove DNA methylation are oxygenases and thus require oxygen for their catalytic activity. Could the difference in oxygen concentration therefore act as a driver for such epigenetic diversification? i.e less, oxygen  less TET activity  less capacity for 5mC removal  more hypermethylated regions?

This hypothesis is interesting. However, we believe our current work is not able to address this question and it is beyond the scope of this study. Nonetheless, it is important to note that we observe a widespread increase in methylation in both the shallow and benthic populations relative to the river population (see Fig. 1g). This is despite differences in environmental oxygen levels (the benthic habitat of the lake is almost anoxic; the shallow habitat remains well oxygenated; Fig. 1d). Therefore, blood oxygen levels might be sufficient in both ecomorphs for an efficient TET enzyme activity, and normal metabolism.

4) Instead of speculating about the possibility of fixation of epigenetic differences, could the authors perform RRBS / WGBS on sperm (or eggs, but more likely sperm given the current literature) on fish from their “garden experiment”? Given the strong enrichment for CpG-rich regions even RRBS might do. Such an experiment would demonstrate the potential for inter- or trans-generational inheritance of acquired methylation differences. Even if only a small subset of DMRs is picked up in sperm, this would be an extremely valuable piece of data.

We agree with the reviewer that our study lays the groundwork for further experiments, especially regarding the assessment of multigenerational inheritance of methylome patterns. We thank the reviewer for suggesting this interesting experiment. Our lab is currently initiating work on this aspect (F2 experiments for common garden fish, as well as the dynamics of epigenetic reprogramming/inheritance in cichlid), but such experiments represent a significant amount (several years) of work, and are beyond the scope of this current paper. We have nonetheless improved the discussion section about the inheritance of such patterns and the experiments required to assess it.

5) Reference 24 is wrong – the authors probably refer to Skvortsova et al, 2019 Nature Comms, rather than Skvortsova et al 2018 (Nat Rev Mol Cell Biol), where the inheritance of paternal methylome patterns in the zebrafish germline is not discussed.

We thank the reviewer for pointing this mistake out. The correct reference has now been used (Skvortsova et al 2018 Nat Comms).

Reviewer #3 (Remarks to the Author):

This work by Vernaz and colleagues assess variation in DNA methylation in benthic versus littoral populations of A. calliptera cichlids from river and Lake populations. The role of such epigenetic variation in an ongoing sympatric speciation event like this is largely unexplored. This work assesses overall patterns of methylation, potential relationship to gene expression, and plastic potential in a common garden experiment. While a strong work, there are a number of instances of missing logic of analyses or weak explanations that would strengthen conclusions. These are needed to bring this work from statistical connections in large-scale genomic analyses to the biological relevance and importance for evolutionary processes as pitched.

We are pleased that the reviewer recognises the novelty of our work. Following reviewer 3's comments, we have extensively re-written the manuscript to include all experimental details and clearly state the rationale behind each analysis/methodology. We believe the manuscript as well as the Methods section read much better now. Thank you for your feedback and suggestions.

Major comments:

A. Biological relevance of methylation differences

A1. The suggestion in the paper is that differences in metabolism due to methylation are driving benthic versus littoral adaptation. However, samples were taken from these 2 environments that are very different in terms of oxygen, etc, which will cause metabolism differences. Is it that methylation differences drive this habitat divergence or that methylation differences in the liver become distinct once animals diverge in habitat usage?

We thank the reviewer for raising this point. We agree that we could be clearer on when methylome divergence takes place and the relevance to speciation. We have now expanded and discussed this important point further in the revised manuscript (see Discussion).

Our hypothesis is that:

- I. methylation differences occur after the fish arrive in the specific habitats (e.g., deep water). They may have low initial fitness, which is not problematic at first given the abundance of food and lack of competition. After a period of acclimation due in part to DNA methylation affecting expression of key ecologically important genes, they are able to thrive. The positive traits are present in offspring, through inheritance, and through exposure to the same environment from young;*
- II. any new coloniser from the shallow environment will now be outcompeted in the deep water, and relative fitness will matter now due to competition. Equally, a fish of a deep-water origin will struggle in shallow water environment because they will not have been reared in such shallow environment, and/or possess the beneficial epialleles;*
- III. this process becomes amplified when genetic variants start to become fixed.*

A2. L63 notes that system is addressing sympatric speciation. However, the populations used do not live in sympatry, but are already distinct in their habitat usage.

*The exact definition of sympatric speciation is debated. To improve readability and avoid confusion, we have replaced any reference of "sympatric speciation" with "**ecological speciation**", which we believe better reflects the speciation process that takes place in Lake Masoko Astatotilapia ecomorphs.*

A3. The connection between differential methylation patterns and transcription factor binding was not clearly defined. Is the suggestion that differential methylation is happening in a way that only impacts certain TFs? How would this work biologically? Or is it that those genes active in the liver are enriched for those TF binding sites in their promoters? See also point C5.

We agree this connection should be clarified in the text. We have now discussed the complex relationship between differential methylation and TF activity in more detail. While it is well established that differential methylation at promoters of genes might impact the binding ability of TFs, since many TFs are methyl-sensitive, changes in TF activity might as well result in methylation changes at targeted gene promoters. In addition, SNPs in TF binding domain sequences or in binding motifs might influence focal methylation levels as well (Schübeler 2015; Yin et al. 2017; Zhu, Wang, and Qian 2016).

Page 12 of 22

Here we observed changes in promoter methylation associated with key ecologically relevant genes differentially expressed between populations, and enriched for certain TF binding sites. Interestingly, some of those TFs are differentially expressed between populations. Altogether, we postulate this reflects substantial differences in the activity of specific TFs, associated with transcriptional changes, which may be influenced by heritable methylation changes. We hypothesise that heritable methylation divergence might underlie change in TF activity and in transcriptional activity, however further experiments would be required to further decipher the exact mechanisms, as our current dataset does not allow for such conclusion. All those points are now discussed in the text, and the relevant literature has been added as well.

A4. I love the common garden experiment as it begins to address considerations such as that in point A1. However, details of this seem unclear. Were the conditions in this experiment similar to any of the natural environments? For example, were the tanks similar in oxygen and temperature of the littoral, but not benthic fish? How would this bias analysis or alter conclusions? How long was the experiment conducted, what is the logic of that choice, what is the developmental age of the fish at start and end of the experiment, and how does that affect the data obtained? Again, is it that you are catching all the regions of the genome that are responsive to environmental changes or those that are driving habitat changes and speciation? Some of the analysis in this section also highlights limitations of drawing conclusions from large-scale enrichment analyses. For example, GO analysis shows enrichment for fixed DMRs in developmental genes, but genes associated with neuron projection, neural tube development, and mastoid bone, none of which make sense given the tissue analyzed was liver. While interesting, the section about effect of fixed DMGs and brain development in L250-261 thus reads as examples picked from a list rather than making a functional connection between liver activity and brain development.

We thank the reviewer for the positive comment regarding the common garden experiment and for raising interesting points.

Regarding the methodology, we have now included all the requested experimental detail in the methods. In brief, to minimise confounding variation, all experiments involving common garden fish were performed in the same laboratory conditions (food, water oxygen levels, light/8 cycles, temperature), in parallel and for the same duration (~6 months post hatching). Fish were bred from wild caught parental specimens (collected and acclimatised to the same laboratory conditions at the same time), reared in tanks and euthanised at the same biological stages (i.e., adult males; ~6 months post hatching).

It is important to note that the laboratory rearing conditions are highly artificial. The aim of this experiment was to ensure a constant rearing environment to assess the inheritance and plasticity of epigenetic divergence across generations of ecomorphs (i.e., which sites are fixed/reset between wild and common garden fish). We can then identify genomic regions that are robust to environmental perturbations, such as the transition to laboratory conditions, and are likely 'fixed' in at least one population. Further experiments would be required to clearly delineate the regions that are 'responsive' to different types of environmental perturbations (temperature, diet, oxygen levels etc.) - this will inform on the extent to which each environmental condition might influence epigenetic landscape and inheritance thereof. Our labs are currently initiating such experiments, and we hope to have the answer in a few years from now. The methods section has been extensively re-written to include all the aforementioned experimental details. Furthermore, we have now discussed the limitations of our common garden setup in the main text.

With regard to the GO analysis, we have previously shown that fully differentiated organs, such as liver or muscle, can exhibit epigenetic states of early-life developmental stages, carried over across cell divisions (Vernaz et al. 2021) see also (Bogdanović et al. 2016). Approximately half of methylome divergence found between livers of Lake Malawi cichlid fishes has been associated with genes with functions related to embryonic and developmental processes (and are tissue-independent). While possibly functionally irrelevant in fully differentiated organs, such as liver, this suggests that methylome divergence can be associated with developmental processes, possibly participating in core phenotypic diversification, including tissue differentiation and development (Bogdanović et al. 2016; Vernaz et al. 2021). GO terms for fixed DMRs are similar (Fig.3c), in that they are associated with developmental/embryonic genes. We show some examples of fixed DMRs in Fig.3d too. We hypothesise that fixed DMRs might have functions not directly relevant to liver tissue, but rather to other tissues, in particular during development, participating in core phenotypic differences

between the populations, which might be relevant to ecomorph diversification. We have therefore shown in Fig.3c GO graph highlighting those terms. Reset DMRs, on the other hand, show GO terms relevant to liver metabolism. We have now moved the GO graph associated with reset DMR to Fig.3c. In addition, we have rewritten the main text to highlight our hypothesis about the potential functions of reset/fixated DMRs.

B. Choice of samples

B1. There was no explanation of why liver tissue was used for this analysis. What biases and limitations of conclusions does this introduce into the analysis?

*Reviewer 1 made the same point, which we have addressed above. The response was as follows:
"Regarding the choice of tissues: liver is a key organ involved in dietary metabolism, hormone production, and haematopoiesis. We know that the ecomorphs differ in their diet (Fig.1d and (Malinsky et al. 2015) and live in habitats substantially differing in oxygen levels (Fig.1b). Therefore, liver is a highly relevant organ to investigate the role of epigenetic in phenotypic diversification. Moreover, the liver is a largely homogenous organ, composed at >70% of hepatocytes (Aloia et al. 2019), which limits confounding variation associated with heterogenous cell populations when performing methylation studies. In the revision we have now explained the rationale behind the use of liver tissue."*

Although we expect that most methylome divergence will be directly relevant to the liver, some population-specific methylation variation could be associated to other biological processes and be relevant to other tissues (development) - see point A.4. We have reformulated this in the text, and have explained the rationale and limitation of using liver tissue in the Methods.

B2. Also not discussed is why males only were used. What is the logic of choosing males with bright nuptial coloring, and could this add any bias to your analysis or our understanding of evolutionary processes?

Reviewer 1 also brought up this point and our response reads as follows:

"There are three reasons we focus on males:

- 1. Practically, we are more confident in our assignment of male individuals to ecomorphs.*
- 2. In our sampling, to test our hypotheses, we aimed to maximise between-population variability, while minimising within-population variability, within the constraints of resources available.*
- 3. Epigenetic inheritance/reprogramming varies substantially among species (Potok et al. 2013; Skvortsova et al. 2019; Wang and Bhandari 2019), yet the nature of this in cichlid species was hitherto unknown. We surmised that if epigenetic inheritance is relevant to speciation, then it must be contributing to between-morph differences in traits possessed by both sexes (habitat occupancy, diet). Since we see differences in males, then the same may well apply to females, and this is a clear hypothesis that can be tackled with future work.*

We have clearly mentioned the rationale for our choice of samples to analyse in the revised manuscript and highlighted the limitations."

C. Analyses

C1. PCA of data sets is stated as having segregation in L78-79, but this is not supported with statistical analysis such as an ANOVA. Does hierarchical clustering of all regions, not just differentially with your specific cutoffs, also show segregation of populations?

We thank the reviewer for pointing this out. We have now performed statistical analysis on PCA related to population-level methylome variation between the three populations (Fig. 1e) and now show that the PC1 and PC2 scores significantly segregate the three populations apart (MANOVA, $p < 2.2e-16$ with all multiple comparison test adjusted p values $< 1e-3$; see Extended Data Figure 2c below). We have included this figure in the revised manuscript (Ext. Data Figure 2c). Furthermore, we have discussed this result in the main text and have added info related to the statistical tests performed.

Extended Data Figure 2c. Boxplot showing PC scores for PC1 and PC2 for all RRBS samples (from PCA, Fig. 1e). PC1 scores are significantly different among the three populations, significantly segregating the three populations apart. MANOVA, $p<2.2e-16$ with multiple comparison two-sided p-values adjusted with the Tukey's method are shown above each comparison

Regarding the second point, it is important to emphasise that both PCAs in Figure 1 (Fig. 1e, f) were performed in an unbiased manner, i.e., using all mapped data, all CG sites with enough read coverage (not only methylation variation at DMRs). Specifically, Fig. 1f shows the variance in methylome across all mapped CG sites genome-wide between all samples. We have clarified this point in the figure legend. Finally, we have chosen PCA plots over hierarchical clustering as PCA plot allows for a more exhaustive and less biased representation of the underlying variances.

C2. What is the logic of cutoffs used for analysis? For example, differentially methylated regions were called based on >25% mCG difference and >4 CpG sites. What difference would using 5 sites make in the conclusions? Why 4 sites? This is stated as "significantly" different in L93 and a p-value is given, but there is not information on how this pvalue is calculated in methods L368-370.

This is an important this point and we agree that additional information regarding our methodology would improve the manuscript.

Those cut-off parameters ($\geq 25\%$ methylation difference, ≥ 4 differentially methylated CG sites) were based on previous methylome studies and are recognised as being the optimal cut-offs to identify putative functional DMRs - i.e., cis regulatory elements bound by DNA-binding proteins and showing considerable methylation differences between ecomorphs with a typical size of a gene-regulatory region (Bock 2012; Deaton and Bird 2011; Wu et al. 2015). Even though the minimum cut-off values we used were ≥ 50 bp in length, $\geq 25\%$ of methylation difference, ≥ 4 CG sites covered, most DMRs covered more than 4 CG sites (median, 14-16 CG sites in all predicted DMR; see Extended Data Figure 3d [below]), showed $\sim 45\%$ methylation differences on average (Ext. Data Figure 3c) and were far larger than 50bp on average (median, 250bp; see Ext Fig Data 3d). We have now added and discussed the Figure below (now Ext Data Figure 3d) in the manuscript and have clearly discussed the choice of those cut-off values as well, citing the relevant literature (see revised Methods). Of note, only 4.7-5.6% of all DMRs predicted between the three populations had exactly 4 CG sites.

Extended Data Fig. 3d. Violin plot showing the number of CG sites covered by DMRs predicted between each pairwise comparison. Black horizontal bars indicate median values, red dots indicate mean values. y-axis, \log_{10} scale. This figure has now been added and discussed in the revised manuscript and Method section.

Regarding the statistical tests used to identify significant DMRs, Wald tests on methylation differences at all CG sites between any two groups were performed using DSS (Park and Wu 2016) - DMRs consisted of (≥ 4) adjacent CG sites showing significant methylation differences (Wald test's two-sided P -values ≤ 0.05 at any CG sites). The Methods section, as well as the main text (and figure legends), have been revised to include all information related to the statistical tests used.

C3. Assigning non-coding regions to gene function is a critical assumption that can bias downstream assessment of biological processes. What is the quality of the annotation of genes in the genome used for this (a critical factor for these analyses)? Why did you chose 1kb for a promoter versus 5kb? How were genes assigned to a specific gene for GO analysis?

First of all, the quality of the genome assembly used for this work (Lake Malawi mbuna cichlid, NCBI GCF_000238955.4) is of high quality, making use of PacBio long-read sequencing technology (Conte et al. 2019) and multi-tissue RNAseq data. Such a high quality, chromosomal genome assembly (almost unprecedented for teleost fish) is the result of intense genomic work carried out on Lake Malawi cichlid genomes over the last decades. In fact, Lake Malawi cichlid genome is comparable to the state-of-the-art genome assembly of the model organism zebrafish (*Danio rerio*; NCBI GCF_000002035.6); they are both high-quality chromosome-level assemblies (N50 of 1.4Mb in both cases). Second, gene annotation of Lake Malawi cichlid genome is also of high quality and comparable to zebrafish gene annotation. In total, 32,471 genes have been annotated in Lake Malawi cichlid genome, compared to 40,031 genes annotated in the latest zebrafish genome assembly.

As no functional annotation for the Lake Masoko genome currently exists (for example, histone marks profiles), we have defined genome annotations *in silico* following commonly used annotation parameters in the methylome and chromatin immunoprecipitation sequencing fields, and include TSS ± 1 kbp and gene body including the 1 kbp downstream of TSS (Hu et al. 2021; Li et al. 2019; Vernaz et al. 2021). The correlation between DNA methylation and gene expression activity is at its highest in regions ± 1 kbp around TSS (as highlighted in Extended Data Fig 5b but also see (Lister et al. 2009; Schlosberg, VanderKraats, and Edwards 2017)). The exclusion of the first 1 kbp of gene bodies was also required to avoid any overlap between genomic feature categories during the DMR localisation enrichment analysis (promoter, gene bodies and intergenic regions are therefore mutually exclusive categories).

And finally, DMRs were assigned to genes when located in their promoters (TSS ± 1 kbp) and within gene bodies (minus 1 kbp downstream TSS to avoid overlap). For intergenic DMRs, we also assigned them to genes if located 1-5 kbp away from any genes (closest-to-gene approach). We believe such stringent parameters offer the best functional associations, as long as experimental validation of regulatory elements is lacking. We have improved the Methods section to clearly define our choices of annotation, while noting the limitations arising from those. We have cited the relevant literature in the revised manuscript.

C4. Analysis for transcription factor binding motifs appears to only be conducted on DMRs in one particular region (outside the gene body). How does this compare to the groupings presented in Figure 1I? How do conclusions change when different genomic features are used? Is there a reason only 2 of 3 combinations are shown (river vs littoral is omitted in extended data Figure 3F)?

Our initial analysis was focused indeed on TF binding motif enriched in promoters and intergenic regions (ectopic promoter, enhancers) only, as the roles of methylation variation in such genomic regions are well known. However, gene bodies might also contain cis-regulatory elements.

We have now carried out enrichment analysis for TF binding motifs in gene bodies as well for the three pairwise comparisons. See revised Extended Data 5b. Overall, TFs whose binding motifs were enriched in DMRs were associated with the same biological processes irrespective of their genomic localisations (strong overlap of biological pathways associated with enriched TFs found between groups), suggesting that methylation variation might be associated with altered TF landscapes in the three genomic context. In addition and following the reviewer's comment, we have now also discussed in the text the close association between the roles of TFs whose binding motifs are enriched in DMRs and the functions of the genes showing DMR enrichment.

C5. What is the directionality of relationships between DMRs and DEGs? Given that DNA methylation has a predictable effect on gene expression, are those regions with high methylation associated with low gene expression? This is explored for *eklf* and *epoR* in L183-203. However, the *chbB* locus is exclusively expressed in benthic fish that hypermethylation, and not expressed in fish populations with low to no methylation. Thus, further explanation is needed to clarify clear how the biological function of methylation relates to gene expression. Without it, the text in L166-213 reads more like anything differentially expressed in the methylation data and transcriptional data were simply compared for overlap, regardless if the biology lines up.

This is an important point to clarify. Methylation at promoter regions generally negatively correlates with transcriptional activity, in line with a suppressive function of DNA methylation on gene expression. We have now included a graph to show this correlation (see Extended Figure 6b). However, this is not always true: it has been reported that some transcription factors might require methylated binding sequences to activate gene expression (Yin et al. 2017) - we have discussed this in the revised manuscript. In the case of Haemoglobin gene expression regulation, we have provided a graph showing significant methylome variation differences among populations at the LCR Major Control Locus, specifically benthic specific hypermethylation associated with increased gene expression - it is currently unclear how methylome variation at this site might influence, potentially through the binding of a methyl sensitive TFs. Further experiments would be required to test this hypothesis. We have now discussed this in the main text.

Minor comments:

D. Abbreviations. I encourage authors to critically evaluate if all abbreviations are necessary. Three specific examples, though I encourage a careful read for this throughout: (1) HDR and DMR get confusing for readers and are not always clearly defined in figure legends. (2) oCGI, pCGI, and LCR abbreviations are only used once, so there does not seem to be a utility to adding more abbreviations here. (3) GO categories MF, BP, CC, and KEGG in Figure 3 are never defined in the legend or the text.

We thank the reviewer for pointing this out. We have reviewed all the abbreviations used in the text. We agree that some are superfluous and have therefore been removed to increase readability - this include oCGI, pCGI, LCR and GO categories (MF, BP, CC). KEGG and HDR refer to specific, well recognised terms - we have made sure to clearly define such abbreviations in the revised version of the manuscript.

Reviewer #4 (Remarks to the Author):

In this paper, the authors have investigated divergence in DNA methylation pattern of the liver between littoral and benthic ecomorphs of an African crater lake cichlid fish. They found that these two ecomorphs diverge in DNA methylation patterns in several genes related to ecological adaptation and also report that some of the variations may be associated with gene expression variation. They also claim that the majority of the variations are reset during a common garden experiment, but some epigenetic variations are genetically determined.

I agree that the topics of this study, namely the role of DNA methylation in incipient speciation, is one of the important topics in evolutionary biology. However, I found several weaknesses in the paper.

We thank Reviewer 4 for reviewing our work and providing constructive feedback. We are pleased that the reviewer recognises the importance of the work.

First, they used only 2-3 wild-caught fish for the the whole genome bisulfite sequencing (WGBS). Although they used more individuals for reduced representation bisulfite sequencing (RRBS), the majority of the analysis are based on WGBS. I wonder whether they can validate this small sample size by comparing the results between WGBS and RRBS. Some of the RRBS sites should be included in WGBS data. I suggest that they should test how consistent the results of WGBS with N=2-3 and RRBS with N=11-12 are in such overlapping sites.

We thank the reviewer for raising this point. Reviewer 1 made a similar point, that we responded to in full above (Question 2, page 1).

In brief, and following Rev#4's suggestion, we have reanalysed both WGBS and RRBS datasets by performing an extensive cross-validation (see Extended Data Figure 4a,b). We now provide evidence that methylome variation in RRBS and WGBS samples at DMRs predicted using WGBS and overlapped with RRBS datasets highly correlate. This confirms that WGBS samples highly represent methylome variation at a population level. We believe this justifies the parallel use of both techniques, and the subsequent use of WGBS, offering whole-genome resolution, for all downstream analyses.

Second, I found no evidence for supporting that the DNA methylation differences they found are adaptive. I agree that DNA methylation differences in physiologically important genes are consistent with their claim, but they are not evidence for the adaptive roles of DNA methylation. Do the differentially methylated sites show any signatures of divergent selection?

Reviewer 1 made a similar point, and we agree with it. We cannot unequivocally conclude the methylome differences we observe are adaptive without performing fitness experiments, which would be beyond the scope of this study. We have modified the text to clarify this point, have removed any mention of a direct adaptive role for the DMRs found and have briefly mentioned the need to investigate any adaptive roles of such divergence.

Third, they found that some of the DNA methylation differences detected in the wild-caught fish were reset in the lab-raised fish. However, I am afraid that this may simply result from the chance effect due to the small sample size of the wild fish (i.e., morph-specific DNA methylation in the wild fish may be an artifact).

With respect, we do not agree with the reviewer's comment here. The suggestion that ecomorph-specific DNA methylation differences in wild fish are an artefact is not supported by the RRBS dataset, where genome-wide methylome variation clearly and statistically segregates populations apart (Fig.1e and Extended Data Figure 2c) - which is also supported by WGBS samples, that, we have shown above, is highly correlated with RRBS methylome variation (Ext. Data Fig. 4 and Supplementary Notes). This is the reason why we included both the WGBS and RRBS dataset (genome-wide and population levels).

Overall, I think that this paper tries to address an important topic, but the conclusions on the adaptive significance and the inheritance of DNA methylation are not very solid.

The adaptive significance is indeed not tested in our study, and we do not claim that it has been tested. For that, we would need to associate patterns of DNA methylation with differential fitness of individuals across different habitats. Here, we show (a) patterns of DNA methylation do differ between environments, (b) those differences are associated with gene expression and (c) there is evidence for cross-generational inheritance of epigenetic markers. Each of these patterns is known from other systems, but we are unaware of any other study that has included them collectively in a speciation context. Our evidence is collectively supportive of a role for epigenetic divergence in speciation, but we agree the direct association between epigenomic variation and fitness needs to be studied - we have mentioned this point in the revised manuscript and have extensively rewritten the manuscript to clearly highlight that our research now lays the groundwork for further work that will elucidate the adaptive role of such divergence as well as the mechanism of DNA methylation reprogramming in cichlid fish.

Additionally, given the fact that there are already several papers reporting DNA methylation differences between young species or ecomorphs in darter fishes (eg. Smith et al. 2016 Mol Ecol) and stickleback fishes (Hu et al 2021 Genetics), I have difficulty in finding novelty of the work.

There are striking differences between our work and that highlighted by the reviewer. The darter study used MSAP loci (a rather antiquated and imprecise method) and did not consider the variation in relation to gene expression or inheritance. It focussed on allopatric methylome variation in one species (which might have nothing to do with speciation) and variation among long-diverged species pairs (3-18Mya).

The stickleback study focussed exclusively on RRBS data. It showed evidence of divergence of allopatric marine and freshwater ecotypes, and cross-generation inheritance. Thus, the results support those analyses support those of our study. However, the stickleback study does not quantify variation at the genome scale, or relate methylome variation to gene expression. Notably, the study suggests epigenetic differences among populations across a geographic distance of 150 kilometres, while our study - remarkably - suggests divergence across only 20 metres!

We have now cited and discussed this work in more detail in the revised version of the manuscript.

Minor comments

How did you treat the sex? Is there any difference in sex difference in DNA methylation and gene expression? Sometimes, the sex explains the largest amount of gene expression variation.

Reviewer 1 made the same point, that we have addressed above. Please refer to Question 4, page 3.

L181-182: Is there any correlation between the direction of DNA methylation and gene expression (i.e., high methylation is associated with low expression etc)?

Two other reviewers have discussed this point as well, rightfully. Following this comment, we have now included a graph showing the correlation between methylation over promoters/gene bodies and gene expression activity (see below, Ext. Data Fig. 6b). As previously shown in cichlid fishes (Vernaz et al. 2021) as well as in other vertebrates (Schübeler 2015; Zemach et al. 2010), we could confirm that methylation at promoter regions (in particular, TSS \pm 1kbp) was negatively correlated with gene expression activity overall (Spearman's rank correlation, $\rho = -0.31$, $p < 2.2 \times 10^{-16}$), consistent with an overall repressive role for DNA methylation on transcriptional activity in vertebrates - although this is not always the case: certain classes of transcription factors bind methylated promoters as well for example (Yin et al. 2017). Gene body methylation, on the other hand, was not correlated with transcription activity overall. The function of gene body methylation remains poorly understood, and could be involved in alternative splicing (Lev Maor, Yearim, and Ast 2015). We have now discussed this result in the revised manuscript, have added this figure as Extended Data Figure 5b and have added the methodology to the Methods section.

Extended Data Figure 6b. Correlation between DNA methylation levels (mCG/CG) over promoter/gene body and gene expression activity. Genes were split into 5 categories according to their transcription activity: from non-expressed ('OFF'), to 'ON' genes showing very low (1), low (2), high (3) and very high (4) levels of expression (similar amount of genes per ON category). Methylation at promoter regions (TSS) is negatively correlated with gene expression, while no correlation was robustly detected for gene body methylation. Rho and P values for Spearman's rank correlation tests are shown (bottom left). Data are for *A. calliptera* Benthic fish and are representative of the other populations. WGBS data, n=3 biological samples (averaged mCG/CG); RNAseq, n=5 biological replicates (averaged tpm per gene).

L408: Is it common to exclude the 1 kb downstream of TSS for defining the gene body?

As any functional annotation for the Lake Masoko genome is currently lacking (for example, histone marks profiles), we have defined genome annotations in silico following commonly used annotation parameters in the methylome and chromatin immunoprecipitation sequencing fields, and include TSS ± 1 kbp and gene body including the 1 kbp downstream of TSS (Hu et al. 2021; Li et al. 2019; Vernaz et al. 2021). The correlation between DNA methylation and gene expression activity is at its highest in regions ± 1 kbp around TSS (as highlighted in Fig R but also see (Lister et al. 2009; Schlosberg et al. 2017)). The exclusion of the first 1 kbp of gene bodies was also required to avoid any overlap between genomic feature categories during the DMR localisation enrichment analysis (promoter, gene bodies and intergenic regions are therefore mutually exclusive categories).

We nonetheless repeated the DMR enrichment analysis using a narrower genomic window for promoter regions (TSS +1 kbp and -500) and could not find any significant differences in DMR enrichment for promoter or gene body regions (see Table below), suggesting that DMRs tend to localise within the first 1 kbp upstream and 500bp downstream of TSS. We have now better explained the clarified our definition of the in silico annotation.

Enrichment TSS ± 1 kbp				Enrichment (TSS +1kbp -500bp)			
	RvL	RvB	LvB		RvL	RvB	LvB
pCGI	1.2	1.8	5.6	pCGI	1.2	1.8	5.6
oCGI	1.3	1.8	4.2	oCGI	1.3	1.8	4.2
TSS ± 1 kbp	2.2	2.6	2.7	TSS +1kbp -500bp	2.3	2.7	2.8
Intergenic	0.6	0.7	1.4	Intergenic	0.6	0.7	1.4
Gene (-1kbp)	1.2	1.0	0.4	Gene (-500bp)	1.2	1.1	0.5
Repeats	0.7	0.7	1.2	Repeats	0.7	0.7	1.2

Table showing DMR enrichment in different genomic features using two different promoter annotation parameters (TSS ± 1 kbp [left] and TSS +1 kbp -500bp). Values represent Observed/Expected ratios for DMR localisations in CpG islands in and outside promoter (pCGI, oCGI, respectively), promoter regions (TSS), intergenic regions, gene bodies and repeats. Promoter, gene body and intergenic regions are mutually exclusive. Differences in O/E are highlighted in pink.

References

- Aloia, Luigi, Mikel Alexander McKie, Grégoire Vernaz, Lucía Cordero-Espinoza, Niya Aleksieva, Jelle van den Aemele, Francesco Antonica, Berta Font-Cunill, Alexander Raven, Riccardo Aiese Cigliano, German Belenguer, Richard L. Mort, Andrea H. Brand, Magdalena Zernicka-Goetz, Stuart J. Forbes, Eric A. Miska, and Merixell Huch. 2019. "Epigenetic Remodelling Licences Adult Cholangiocytes for Organoid Formation and Liver Regeneration." *Nature Cell Biology* 21(11):1321–33.
- Anastasiadi, Dafni, Anna Esteve-Codina, and Francesco Piferrer. 2018. "Consistent Inverse Correlation between DNA Methylation of the First Intron and Gene Expression across Tissues and Species." *Epigenetics and Chromatin* 11(1):1–17.
- Barau, Joan, Aurélie Teissandier, Natasha Zamudio, Stéphanie Roy, Valérie Nalesso, Yann Hérault, Florian Guillou, and Déborah Bourhis. 2016. "The DNA Methyltransferase DNMT3C Protects Male Germ Cells from Transposon Activity." *Science* 354(6314):909–12.
- Berrens, Rebecca V., Simon Andrews, Dominik Spensberger, Fátima Santos, Wendy Dean, Poppy Gould, Jafar Sharif, Nelly Olova, Tamir Chandra, Haruhiko Koseki, Ferdinand von Meyenn, and Wolf Reik. 2017. "An EndosRNA-Based Repression Mechanism Counteracts Transposon Activation during Global DNA Demethylation in Embryonic Stem Cells." *Cell Stem Cell* 21(5):694–703.e7.
- Book, Christoph. 2012. "Analysing and Interpreting DNA Methylation Data." *Nature Reviews. Genetics* 13(10):705–19.
- Bogdanović, Ozren, Arne H. Smits, Elisa de la Calle Mustienes, Juan J. Tena, Ethan Ford, Ruth Williams, Upeka Senanayake, Matthew D. Schultz, Saartje Hontelez, Ila van Kruijsbergen, Teresa Rayon, Felix Gnerlich, Thomas Carell, Gert Jan C. Veenstra, Miguel Manzanares, Tatjana Sauka-Spengler, Joseph R. Ecker, Michiel Vermeulen, José Luis Gómez-Skarmeta, and Ryan Lister. 2016. "Active DNA Demethylation at Enhancers during the Vertebrate Phylogenic Period." *Nature Genetics* 48(4):417–26.
- Campos, Catarina, Luisa M. P. Valente, and Jorge M. O. Fernandes. 2012. "Molecular Evolution of Zebrafish Dnmt3 Genes and Thermal Plasticity of Their Expression during Embryonic Development." *Gene* 500(1):93–100.
- Chen, Haifeng, Andrew D. Smith, and Ting Chen. 2016. "WALT: Fast and Accurate Read Mapping for Bisulfite Sequencing." *Bioinformatics* 32(22):3507–9.
- Conte, Matthew A., Rajesh Joshi, Emily C. Moore, Sri Pratima Nandamuri, William J. Gammerding, Reade B. Roberts, Karen L. Carleton, Sigbjørn Lien, and Thomas D. Kocher. 2019. "Chromosome-Scale Assemblies Reveal the Structural Evolution of African Cichlid Genomes." *GigaScience* 8(4):1–20.
- Deaton, Aimée M. and Adrian Bird. 2011. "CpG Islands and the Regulation of Transcription." *Genes & Development* 25(10):1010–22.
- Hu, Juntao, Sara J. S. Wuitthik, Tegan N. Barry, Heather A. Jamniczky, Sean M. Rogers, and Rowan D. H. Barrett. 2021. "Heritability of DNA Methylation in Threespine Stickleback (*Gasterosteus aculeatus*)" edited by C. Peichel. *Genetics* 217(1):1–15.
- Krueger, Felix and Simon R. Andrews. 2011. "Bismark: A Flexible Aligner and Methylation Caller for Bisulfite-Seq Applications." *Bioinformatics* 27(11):1571–72.
- Lee, Hyung Joo, Yiran Hou, Yujie Chen, Zea Z. Dailey, Aiyana Riddihough, Hyo Sik Jang, Ting Wang, and Stephen L. Johnson. 2020. "Regenerating Zebrafish Fin Epigenome Is Characterized by Stable Lineage-Specific DNA Methylation and Dynamic Chromatin Accessibility." *Genome Biology* 21(1):1–17.
- Levy Maor, Galit, Ahuvi Yearim, and Gil Ast. 2015. "The Alternative Role of DNA Methylation in Splicing Regulation." *Trends in Genetics* 31(5):274–80.
- Li, Jin, Yan Li, Wei Li, Huaibing Luo, Yanping Xi, Shihua Dong, Ming Gao, Peng Xu, Baolong Zhang, Ying Liang, Qingping Zou, Xin Hu, Lina Peng, Dan Zou, Ting Wang, Hongbo Yang, Cizhong Jiang, Shaoliang Peng, Feizhen Wu, and Wenqiang Yu. 2019. "Guide Positioning Sequencing Identifies Aberrant DNA Methylation Patterns That Alter Cell Identity and Tumor-Immune Surveillance Networks." *Genome Research* 29(2):270–80.
- Lister, Ryan, Mattia Pelizzola, Robert H. Dowen, R. David Hawkins, Gary Hon, Julian Tonti-Filippini, Joseph R. Nery, Leonard Lee, Zhen Ye, Que-Minh Ngo, Lee Edsall, Jessica Antosiewicz-Bourget, Ron Stewart, Victor Ruotti, A. Harvey Millar, James A. Thomson, Bing Ren, and Joseph R. Ecker. 2009. "Human DNA Methylomes at Base Resolution Show Widespread Epigenomic Differences." *Nature* 462(7271):315–22.
- Malinsky, Milan, Richard J. Challis, Alexandra M. Tyers, Stephan Schiffels, Yohey Terai, Benjamin P. Ngatunga, Eric A. Miska, Richard Durbin, Martin J. Genner, and George F. Turner. 2015. "Genomic Islands of Speciation Separate Cichlid Ecomorphs in an East African Crater Lake." *Science* 350(6267):1493–98.
- Munby, Hannah, Tyler Phillip Linderoth, Bettina Fischer, Mingliu Du, Grégoire Vernaz, Alexandra M. Tyers, Benjamin P. Ngatunga, Asilatu H. Shechonge, Hubert Denise, Shane A. McCarthy, Iliana Bista, Eric A. Miska, M. Emilia Santos, Martin J. Genner, George F. Turner, and Richard Durbin. 2021. "Differential Use of Multiple Genetic Sex Determination Systems in Divergent Ecomorphs of an African Crater Lake Cichlid." *BioRxiv* (doi.org/10.1101/2021.08.05.455235):1–46.
- O'Brien, Diane M. 2015. "Stable Isotope Ratios as Biomarkers of Diet for Health Research." *Annual Review of Nutrition* 35(1):565–94.
- Park, Yongseok and Hao Wu. 2016. "Differential Methylation Analysis for BS-Seq Data under General Experimental Design." *Bioinformatics* 32(10):1446–53.
- Potok, Magdalena E., David A. Nix, Timothy J. Parnell, and Bradley R. Cairns. 2013. "Reprogramming the Maternal Zebrafish Genome after Fertilization to Match the Paternal Methylation Pattern." *Cell* 153(4):759–72.
- Schlosberg, Christopher E., Nathan D. VanderKraats, and John R. Edwards. 2017. "Modeling Complex Patterns of Differential DNA Methylation That Associate with Gene Expression Changes." *Nucleic Acids Research* 45(9):5100–5111.
- Schübeler, Dirk. 2015. "Function and Information Content of DNA Methylation." *Nature* 517(7534):321–26.
- Seritrukul, Pawat and Jeffrey M. Gross. 2014. "Expression of the de Novo DNA Methyltransferases (Dnmt3 - Dnmt8) during Zebrafish Lens Development." *Developmental Dynamics* 243(2):350–56.
- Shimoda, Nobuyoshi, Kimi Yamakoshi, Akimitsu Miyake, and Hiroyuki Takeda. 2005. "Identification of a Gene Required for de Novo DNA Methylation of the Zebrafish No Tail Gene." *Developmental Dynamics* 233(4):1509–16.
- Skvortsova, Ksenia, Katsiaryna Tarbashevich, Martin Stehling, Ryan Lister, Manuel Irimia, Erez Raz, and

- Ozren Bogdanovic. 2019. "Retention of Paternal DNA Methylome in the Developing Zebrafish Germline." *Nature Communications* 10(1):3054.
- Smith, Tamara H. L., Terry Mark Collins, and Ross A. McGowan. 2011. "Expression of the Dnmt3 Genes in Zebrafish Development: Similarity to Dnmt3a and Dnmt3b." *Development Genes and Evolution* 220(11-12):347-53.
- Tran, Hong, Jacob Porter, Ming-an Sun, Hehuang Xie, and Liqing Zhang. 2014. "Objective and Comprehensive Evaluation of Bisulfite Short Read Mapping Tools." *Advances in Bioinformatics* 2014:1-11.
- Vernaz, Grégoire, Milan Malinsky, Hannes Svoldal, Mingliu Du, Alexandra M. Tyers, M. Emilia Santos, Richard Durbin, Martin J. Genner, George F. Turner, and Eric A. Miska. 2021. "Mapping Epigenetic Divergence in the Massive Radiation of Lake Malawi Cichlid Fishes." *Nature Communications* 12(1):5870.
- Wang, Xuegeng and Ramji Kumar Bhandari. 2019. "DNA Methylation Dynamics during Epigenetic Reprogramming of Medaka Embryo." *Epigenetics* 14(6):611-22.
- Wu, Hao, Tianlei Xu, Hao Feng, Li Chen, Ben Li, Bing Yao, Zhaohui Qin, Peng Jin, and Karen N. Conneely. 2015. "Detection of Differentially Methylated Regions from Whole-Genome Bisulfite Sequencing Data without Replicates." *Nucleic Acids Research* 43(21):gkv715.
- Yin, Yimeng, Ekaterina Morgunova, Arttu Jolma, Eevi Kaasinen, Biswajyoti Sahu, Syed Khund-Sayeed, Pratyush K. Das, Teemu Kivioja, Kashyap Dave, Fan Zhong, Kazuhiro R. Nitta, Minna Taipale, Alexander Popov, Paul A. Ginno, Silvia Domcke, Jian Yan, Dirk Schübeler, Charles Vinson, and Jussi Taipale. 2017. "Impact of Cytosine Methylation on DNA Binding Specificities of Human Transcription Factors." *Science* 356(6337):eaaj2239.
- Zemach, Assaf, Ivy E. McDaniel, Pedro Silva, and Daniel Zilberman. 2010. "Genome-Wide Evolutionary Analysis of Eukaryotic DNA Methylation." *Science* 328(5980):916-19.
- Zhu, Heng, Guohua Wang, and Jiang Qian. 2016. "Transcription Factors as Readers and Effectors of DNA Methylation." *Nature Reviews Genetics* 17(9):551-65.
- Ziller, Michael J., Kasper D. Hansen, Alexander Meissner, and Martin J. Aryee. 2014. "Coverage Recommendations for Methylation Analysis by Whole-Genome Bisulfite Sequencing." *Nature Methods* 12(3):230-32.

Decision Letter, second revision:

25th April 2022

Dear Professor Miska,

Your manuscript entitled "Epigenetic Divergence during Early Stages of Speciation in an African Crater Lake Cichlid Fish" has now been seen again by our 3 reviewers, whose comments are attached. While two of the reviewers are now fully satisfied, Reviewer 1 has a couple of further concerns that need addressing before we can reach a final decision regarding publication.

We therefore invite you to revise your manuscript again. Please highlight all changes in the manuscript text file.

* If you have not done so already please begin to revise your manuscript so that it conforms to our Article format instructions at <http://www.nature.com/natecolevol/info/final-submission>. Refer also to any guidelines provided in this letter.

[REDACTED]

Nature Ecology & Evolution is committed to improving transparency in authorship. As part of our efforts in this direction, we are now requesting that all authors identified as 'corresponding author' on published papers create and link their Open Researcher and Contributor Identifier (ORCID) with their account on the Manuscript Tracking System (MTS), prior to acceptance. ORCID helps the scientific

community achieve unambiguous attribution of all scholarly contributions. You can create and link your ORCID from the home page of the MTS by clicking on 'Modify my Springer Nature account'. For more information please visit www.springernature.com/orcid.

[REDACTED]

Reviewers' comments:

Reviewer #1 (Remarks to the Author):

This is the second time I read this MS, and I find it greatly improved after revision. I only have two more comments below that I would like to see the responses from authors before the publication of this MS.

1. While the authors have demonstrated a correlation between WGBS and RRBS by quantifying methylation levels at DMRs (Extended Data Figure 4), properly justifying the choice of proceeding with WGBS for downstream analysis, the small sample size in WGBS will intrinsically bring higher methylation variance and result in larger number of DMRs than RRBS. In addition, methylation levels of DMRs derived from WGBS are sometime quite difference from the same DMRs derived from RRBS in panel B of the Extended Data Figure 4. I may miss it, but I think the caveat of using small sample size should be noted. Also, the overlap vs. non-overlap of the number of regions identified as DMRs between WGBS and RRBS, i.e., how many DMRs out of the total DMRs identified using WGBS are also called as DMRs using RRBS, and vice versa, could be reported in addition to only cross-validating the methylation levels at DMRs shared by the two techniques in Extended Data Figure 4A.

2. For fixed DMRs, it seems the current definition only considers the significance of methylation change in pairwise comparison, but misses the direction of methylation change. For example, it is possible that a DMR can be significantly hypermethylated in littoral fish relative to benthic fish in wild populations, and at the same time significantly hypomethylated in littoral fish relative to benthic fish in common garden populations. Such DMR fits into the current definition of fixed DMRs but apparently does not reflect evolutionary divergence between ecomorphs. It might be good to double check the direction of methylation change, and only select those had concordant direction of change.

Reviewer #2 (Remarks to the Author):

The authors did an excellent job in answering the Reviewers' comments and in my opinion the manuscript is ready for publication. This is a very interesting study that provides further evidence that epigenetics and speciation might be linked. Moreover, the study lays important groundwork for future multigenerational epigenome inheritance studies in cichlids that could help to further our understanding of the function of epigenetic inheritance in vertebrates.

Reviewer #4 (Remarks to the Author):

The authors made suitable revisions in response to my previous comments. I recommend publication of this revised paper. I think that this is one of the most comprehensive DNA methylation studies in the context of incipient speciation.

*****END*****

Author Rebuttal, second revision:

Point-by-point response to reviewers - NATECOLEVOL-210814388C

Authors' response:

We would like to thank the reviewers for their time to review our work and for highlighting the quality of the revised manuscript. Following reviewer's 1 comments, we have further revised our analysis, in particular regarding the directionality of methylation in our inheritance experiment, which we believe has greatly strengthened the main conclusions of our work.

Reviewer #1 (Remarks to the Author):

This is the second time I read this MS, and I find it greatly improved after revision. I only have two more comments below that I would like to see the responses from authors before the publication of this MS.

We are pleased that reviewer 1 finds the revised manuscript much improved. We thank the reviewer for the comments.

1. While the authors have demonstrated a correlation between WGBS and RRBS by quantifying methylation levels at DMRs (Extended Data Figure 4), properly justifying the choice of proceeding with WGBS for downstream analysis, the small sample size in WGBS will intrinsically bring higher methylation variance and result in larger number of DMRs than RRBS. In addition, methylation levels of DMRs derived from WGBS are sometime quite difference from the same DMRs derived from RRBS in panel B of the Extended Data Figure 4. I may miss it, but I think the caveat of using small sample size should be noted. Also, the overlap vs. non-overlap of the number of regions identified as DMRs between WGBS and RRBS, i.e., how many DMRs out of the total DMRs identified using WGBS are also called as DMRs using RRBS, and vice versa, could be reported in addition to only cross-validating the methylation levels at DMRs shared by the two techniques in Extended Data Figure 4A.

Following reviewer's 1 comments, we have now performed DMR analysis between the three populations using RRBS samples. In total, we identified 333, 534 and 342 DMRs between River-Littoral, River-Benthic and Littoral-Benthic fish, respectively. This compares with 5244, 4594 and 341 for the same populations using the WGBS samples.

The higher number of DMRs found using the WGBS dataset will in part stem from the whole-genome, single CG site approach enabled by WGBS - as opposed to the reduced genomic representation offered by the RRBS technique. Additionally, we agree that the lower sample size of the WGBS dataset could increase the number of DMRs due to higher within-population variation. We have now mentioned these two issues in 'Supplementary Notes'.

Finally, we have now performed a direct comparison between DMRs found using RRBS and WGBS datasets (see graph below, which is Extended Data Fig. 4c-d) and found that between 4.4 and 23.1% of the DMRs identified among RRBS samples were identified using WGBS, highlighting the overlapping yet contrasting sequencing approaches of WGBS and RRBS. In summary, the methylation variation derived from both techniques reveals high population segregation (Fig. 1). However, a direct comparison is somehow complicated given the difference in sample sizes, in genome resolution, and our stringent statistical parameters. Nonetheless, we have shown that methylation at epigenetically divergent loci inferred using WGBS were highly recapitulated in RRBS samples (when overlapping).

D

	RRBS	WGBS	Overlap with WGBS-DMRs	Overlap %
RvL	333	5,244	77	23.1%
RvB	534	4,594	118	22.1%
LvB	342	341	15	4.4%

2. For fixed DMRs, it seems the current definition only considers the significance of methylation change in pairwise comparison, but misses the direction of methylation change. For example, it is possible that a DMR can be significantly hypermethylated in littoral fish relative to benthic fish in wild populations, and at the same time significantly hypomethylated in littoral fish relative to benthic fish in common garden populations. Such DMR fits into the current definition of fixed DMRs but apparently does not reflect evolutionary divergence between ecomorphs. It might be good to double check the direction of methylation change, and only select those had concordant direction of change.

We agree with this excellent point. We have now revised our definition of fixed DMRs to account for methylation directionality and have revised our methodology to include this change. Overall, we found that only a small fraction of fixed DMRs did not fit the current definition of fixed DMRs (a reduction of ~1% of total number of fixed DMRs across all comparisons; see Extended Data Figure 9a and the table below for a summary). We have rerun all the analyses related to this part to account for this revised definition (i.e., Fig. 3 and Extended Data Figure 9) and have updated all the associated tables and figures, as well as the main text and Methods accordingly. The overall results remain unchanged, and we believe this analysis has greatly strengthened our conclusions.

	Total DMR	Rev_1				Rev_2 mC Directionality accounted for				Directionality incoherence
		n_fixed	n_reset	p_fixed	p_reset	n_fixed	n_reset	p_fixed	p_reset	p_diff
River Benthic	4594	635	3959	13.8%	86.2%	611	3983	13.3%	86.7%	-0.5%
River Littoral	5244	448	4796	8.5%	91.5%	417	4827	8.0%	92.0%	-0.6%
Littoral Benthic	341	50	291	14.7%	85.3%	42	299	12.3%	87.7%	-2.3%

Re-analysis of fixed/reset DMRs to account for methylation directionality between DMRs found in wild vs. common-garden fish. Table showing the total count (n) and percentage (p) of fixed/reset DMRs found between each respective pairwise using the new analysis presented in the second manuscript revision and accounting for methylation directionality (green), compared to the previous analysis (blue). The column in red shows a summary of the difference in fixed DMR count found using the new pipeline. See Extended Data Figure 9a.

Reviewer #2 (Remarks to the Author):

The authors did an excellent job in answering the Reviewers' comments and in my opinion the manuscript is ready for publication. This is a very interesting study that provides further evidence that epigenetics and speciation might be linked. Moreover, the study lays important groundwork for future multigenerational epigenome inheritance studies in cichlids that could help to further our understanding of the function of epigenetic inheritance in vertebrates.

We thank the reviewer for the enthusiastic comments and for the constructive review.

Reviewer #4 (Remarks to the Author):

The authors made suitable revisions in response to my previous comments. I recommend publication of this revised paper. I think that this is one of the most comprehensive DNA methylation studies in the context of incipient speciation.

We thank the reviewer for the comments and time spent reviewing our work.

Decision Letter, third revision:

8th June 2022

Dear Dr. Miska,

Thank you for submitting your revised manuscript "Epigenetic Divergence during Early Stages of Speciation in an African Crater Lake Cichlid Fish" (NATECOLEVOL-210814388C). It has now been seen again by the original reviewers and their comments are below. The reviewers find that the paper has improved in revision, and therefore we'll be happy in principle to publish it in Nature Ecology & Evolution, pending minor revisions to satisfy the reviewers' final requests and to comply with our editorial and formatting guidelines.

[REDACTED]

Reviewer #1 (Remarks to the Author):

The authors have well addressed my last comments. I do not have further comments, and recommend publication of the current version of manuscript. This is a very interesting and timing study that demonstrates the role of epigenetics in speciation.

Our ref: NATECOLEVOL-210814388C

4th July 2022

Dear Dr. Miska,

Thank you for your patience as we've prepared the guidelines for final submission of your Nature Ecology & Evolution manuscript, "Epigenetic Divergence during Early Stages of Speciation in an African Crater Lake Cichlid Fish" (NATECOLEVOL-210814388C). Please carefully follow the step-by-step instructions provided in the attached file, and add a response in each row of the table to indicate the changes that you have made. Please also check and comment on any additional marked-up edits we have proposed within the text. Ensuring that each point is addressed will help to ensure that your revised manuscript can be swiftly handed over to our production team.

****We would like to start working on your revised paper, with all of the requested files and forms, as soon as possible (preferably within two weeks). Please get in contact with us immediately if you anticipate it taking more than two weeks to submit these revised files.****

In recognition of the time and expertise our reviewers provide to Nature Ecology & Evolution's editorial process, we would like to formally acknowledge their contribution to the external peer review of your manuscript entitled "Epigenetic Divergence during Early Stages of Speciation in an African Crater Lake Cichlid Fish". For those reviewers who give their assent, we will be publishing their names alongside the published article.

Nature Ecology & Evolution offers a Transparent Peer Review option for new original research manuscripts submitted after December 1st, 2019. As part of this initiative, we encourage our authors to support increased transparency into the peer review process by agreeing to have the reviewer comments, author rebuttal letters, and editorial decision letters published as a Supplementary item. When you submit your final files please clearly state in your cover letter whether or not you would like to participate in this initiative. Please note that failure to state your preference will result in delays in accepting your manuscript for publication.

Cover suggestions

As you prepare your final files we encourage you to consider whether you have any images or illustrations that may be appropriate for use on the cover of Nature Ecology & Evolution.

Nature Ecology & Evolution has now transitioned to a unified Rights Collection system which will allow our Author Services team to quickly and easily collect the rights and permissions required to publish your work. Approximately 10 days after your paper is formally accepted, you will receive an email in providing you with a link to complete the grant of rights. If your paper is eligible for Open Access, our Author Services team will also be in touch regarding any additional information that may be required to arrange payment for your article.

Please note that *Nature Ecology & Evolution* is a Transformative Journal (TJ). Authors may

publish their research with us through the traditional subscription access route or make their paper immediately open access through payment of an article-processing charge (APC). Authors will not be required to make a final decision about access to their article until it has been accepted. [Find out more about Transformative Journals](https://www.springernature.com/gp/open-research/transformative-journals)

Authors may need to take specific actions to achieve [compliance with funder and institutional open access mandates](https://www.springernature.com/gp/open-research/funding/policy-compliance-faqs). If your research is supported by a funder that requires immediate open access (e.g. according to [Plan S principles](https://www.springernature.com/gp/open-research/plan-s-compliance)) then you should select the gold OA route, and we will direct you to the compliant route where possible. For authors selecting the subscription publication route, the journal's standard licensing terms will need to be accepted, including [self-archiving-and-license-to-publish](https://www.nature.com/nature-portfolio/editorial-policies/self-archiving-and-license-to-publish). Those licensing terms will supersede any other terms that the author or any third party may assert apply to any version of the manuscript.

[REDACTED]

[REDACTED]

Reviewer #1:

Remarks to the Author:

The authors have well addressed my last comments. I do not have further comments, and recommend publication of the current version of manuscript. This is a very interesting and timing study that demonstrates the role of epigenetics in speciation.

Final Decision Letter:

26th August 2022

Dear Professor Miska,

We are pleased to inform you that your Article entitled "Epigenetic Divergence during Early Stages of Speciation in an African Crater Lake Cichlid Fish", has now been accepted for publication in Nature Ecology & Evolution.

Over the next few weeks, your paper will be copyedited to ensure that it conforms to Nature Ecology and Evolution style. Once your paper is typeset, you will receive an email with a link to choose the appropriate publishing options for your paper and our Author Services team will be in touch regarding any additional information that may be required

You will not receive your proofs until the publishing agreement has been received through our system

Due to the importance of these deadlines, we ask you please us know now whether you will be difficult to contact over the next month. If this is the case, we ask you provide us with the contact information (email, phone and fax) of someone who will be able to check the proofs on your behalf, and who will be available to address any last-minute problems . Once your paper has been scheduled for online publication, the Nature press office will be in touch to confirm the details.

Acceptance of your manuscript is conditional on all authors' agreement with our publication policies (see www.nature.com/authors/policies/index.html). In particular your manuscript must not be published elsewhere and there must be no announcement of the work to any media outlet until the publication date (the day on which it is uploaded onto our web site).

Please note that *Nature Ecology & Evolution* is a Transformative Journal (TJ). Authors may publish their research with us through the traditional subscription access route or make their paper immediately open access through payment of an article-processing charge (APC). Authors will not be required to make a final decision about access to their article until it has been accepted. [Find out more about Transformative Journals](https://www.springernature.com/gp/open-research/transformative-journals)

Authors may need to take specific actions to achieve [compliance](https://www.springernature.com/gp/open-research/funding/policy-compliance-faqs) with funder and institutional open access mandates. If your research is supported by a funder that requires immediate open access (e.g. according to [Plan S principles](https://www.springernature.com/gp/open-research/plan-s-compliance)) then you should select the gold OA route, and we will direct you to the compliant route where possible. For authors selecting the subscription publication route, the journal's standard licensing terms will need to be accepted, including [self-archiving-and-license-to-publish](https://www.nature.com/nature-portfolio/editorial-policies/self-archiving-and-license-to-publish). Those licensing terms will supersede any other terms that the author or any third party may assert apply to any version of the manuscript.

An online order form for reprints of your paper is available at <https://www.nature.com/reprints/author-reprints.html>. All co-authors, authors' institutions and authors' funding agencies can order reprints using the form appropriate to their

geographical region.

We welcome the submission of potential cover material (including a short caption of around 40 words) related to your manuscript; suggestions should be sent to Nature Ecology & Evolution as electronic files (the image should be 300 dpi at 210 x 297 mm in either TIFF or JPEG format). Please note that such pictures should be selected more for their aesthetic appeal than for their scientific content, and that colour images work better than black and white or grayscale images. Please do not try to design a cover with the Nature Ecology & Evolution logo etc., and please do not submit composites of images related to your work. I am sure you will understand that we cannot make any promise as to whether any of your suggestions might be selected for the cover of the journal.

You can generate the link yourself when you receive your article DOI by entering it here: <http://authors.springernature.com/share>.

[REDACTED]

P.S. Click on the following link if you would like to recommend Nature Ecology & Evolution to your librarian <http://www.nature.com/subscriptions/recommend.html#forms>

** Visit the Springer Nature Editorial and Publishing website at http://editorial-jobs.springernature.com?utm_source=ejP_NEcoE_email&utm_medium=ejP_NEcoE_email&utm_campaign=ejP_NEcoE for more information about our career opportunities. If you have any questions please click [here](mailto:editorial.publishing.jobs@springernature.com).**